# Carbon/nitrogen interactions in European forests and semi-natural vegetation. Part II: Untangling climatic, edaphic, management and nitrogen deposition effects on carbon sequestration potentials

Chris R. Flechard[1], Marcel van Oijen[2], David R. Cameron[2], Wim de Vries[3], Andreas Ibrom[4], Nina Buchmann[5], Nancy B. Dise[2], Ivan A. Janssens[6], Johan Neirynck[7], Leonardo Montagnani[8,9], Andrej Varlagin[10], Denis Loustau[11], Arnaud Legout[12], Klaudia Ziemblińska[13], Marc Aubinet[14], Mika Aurela[15], Bogdan H. Chojnicki[16], Julia Drewer[2], Werner Eugster[5], André-Jean Francez[17], Radosław Juszczak[16], Barbara Kitzler[18], Werner L. Kutsch[19], Annalea Lohila[20,15], Bernard Longdoz[21], Giorgio Matteucci[22], Virginie Moreaux[11,23], Albrecht Neftel[24], Janusz Olejnik[13,25], Maria J. Sanz[26], Jan Siemens[27], Timo Vesala[20,28], Caroline Vincke[29], Eiko Nemitz[2], Sophie Zechmeister-Boltenstern[30], Klaus Butterbach-Bahl[31], Ute M. Skiba[2] and Mark A. Sutton[2]

[1] Institut National de la Recherche Agronomique (INRA), UMR 1069 SAS, 65 rue de Saint-Brieuc, F-35042 Rennes, France
[2] Centre for Ecology and Hydrology (CEH), Bush Estate, Penicuik, EH26 0QB, UK
[3] Wageningen University and Research, Environmental Systems Analysis Group, PO Box 47, NL-6700 AA Wageningen, the Netherlands
[4] Department of Environmental Engineering, Technical University of Denmark, Bygningstorvet, DK-2800 Kgs. Lyngby, Denmark
[5] Department of Environmental Systems Science, Institute of Agricultural Sciences, ETH Zurich, LFW C56, Universitatstr. 2, CH-8092 Zurich, Switzerland
[6] Centre of Excellence PLECO (Plant and Vegetation Ecology), Department of Biology, University of Antwerp, BE-2610 Wilrijk, Belgium
[7] Research Institute for Nature and Forest (INBO), Gaverstraat 35, BE-9500 Geraardsbergen, Belgium
[8] Forest Services, Autonomous Province of Bolzano, Via Brennero 6, I-39100 Bolzano, Italy
[9] Faculty of Science and Technology, Free University of Bolzano, Piazza Università 5, I-39100 Bolzano, Italy
[10] A.N. Severtsov Institute of Ecology and Evolution, Russian Academy of Sciences, 119071, Leninsky pr.33, Moscow, Russia
[11] Institut National de la Recherche Agronomique (INRA), UMR 1391 ISPA, F-33140 Villenave d'Ornon, France
[12] INRA, BEF, F-54000 Nancy, France
[13] Department of Meteorology, Poznań University of Life Sciences, Piątkowska 94, 60-649 Poznań, Poland
[14] TERRA Teaching and Research Centre, Gembloux Agro-Bio Tech, University of Liège, Belgium
[15] Finnish Meteorological Institute, Climate System Research, PL 503, FI-00101, Helsinki, Finland
[16] Laboratory of Bioclimatology, Department of Ecology and Environmental Protection, Poznan University of Life Sciences, Piatkowska 94, 60-649 Poznan, Poland
[17] University of Rennes, CNRS, UMR 6553 ECOBIO, Campus de Beaulieu, 263 avenue du Général Leclerc, F-35042 Rennes cedex, France
[18] Federal Research and Training Centre for Forests, Natural Hazards and Landscape, Seckendorff-Gudent-Weg 8, A-1131 Vienna, Austria
[19] Integrated Carbon Observation System (ICOS ERIC) Head Office, Erik Palménin aukio 1, FI-00560 Helsinki, Finland
[20] Institute for Atmospheric and Earth System Research/Physics, Faculty of Science, POBox 68, FI-00014 University of Helsinki, Finland
[21] Gembloux Agro-Bio Tech, Axe Echanges Ecosystèmes Atmosphère, 8, Avenue de la Faculté, BE-5030 Gembloux, Belgium
[22] National Research Council of Italy, Institute for Agriculture and Forestry Systems in the Mediterranean (CNR-ISAFOM), Via Patacca, 85 I-80056 Ercolano (NA), Italy
[23] Université Grenoble Alpes, CNRS, IGE, F-38000 Grenoble, France
[24] NRE, Oberwohlenstrasse 27, CH-3033 Wohlen b. Bern, Switzerland
[25] Department of Matter and Energy Fluxes, Global Change Research Centre, AS CR, v.v.i. Belidla 986/4a, 603 00 Brno, Czech Republic
[26] Basque Centre for Climate Change (BC3), Scientific Park, Sede Building, s/n Leioa, Bizkaia, Spain
[27] Institute of Soil Science and Soil Conservation, iFZ Research Centre for Biosystems, Land Use and Nutrition, Justus Liebig University Giessen, Heinrich-Buff-Ring 26-32, D-35392 Giessen, Germany
[28] Institute for Atmospheric and Earth System Research/Forest Sciences, Faculty of Agriculture and Forestry, PO. Box 27, FI-00014 University of Helsinki, Finland
[29] Earth and Life Institute (Environmental sciences), Université catholique de Louvain, Louvain-la-Neuve, Belgium.
[30] Institute of Soil Research, Department of Forest and Soil Sciences, University of Natural Resources and Life Sciences Vienna, Peter Jordan Str. 82, A-1190 Vienna, Austria
[31] Karlsruhe Institute of Technology (KIT), Institute of Meteorology and Climate Research, Atmospheric Environmental Research (IMK-IFU), Kreuzeckbahnstr. 19, D-82467 Garmisch-Partenkirchen, Germany

*Correspondence to*: Chris R. Flechard (christophe.flechard@inra.fr)

**Abstract**. The effects of atmospheric nitrogen deposition ($N_{dep}$) on carbon (C) sequestration in forests have often been assessed by relating differences in productivity to spatial variations of $N_{dep}$ across a large geographic domain. These correlations generally suffer from covariation of other confounding variables related to climate and other growth-limiting factors, as well as large uncertainties in total (dry + wet) reactive nitrogen ($N_r$) deposition. We propose a methodology for untangling the effects of $N_{dep}$ from those of meteorological variables, soil water retention capacity and stand age, using a

mechanistic forest growth model in combination with eddy covariance $CO_2$ exchange fluxes from a Europe-wide network of 22 forest flux towers. Total $N_r$ deposition rates were estimated from local measurements as far as possible. The forest data were compared with data from natural or semi-natural, non-woody vegetation sites.

The response of forest net ecosystem productivity to nitrogen deposition ($dNEP/dN_{dep}$) was estimated after accounting for the effects on gross primary productivity (GPP) of the co-correlates by means of a meta-modelling standardization procedure,

which resulted in a reduction by a factor of about 2 of the uncorrected, apparent $dGPP/dN_{dep}$ value. This model-enhanced analysis of the C and $N_{dep}$ flux observations at the scale of the European network suggests a mean overall $dNEP/dN_{dep}$ response of forest lifetime C sequestration to $N_{dep}$ of the order of 40–50 g (C) g$^{-1}$ (N), which is slightly larger but not significantly different from the range of estimates published in the most recent reviews. Importantly, patterns of gross primary and net ecosystem productivity versus $N_{dep}$ were non-linear, with no further growth responses at high $N_{dep}$ levels

($N_{dep} > 2.5$–3 g (N) m$^{-2}$ yr$^{-1}$) but accompanied by increasingly large ecosystem N losses by leaching and gaseous emissions. The reduced increase in productivity per unit N deposited at high $N_{dep}$ levels implies that the forecast increased $N_r$ emissions and increased $N_{dep}$ levels in large areas of Asia may not positively impact the continent's forest $CO_2$ sink. The large level of unexplained variability in observed carbon sequestration efficiency (CSE) across sites further adds to the uncertainty in the dC/dN response.

**1 Introduction**

Atmospheric reactive nitrogen ($N_r$) deposition ($N_{dep}$) has often been suggested to be a major driver of the large forest carbon (C) sink observed in the Northern Hemisphere (Reay et al., 2008; Ciais et al., 2013), but this view has been challenged, both in temperate (Nadelhoffer et al., 1999; Lovett et al., 2013) and in boreal regions (Gundale et al., 2014). In principle, there is a general consensus that N limitation significantly reduces net primary productivity (NPP) (LeBauer and Treseder, 2008;

Zaehle and Dalmonech, 2011; Finzi et al., 2007). However, the measure of carbon sequestration is not the NPP, but the long term net ecosystem carbon balance (NECB; Chapin et al., 2006) or the net biome productivity at a large spatial scale (NBP; Schulze et al., 2010), whereby heterotrophic respiration ($R_{het}$) and all other C losses, including exported wood products and other disturbances over a forest lifetime, reduce the fraction of photosynthesized C (gross primary production, GPP) that is actually sequestered in the ecosystem. Indeed, it is possible to view this ratio of NECB to GPP as the efficiency of the long

term retention in the system of the assimilated C, in other words a carbon sequestration efficiency (CSE = NECB/GPP) (Flechard et al., 2020).

There is considerable debate as to the magnitude of the "fertilisation" role that atmospheric $N_r$ deposition may play on forest carbon balance, as illustrated by the controversy over the study by Magnani et al. (2007) and subsequent comments by Högberg (2007), De Schrijver et al. (2008), Sutton et al. (2008), and others. Estimates of the dC/dN response (mass C stored

in the ecosystem per mass atmospheric N deposited) vary across these studies over an order of magnitude, from 30–70 g (C) g$^{-1}$ (N) (de Vries et al., 2008; Sutton et al., 2008; Högberg, 2012), to 121 (in a model-based analysis by Dezi et al., 2010), to 200–725 (Magnani et al., 2007, 2008). Recent reviews have suggested mean dC/dN responses generally well below 100 g (C) g$^{-1}$ (N), ranging from 61–98 for above-ground biomass increment in US forests (Thomas et al., 2010), 35–65 for above-ground biomass and soil organic matter (Erisman et al., 2011; Butterbach-Bahl and Gundersen, 2011), 16–33 for the whole

ecosystem (Liu and Greaver, 2009), 5–75 (mid-range 20–40) for the whole ecosystem in European forests and heathlands (de Vries et al., 2009), and down to 13–14 for aboveground woody biomass in temperate and boreal forests (Schulte-Uebbing and de Vries, 2018), and 10–70 for the whole ecosystem for forests globally, increasing from tropical, to temperate, to boreal forests (de Vries et al., 2014a; Du and de Vries, 2018).

A better understanding of processes controlling the dC/dN response is key to predicting the magnitude of the forest C sink under global change in response to changing patterns of reactive nitrogen ($N_r$) emissions and deposition (Fowler et al., 2015). The questions of the allocation and fate of both the assimilated carbon (Franklin et al., 2012) and deposited nitrogen (Nadelhoffer et al., 1999; Templer et al., 2012; Du and de Vries, 2018) appear to be crucial. It has been suggested that $N_r$ deposition plays a significant role in promoting the carbon sink strength only if N is stored in woody tissues with high C/N ratios (>200–500) and long turnover times, as opposed to soil organic matter (SOM) with C/N ratios that are an order of magnitude smaller (de Vries et al., 2008). Nadelhoffer et al. (1999) argued on the basis of a review of $^{15}$N tracer experiments that soil, rather than tree biomass, was the primary sink for the added nitrogen in temperate forests. However, based on a recent synthesis of $^{15}$N tracer field experiments (only including measurements of $^{15}$N recovery after > 1 year of $^{15}$N addition), Du and de Vries (2018) estimated that tree biomass was the primary sink for the added nitrogen in both boreal and temperate forests (about 70%), with the remaining 30% retained in soil. At sites with elevated N inputs, increasingly large fractions are lost as nitrate ($NO_3^-$) leaching. Lovett et al. (2013) found in north-eastern US forests that added N increased C and N stocks and the C/N ratio in the forest floor, but did not increase woody biomass or aboveground NPP.

In fact, Aber et al. (1989) even predicted 30 years ago that the last stage of nitrogen saturation in forests, following long term exposure to excess $N_r$ deposition, would be characterized by reduced NPP or possibly tree death, even if during the early or intermediate stages the addition of N could boost productivity with no visible negative ecosystem impact beyond $NO_3^-$ leaching. In that initial theory, Aber et al. (1989) suggested that plant uptake was the main N sink and led to increased photosynthesis and tree growth, while N was recycled through litter and humus to the available pool; this fertilization mechanism would saturate quickly, resulting in nitrate mobility. However, observations of large rates of soil nitrogen retention gradually led to the hypothesis that pools of dissolved organic carbon in soils allowed free-living microbial communities to compete with plants for N uptake. A revision of that theory by Aber et al. (1998) hypothesized the important role of mycorrhizal assimilation and root exudation as a process of N immobilization, and suggested that the process of nitrogen saturation involved soil microbial communities becoming bacterial dominated, rather than fungal or mycorrhizal dominated in pristine soils.

Atmospheric $N_r$ deposition is rarely the dominant source of N supply for forests and semi-natural vegetation. Ecosystem internal turnover (e.g. leaf fall and subsequent decomposition of leaf litter) and mineralization of SOM provide annually larger amounts of mineral N than $N_{dep}$ (although ultimately, over pedogenic time scales much of the N contained in SOM is of atmospheric origin). In addition, resorption mechanisms help conserve within the tree the externally acquired N (and other nutrients), whereby N is re-translocated from senescing leaves to other growing parts of the tree, prior to leaf shedding, with resorption efficiencies of potentially up to 70% and larger at N-poor sites than at N-saturated sites (Vergutz et al., 2012; Wang et al., 2013). Biological $N_2$ fixation can also be significant in forests (Vitousek et al., 2002). Högberg (2012) showed for eleven European forest sites that $N_r$ deposition was a relatively small fraction (13–14% on average) of the total N supply, which was dominated by SOM mineralization (up to 15–20 g (N) m$^{-2}$ yr$^{-1}$). He further argued that there may be a correlation between soil fertility (of which the natural N supply by mineralization is an indicator) and $N_r$ deposition, since historically human populations have tended to develop settlements in areas of favourable edaphic conditions, in which over time agriculture, industry and population intensified, leading to increased emissions and deposition. Thus, an apparent effect of ambient $N_{dep}$ on current net ecosystem productivity (NEP) levels could also be related to the legacy of more than a century of $N_r$ deposition on a modified internal ecosystem cycle. Importantly, unlike other ecosystem mechanisms for acquiring N from the environment (resorption from senescing leaves, biological $N_2$ fixation, mobilization and uptake of N from soil solution or

from SOM), the nitrogen supplied from atmospheric deposition comes at little or zero energetic cost (Shi et al., 2016), especially if absorbed directly at leaf level (Nair et al., 2016).

Some previous estimates of forest dC/dN response obtained by meta-analyses of NEP or NECB across a geographic gradient did not account for the major drivers of plant growth apart from nitrogen (e.g. Magnani et al., 2007). These include climate (precipitation, temperature, photosynthetically active radiation), soil physical and chemical properties (e.g. soil drainage, depth, and water holding capacity, nutrients, pH), site history and land use. Using univariate statistics such as simple regressions of NECB as a function of $N_r$ deposition is flawed if $N_r$ deposition is co-correlated with any of these other drivers

(Fleischer et al., 2013), as can be the case in spatial gradient survey analyses across a wide geographic domain. This is because all of the variability in ecosystem C sequestration across the physical space is only allowed to be explained by one factor, $N_r$ deposition. For example, Sutton et al. (2008) showed (using forest ecosystem modelling) that the apparently large dC/dN slope in the dataset of Magnani et al. (2007) was reduced by a factor of 2–3 when accounting for climatic differences between sites, i.e. when co-varying limitations in (photosynthetic) energy and water were factored out.

Similarly, ignoring the growth stage (forest age) and the effects of management (thinning) in the analysis introduces additional uncertainty in the estimated dC/dN response. Contrasting C cycling patterns and different N use efficiencies are expected between young and mature forests. Nutrient demand is highest in the early stages of forest development (especially pole stage); a recently planted forest becomes a net C sink only after a few decades, while at maturity NPP and NEP may or may not decrease, depending on a shift in the balance between autotrophic and heterotrophic respiration ($R_{aut}$ and $R_{het}$,

respectively) and GPP (Odum, 1969; Besnard et al., 2018). Thinning can initially increase ecosystem respiration by increasing litter and SOM stocks and reducing NPP in the short term, and some biomass can be exported (tree trunks), but the ultimate effect after a year or two is to boost forest growth as thinning indirectly increases nutrient availability at the tree level by reducing plant–plant competition. Thus, the frequency and intensity of thinning will also affect long-term or lifetime NECB. Severe storms, fire outbreaks and insect infestations may have a similar effect.

Altogether, these complex interactions mean that it is far from a simple task to untangle the $N_r$ deposition effect on ecosystem C sequestration from the impacts of climatic, edaphic and management factors, when analysing data from diverse monitoring sites situated over a large geographic area (Laubhann et al., 2009; Solberg et al., 2009; Thomas et al., 2010). This is in contrast to fertilisation experiments, where the N effect can be quantified with all other variables being equal between manipulation plots (Nohrstedt, 2001; Saarsalmi and Mälkönen, 2001), although their results are only valid for the conditions

at the specific location where the experiment has been performed (Schulte-Uebbing and de Vries, 2018).

There are also potentially large uncertainties in the C and N flux measurements or model estimates used to calculate a dC/dN response. In the companion paper (Flechard et al., 2020), we presented – and discussed uncertainties in – plausible estimates of C and N budgets of 40 forests and natural or semi-natural ecosystems covering the main climatic zones of Europe (from Mediterranean to temperate to boreal, from oceanic to continental), investigated as part of the CarboEurope Integrated Project

(CEIP, 2004–2008) and the parallel NitroEurope Integrated Project (NEU, 2006–2011). The NEP budgets were based on multi-annual eddy covariance (EC) datasets following well-established protocols, and in order to better constrain the N budgets, specific local measurements of dry and wet $N_r$ deposition were made. Nitrogen losses by leaching and gaseous emissions were estimated by a combination of measurements and modelling. The data showed that observation-based GPP and NEP peaked at sites with $N_{dep}$ of the order of 2–2.5 g (N) m$^{-2}$ yr$^{-1}$, but decreased above that, and that increasingly large $N_r$

losses occurred at larger $N_{dep}$ levels, implying that the net dC/dN response was likely non-linear, in line with an overview of dC/dN response results from various approaches (De Vries et al., 2014a), possibly due to the onset of N saturation as predicted by Aber et al. (1989), and associated with enhanced acidification and increase sensitivity to drought, frost and disseases (De Vries et al., 2014b). The data also showed that at the scale of the CEIP-NEU flux tower networks, nitrogen deposition was not independent of climate, but peaked in mid-range for both mean annual temperature and precipitation,

which geographically corresponds to mid-latitude Central-Western Europe, where climate is most conducive to forest productivity and growth.

In the present paper, we further the analysis of the same CEIP-NEU observational datasets through forest ecosystem modelling, with the objective of isolating the $N_r$ deposition impact on forest productivity and C sequestration potential from the parallel effects of climate, soil water retention, and forest age and management. A mechanistic modelling framework, driven by environmental forcings, inputs, growth limitations, internal cycling and losses, was required to untangle the relationships in measurement data, because the observed dependence of $N_r$ deposition on climate, combined with the large diversity but limited number of flux observation sites, restricted the applicability and validity of multivariate statistical methods. We describe a methodology to derive, through meta-modelling, standardization factors for observation-based forest productivity metrics, in order to factor out the part of variance that was caused by influences other than $N_r$ deposition (climate, soil, stand age). This original meta-modelling approach involves running multiple simulations of a forest ecosystem model for each site of the flux tower network, using alternative climate input and soil parameter data taken from all other sites of the network, in addition to each site's own data. The model results are then analysed to determine whether conditions at each site are likely to be more, or less, favourable to forest growth and C sequestration, compared with other sites, from climatic, edaphic and age perspectives, but regardless of atmospheric N inputs. This allows the calculation of internal standardization factors, that are subsequently applied to observational flux data within the same collection of sites, aiming to account for a natural variability that may otherwise bias the analysis of a dC/dN response. Further, we examine patterns of C and N use efficiencies both at the decadal time scale of flux towers and over the lifetime of forests.

## 2 Materials and methods

### 2.1 Carbon and nitrogen datasets from flux tower sites

Ecosystem-scale carbon fluxes and atmospheric nitrogen deposition data were estimated within the CEIP and NEU networks at 31 European forests (six deciduous broadleaf forests, DBF; 18 coniferous evergreen needleleaf forests, ENF, of which seven spruce-dominated and eleven pine-dominated; two mixed needleleaf/broadleaf forests, MF; five Mediterranean evergreen broadleaf forests, EBF), and nine short natural or semi-natural (SN) vegetation sites (wetlands, peatlands, unimproved and upland grasslands) (Table S1). In the following we often adopted the terminology «observation-based» rather than simply «measured», to reflect the fact many variables such as e.g. GPP or below-ground C pools rely on various assumptions or even empirical models for their estimation on the basis of measured data (e.g. flux partitioning procedure to derive GPP from NEE; allometric relations for tree and root C stocks; spatial representativeness of soil core sampling for SOM). For convenience in this paper, we use the following sign convention for $CO_2$ fluxes: GPP and $R_{eco}$ are both positive, while NEP is positive for a net sink (a C gain from an ecosystem perspective) and negative for a net source.

The general characteristics of the observation sites (coordinates, dominant vegetation, forest stand age and height, temperature and precipitation, $N_{dep}$, inter-annual mean C fluxes) are provided in Table S1 of the Supplement. The sites, measurement methods and data sources were described in more detail in the companion paper (Flechard et al., 2020); for additional information on vegetation, soils, C and N flux results and budgets, and their variability and uncertainties across the network, the reader is referred to that paper and the accompanying supplement. Briefly, the C datasets include multi-annual (on average, 5-year) mean estimates of NEP, GPP and $R_{eco}$ (total ecosystem respiration) based on 10–20 Hz EC measurements, post-processing, spectral and other corrections, flux partitioning and empirical gap-filling (e.g. Lee et al., 2004; Aubinet et al., 2000; Falge et al., 2001; Reichstein et al., 2005; Lasslop et al., 2010). The fully analysed, validated, gap-filled and partitioned inter-annual mean $CO_2$ fluxes (NEP, GPP, $R_{eco}$), as well as the meteorological data used as ecosystem model inputs (Sect. 2.2), were retrieved from the European Fluxes Database Cluster (2012) and the NEU (2013) database. Dry deposition of reactive nitrogen was estimated by measuring at each site ambient concentrations of the dominant gas-phase ($NH_3$, $HNO_3$, $NO_2$) and aerosol phase ($NH_4^+$, $NO_3^-$) $N_r$ concentrations (data available from the NitroEurope database; NEU, 2013), and applying four different inferential models to the concentration and micro-meteorological data, as described

in Flechard et al. (2011). Wet deposition was measured using bulk precipitation samplers (NEU, 2013, with additional data retrieved from national monitoring networks and from the EMEP chemical transport model (Simpson et al., 2012).

## 2.2 Modelling of forest carbon and nitrogen fluxes and pools

### 2.2.1 General description of the BASFOR ecosystem model

The BASic FORest (BASFOR) model is a process-based, deterministic forest ecosystem model, which simulates the growth and biogeochemistry (C, N and water cycles) of temperate deciduous and coniferous stands at a daily time step (van Oijen et al., 2005; Cameron et al., 2013, 2018). Model code and documentation are available on GitHub (BASFOR, 2016). Interactions with the atmospheric and soil environments are simulated in some detail, including the role of management (thinning or pruning). BASFOR is a one-dimensional model, i.e. no horizontal heterogeneity of the forest is captured, and BASFOR does not simulate some variables which are important in forest production, such as wood quality or pests and diseases.

Nine state variables for the trees describe i) C pools: leaves, branches, stems, roots, reserves (CL, CB, CS, or collectively CLBS, CR, CRES; kg (C) $m^{-2}$); ii) N pool in leaves (NL; kg (N) $m^{-2}$); and iii) Stand density (SD, trees $m^{-2}$), tree phenology (only for deciduous trees): accumulated chill days (chillday; d) and accumulated thermal time (Tsum; °C d). Seven state variables for the soil can be divided into three categories, according to the three biogeochemical cycles being simulated: i) C pools in litter layers of the forest floor (CLITT), soil organic matter (SOM) with fast turn-over (CSOMF), SOM with slow turn-over (CSOMS) (kg (C) $m^{-2}$); ii) N pools as for C but also including mineral N (NLITT, NSOMF, NSOMS, NMIN; kg (N) $m^{-2}$); and iii) the water pool: amount of water to the depth of soil explored by the roots (WA; kg $H_2O$ $m^{-2}$ = mm) (see Table 1).

Carbon enters the system via photosynthesis, calculated as the product of photosynthetically active radiation (PAR) absorption by the plant canopy and light use efficiency (LUE). The leaf and branch pools are subject to senescence, causing carbon flows to litter. Roots are also subject to senescence, causing a flow to fast-decomposing soil organic matter. Litter carbon decomposes to fast-decomposing soil organic matter plus respiration. Fast-decomposing soil organic matter decomposes to slow-decomposing soil organic matter plus respiration. Finally, the slow organic carbon pool decomposes very slowly to $CO_2$. Nitrogen enters the system in mineral form through atmospheric deposition. Nitrogen leaves the system through leaching and through emission of $N_2O$ and NO from the soil to the atmosphere. $N_2$ losses from denitrification and biological $N_2$ fixation are not simulated. Dissolved inorganic nitrogen (DIN) is taken up by the trees from the soil, and nitrogen returns to the soil with senescence of leaves, branches and roots, and also when trees are pruned or thinned. Part of the N from senescing leaves is re-used for growth. The availability of mineral nitrogen is a Michaelis-Menten function of the mineral nitrogen pool and is proportional to root biomass. The model does not include a dissolved organic nitrogen (DON) pool and therefore does not account for the possible uptake of bio-available DON forms (e.g. amino acids, peptides) by trees. Transformation between the four soil nitrogen pools are similar to those of the carbon pools, with mineral nitrogen as the loss term. Water is added to the soil by precipitation and lost through transpiration, evaporation, and drainage. Evaporation and transpiration are calculated using the Penman equation, as functions of the radiation intercepted by soil and vegetation layer, and atmospheric temperature, humidity and wind speed. Drainage of ground water results from water infiltration exceeding field capacity of the soil.

In BASFOR, the C and N cycles are coupled in both trees and soil. The model assumes that new growth of any organ proceeds with a prescribed N/C ratio, which is species-specific but generally higher for leaves and roots than for stems and branches. If the nitrogen demand for growth cannot be met by supply from the soil, some of the foliar nitrogen is recycled until leaves approach a minimum N/C ratio when leaf senescence will be accelerated. The calculation of foliar senescence accounts for a vertical profile of nitrogen content, such that the lowest leaves have the lowest N-C ratio and senesce first. Nitrogen deficiency, as measured by foliar nitrogen content, not only increases leaf senescence, but also decreases GPP and

shifts allocation from leaves to roots. Given that foliar N content is variable in BASFOR, the litter that is produced from leaf fall also has a variable N/C ratio. When the litter decomposes and is transformed, the N/C ratio of the new soil organic matter will therefore vary too in response to the ratio in the litter. Except for woody plant parts, the C and N couplings in BASFOR vegetation and soil are based on the same generic ecophysiological assumptions as those explained in detail for grassland model BASGRA (Höglind et al. 2020).

The major inputs to the model are daily time series of weather variables (global radiation, air temperature, precipitation, wind speed and relative humidity). The last two of these are used in the calculation of potential rates of evaporation and transpiration. Soil properties, such as parameters of water retention (field capacity, wilting point, soil depth) are provided as constants. Further, the model requires time series indicating at which days the stand was thinned or pruned. The model outputs include, amongst others, the state variable for trees and soil as well as evapotranspiration (ET), groundwater recharge, canopy height (H), leaf area index (LAI), diameter at breast height (DBH), GPP, $R_{eco}$ and $R_{soil}$, NEP, N mineralisation, N leaching, NO and $N_2O$ emissions (Table 1).

*{Insert Table 1 here}*

### 2.2.2 Model implementation and calibration

BASFOR simulations of forest growth and C, N and $H_2O$ fluxes were made for all CEIP-NEU forest sites from planting (spanning the interval 1860-2002), until the end of the NEU project (2011). At a few sites, natural regeneration occurred, but for modelling purposes a planting date was assigned based on the age of the trees. Meteorological data measured at each site over several years since the establishment of the flux towers (typically 5-10 yr) were replicated backwards in time in order to generate a time series of model inputs for the whole period since planting. Assumptions were made that inter-annual meteorological variability was sufficiently covered in the span of available measurements and that the impact of climate change since planting was small and could be neglected.

The atmospheric $CO_2$ mixing ratio was provided as an exponential function of calendar year, fitted to Mauna Loa data since the beginning of records in 1958 (NOAA, 2014) and extrapolated backwards to around 1860-1900 for the oldest forests included in this study. The global $CO_2$ mixing ratio driving the model thus increased from around 290 ppm in 1900, to 315 ppm in 1958, to 390 ppm in 2010 (Fig. 1). Similarly, atmospheric $N_r$ deposition was a key input to the model and was forced to vary over the lifetimes of the planted forests; $N_{dep}$ was assumed to rise from pan-European levels well below 0.5 g (N) m$^{-2}$ yr$^{-1}$ at the turn of the 20[th] century, to increase sharply after World War II to reach an all-time peak around 1980, and to decrease subsequently from peak values by about one third until 2005-2010, at which point the NEU $N_{dep}$ estimates were obtained. We assumed that all sites of the European network followed the same relative time course of $N_{dep}$ over the course of the 20[th] century, taken from van Oijen et al. (2008), but scaled for each site using the NEU $N_{dep}$ estimates (Supplement Fig. S1).

Forest management was included as an input to the model in the form of a prescribed time course of stand density and thinning from planting to the present date. Tree density was known at all sites around the time of the CEIP-NEU projects (Table S2 in Flechard et al., 2020), but information on thinning history since planting (dates and fractions removed) was much sparser. A record of the last thinning event was available at only one third of all sites, and a knowledge of the initial (planting) density and a reasonably complete record of all thinning events were available at only a few sites. For the purposes of BASFOR modelling, we attempted to recreate a plausible density and thinning history over the lifetime of the stands. The guiding principle was that after the age of 20 years one could expect a decadal thinning of the order of 20%, following Cameron et al. (2013), while the initial reduction was 40% during the first 20 years. In the absence of an actual record of planting density (observed range: 1400-15000 trees ha$^{-1}$), a default initial value of 4500 trees ha$^{-1}$ was assumed (for around two thirds of the sites). The general principles of this default scheme were then applied to fit the available density and thinning data for each site, preserving all actual data in the time series while filling in the gaps by plausible interpolation. The density time courses thus obtained, underlying all subsequent model runs, are shown in Fig. S2.

The model was calibrated through a multiple site Bayesian calibration (BC) procedure, applied to three groups of plant functional types (PFT), based on C/N/H$_2$O flux and pool data from the CEIP-NEU databases (see Cameron et al., 2018). A total of 22 sites were calibrated, including deciduous broadleaf forests (DB1-6), evergreen needleleaf forests ENF-spruce (EN1-7), and ENF-pine (EN8-18). The model parameters were calibrated generically within each PFT group, i.e. they were not optimized or adjusted individually for each observation site. In the companion paper (Flechard et al., 2020), baseline BASFOR runs were produced for all 31 forest sites of the network, including also those stands for which the model was not calibrated, such as Mediterranean evergreen broadleaf (EB1 through EB5) and mixed deciduous/coniferous (MF1, MF2), to test the predictive capacity of the model beyond its calibration range (see Fig. 6 in Flechard et al., 2020). However, for the analyses and scenarios presented hereafter, these seven uncalibrated sites were removed from the dataset, as were two additional sites: EN9 and EN12 (EN9 because this agrosilvopastoral ecosystem called «dehesa» has a very low tree density (70 trees ha$^{-1}$; Tables S1-S2 in the Supplement to Flechard et al., 2020) and is otherwise essentially dry grassland for much of the surface area, which BASFOR cannot simulate; EN12 because this was a very young plantation at the time of the measurements, also with a very large fraction of measured NEP from non-woody biomass). All the conclusions from BASFOR meta-modelling are drawn from the remaining 22 deciduous, pine and spruce stands (sites highlighted in Table S1).

### 2.2.3 Modelling time frames

In the companion paper (Flechard et al., 2020), C and N budgets were estimated primarily on the basis of ecosystem measurements and for the time horizon of the CEIP and NEU projects (2004–2010). In this paper, BASFOR simulations of the C and N budgets for the 22 forest sites were considered both i) over the most recent 5-year period (around the time of CEIP-NEU) which did not include any thinning event and started at least 3 years after the last thinning event (referred to hereafter as «5-yr»); and ii) over the whole time span since forest establishment, referred to here as «lifetime», which ranged from 30 to 190 years across the network and reflected the age of the stand at the time of the CEIP-NEU projects. Note that the term «lifetime» in this context was not used to represent the expected age of senescence or harvest.

On the one hand, the short term (5-yr) simulations were made to evaluate cases where no disturbance by management impacted fluxes and pools over a recent period, whatever the age of the stands at the time of the C and N flux measurements (ca 2000–2010). On the other hand, the lifetime simulations represent the time-integrated flux and pool history since planting, which reflects the long-term C sequestration (NECB) potential, controlled by the cumulative impact of management (thinning), increasing atmospheric CO$_2$ mixing ratio, and changing N$_r$ deposition over the last few decades. Thinning modifies the canopy structure and therefore light, water and nutrient availability for the trees, reduces the LAI momentarily, and in theory the left-over additional organic residues (branches and leaves) could increase heterotrophic respiration and affect the NEP. However, the impact of the disturbance on NEP and R$_{eco}$ is expected to be small and short-lived (Granier et al., 2008), and a 3-year wait after the last thinning event appears to be reasonable for the modelling. The 5-yr data should in theory reflect the C/N flux observations, although there were a few recorded thinning events during the CEIP-NEU measurement period, and the thinning sequences used as inputs to the model were reconstructed and thus not necessarily accurate (Fig S2).

### 2.2.4 Modelled carbon sequestration efficiency (CSE) and nitrogen uptake efficiency (NUPE)

For both C and N, we define modelled indicators of ecosystem retention efficiency relative to a potential input level, which corresponds to the total C or N supply, calculated over both 5-yr (no thinning) and lifetime horizons to contrast short-term and long-term patterns. For C sequestration, the relevant terms are the temporal changes in carbon stocks in leaves, branches and stems (CLBS), roots (CR), soil organic matter (CSOM), and litter layers (CLITT), and the C export of woody biomass (CEXP), relative to the available incoming C from gross photosynthesis (GPP). We thus define the carbon sequestration

efficiency (CSE) as the ratio of either modelled 5-yr NEP, or modelled lifetime NECB, to modelled GPP in a given environment, constrained by climate, nitrogen availability and other factors included in the BASFOR model:

$$CSE_{5-yr} \ (no \ thinning) = {NEP_{5-yr}} \big/ {GPP_{5-yr}} \tag{1}$$

$$CSE_{lifetime} = {NECB_{lifetime}} \big/ {GPP_{lifetime}} \tag{2}$$

with $\quad NECB = \frac{d(CLBS+CR+CSOM+CLITT)}{dt} \tag{3}$

$$NECB_{5-yr} \ (no \ thinning) = NEP_{5-yr} \tag{4}$$

$$NECB_{lifetime} = NEP_{lifetime} - CEXP_{thinning} \tag{5}$$

The modelled $CSE_{5-yr}$ can be contrasted with observation based $CSE_{obs}$ ($= NEP_{obs} / GPP_{obs}$) derived from flux tower data over a similar, relatively short time period compared with a forest rotation (see Flechard et al., 2020). By extension, the $CSE_{lifetime}$ indicator quantifies the efficiency of C sequestration processes by a managed forest system, reflecting not only biological and ecophysiological mechanisms, but also the long term impact of human management through thinning frequency and severity.

For the N budget we define, by analogy to CSE, the N uptake efficiency (NUPE) as the ratio of N immobilized in the forest system to the available mineral N, i.e. the ratio of tree N uptake ($N_{upt}$) to the total $N_{supply}$ from internal SOM mineralization and N cycling processes ($N_{miner}$) and from external sources such as atmospheric N deposition ($N_{dep}$):

$$NUPE = {N_{upt}} \big/ {N_{supply}} \tag{6}$$

with $\quad N_{supply} = N_{miner} + N_{dep} \tag{7}$

$$N_{supply} \approx N_{upt} + N_{leach} + N_{emission} \tag{8}$$

The fraction of $N_{supply}$ not taken up in biomass and lost to the environment ($N_{loss}$) comprises dissolved inorganic N leaching ($N_{leach}$) and gaseous NO and $N_2O$ emissions ($N_{emission}$):

$$N_{loss} = {(N_{leach} + N_{emission})} \big/ {N_{supply}} \tag{9}$$

Note that i) NUPE is a different concept from the nitrogen use efficiency (NUE), often defined as the amount of biomass produced per unit of N taken up from the soil, or the ratio $NPP/N_{upt}$ (e.g. Finzi et al., 2007), and ii) biological $N_2$ fixation, as well as N loss by total denitrification, are not accounted for in the current BASFOR version; also, leaching of dissolved organic N and C (DON, DOC) and dissolved inorganic C (DIC) is not included either, all of which potentially impact budget calculations.

### 2.2.5 Meta-modelling as a tool to standardize EC-based productivity data

One purpose of BASFOR modelling in this study was to gain knowledge on patterns of C and N fluxes, pools and internal cycling that were not, or could not be, evaluated solely on the basis of the available measurements (for example, SOM mineralization and soil N transfer; retranslocation processes at canopy level; patterns over the lifetime of a stand). The model results were used to complement the flux tower observations to better constrain elemental budgets and assess potential and limitations of C sequestration at the European forest sites considered here. Additionally, we used meta-modelling as an alternative to multivariate statistics (e.g. stepwise multiple regression, mixed non-linear models, residual analysis) to isolate the importance of $N_r$ deposition from other drivers of productivity. This follows from the observations by Flechard et al. (2020) that i) $N_r$ deposition and climate were not independent in the dataset, and that ii) due to the large diversity of sites, the limited size of the dataset, and incomplete information on other important drivers (e.g. stand age, soil type, management), regression analyses were unable to untangle these climatic and other inter-relationships from the influence of $N_r$ deposition. BASFOR (or any other mechanistic model) is useful in this context, not so much to predict absolute fluxes and stocks, but to investigate the relative importance of drivers, which is done by assessing changes in simulated quantities when model inputs are modified. Meta-modelling involves building and using surrogate models that can approximate results from more complicated simulation models; in this case we derived simplified relationships linking forest productivity to the impact of major drivers, which were then used to harmonize observations from different sites. For example, running BASFOR for a

given site using meteorological input data from another site, or indeed from all other sites of the network, provides insight into the impact of climate on GPP or NEP, all other factors (soil, vegetation structure and age, $N_r$ deposition) being equal. Within the boundaries of the network of 22 selected sites, this sensitivity analysis provides *relative* information as to which of the 22 meteorological datasets is most, or least, favourable to growth for this particular site. This can be repeated for all sites (22*22 climate «scenario» simulations). It can also be done for soil physical properties that affect the soil water holding capacity (texture, porosity, rooting depth), in which case the result is a *relative* ranking within the network of the different soils for their capacity to sustain an adequate water supply for tree growth. The procedure for the normalization of data between sites is described hereafter.

Additional nitrogen affects C uptake primarily through releasing N limitations at the leaf level for photosynthesis (Wortman et al., 2012; Fleischer et al., 2013), which scales up to GPP at the ecosystem level. Other major factors affecting carbon uptake are related to climate (photosynthetically active radiation, temperature, precipitation), soil (for example water holding capacity) or growth stage (tree age). In the following section, we postulate that observation-based gross primary productivity (GPP$_{obs}$), which represents an actuation of all limitations in the real world, can be transformed through meta-modelling into a standardized potential value (GPP*) for a given set of environmental conditions (climate, soil, age), common to all sites, thereby enabling comparisons between sites. We define GPP* as GPP$_{obs}$ being modulated by one or several dimensionless factors ($f_X$):

$$GPP^* = GPP_{obs} \times f_{CLIM} \times f_{SOIL} \times f_{AGE} \tag{10}$$

where the standardization factors $f_{CLIM}$, $f_{SOIL}$ and $f_{AGE}$ are derived from BASFOR model simulations corresponding to the CEIP-NEU time interval around 2005–2010, as described below. The factors involved in Eq. (10) address commonly considered drivers, but not nitrogen, which is later assessed on the basis of GPP*, rather than GPP$_{obs}$. Other potentially important limitations such as non-N nutrients, soil fertility, air pollution ($O_3$), poor ecosystem health, soil acidification, etc., are not treated in BASFOR, and cannot be quantified here. Further, the broad patterns of the GPP *vs.* $N_{dep}$ relationships reported in Flechard et al. (2020), i.e. a non-linear increase and eventual saturation of GPP as $N_{dep}$ increases beyond a critical threshold, did not show any marked difference between the three forest PFT (deciduous, pine, spruce), possibly because the datasets were not large enough and fairly heterogeneous. Thus, although PFT may be expected to influence C/N interactions, we did not seek to standardize GPP with an additional $f_{PFT}$ factor.

To determine the $f_{CLIM}$ and $f_{SOIL}$ factors, the model was run multiple times with all climate and soil scenarios for the *n* (=22) sites, a scenario being defined as using model input data or parameters from another site. Specifically, for $f_{CLIM}$, the model weather inputs at each site were substituted in turn by the climate data (daily air temperature, global radiation, rainfall, wind speed and relative humidity) from all other sites; and for $f_{SOIL}$, the field capacity and wilting point parameters ($\Phi_{FC}$, $\Phi_{WP}$) and soil depth that determine the soil water holding capacity at each site (SWHC = ($\Phi_{FC}$ - $\Phi_{WP}$) x soil depth), were substituted in turn by parameters from all other sites. Values of $f_{CLIM}$ and $f_{SOIL}$ were calculated for each site in several steps, starting with the calculation of the ratios of modelled GPP from the scenarios to the baseline value GPP$_{base}$ such that:

$$X(i,j) = GPP(i,j)/GPP_{base}(i) \tag{11}$$

where i (1..*n*) denotes the site being modelled and j (1..*n*) denotes the climate data set ($j_{CLIM}$) or soil parameter set ($j_{SOIL}$) used in the scenario being simulated (see Table S2 for the calculation matrices). The value of the X (i, j) ratio indicates whether the j$^{th}$ scenario is more (> 1) or less (< 1) favourable to GPP for the i$^{th}$ forest site.

For each site, the aim of the $f_{CLIM}$ factor (and similar reasoning for $f_{SOIL}$) (Eq. (10)) is to quantify the extent to which GPP differs from a standard GPP* value that would occur if all sites were placed under the same climatic conditions. Rather than choose the climate of one particular site to normalize to, which could bias the analysis, we normalise GPP to the equivalent of a «mean» climate, by averaging BASFOR results over all (22) climate scenarios (Eq. (14)–(15)). However, since each of the scenarios has a different mean impact across all sites ( $\overline{X(j)}$ , Eq. (12)), we first normalize X(i,j) to $\overline{X(j)}$ value within each j$^{th}$ scenario (Eq. (13)):

$$\overline{X(j)} = \frac{1}{n} \sum_{i=1}^{n} X(i,j) \tag{12}$$

$$X_{norm}(i,j) = \frac{X(i,j)}{\overline{X(j)}} \tag{13}$$

The normalization of X (i, j) to $X_{norm}$ (i, j) ensures that the relative impacts of each scenario on all $n$ sites can be compared between scenarios. The final step is the averaging for each site of $X_{norm}$ (i, j) values from all scenarios (either $j_{CLIM}$ or $j_{SOIL}$) into the overall $f_{CLIM}$ or $f_{SOIL}$ values:

$$f_{CLIM}(i) = \overline{X_{norm}(i)} = \frac{1}{n} \sum_{j_{CLIM}=1}^{n} X_{norm}(i,j_{CLIM}) \tag{14}$$

$$\text{or} \qquad f_{SOIL}(i) = \overline{X_{norm}(i)} = \frac{1}{n} \sum_{j_{SOIL}=1}^{n} X_{norm}(i,j_{SOIL}) \tag{15}$$

The factors $f_{AGE}$ were determined by first normalizing modelled GPP (base run) to the value predicted at age 80, for every year of the simulated GPP time series at those $m$ (=12) mature sites where stand age exceeded 80. The age of 80 was chosen since this was the mean stand age of the whole network. The following ratios were thus calculated:

$$Y(k,yr) = GPP_{base}(k,yr)/GPP_{base}(k,80) \tag{16}$$

where k (1..$m$) denotes the mature forest site being modelled. A mean temporal curve for $f_{AGE}$ (normalized to 80 years) was calculated, to be used subsequently for all sites, after the following:

$$f_{AGE}(yr) = \left(\frac{1}{m} \sum_{k=1}^{m} Y(k,yr)\right)^{-1} \tag{17}$$

## 3 Results

### 3.1 Short term (5-yr) versus lifetime C and N budgets from ecosystem modelling

The time course of modelled (baseline) GPP, NEP and total leaching and gaseous N losses is shown in Fig. 1 for all forest sites over the 20th century and until 2010, forced by climate, increasing atmospheric $CO_2$ and by the assumed time course of $N_r$ deposition over this period (Fig. 1a). For each stand, regardless of its age and establishment date, an initial phase of around 20-25 years occurs, during which GPP increases sharply from zero to a potential value attained upon canopy closure (Fig. 1b), while NEP switches from a net C source to a net C sink after about 10 years (Fig. 1d). Initially $N_r$ losses are very large (typically of the order of 10 g (N) m$^{-2}$ yr$^{-1}$), then decrease rapidly to pseudo steady-state levels when GPP and tree N uptake reach their potential.

After this initial phase, modelled GPP increases steadily in response to increasing $N_{dep}$ and atmospheric $CO_2$, but only for the older stands established before around 1960, i.e. those stands that reach canopy closure well before the 1980's, when $N_r$ deposition is assumed to start declining. Thereafter, modelled GPP ceases to increase, except for the recently established stands that have not yet reached canopy closure. The stabilization of GPP for mature trees at the end of the 20th century in the model is likely a consequence of the effects of decreasing $N_{dep}$ and increasing $CO_2$ cancelling each other out to a large extent. In parallel, modelled total N losses start to decrease after the 1980's, even for sites long past canopy closure (Fig. 1e-f), but this mostly applies to stands subject to the largest $N_{dep}$ levels, i.e. where the historical high $N_{dep}$ of the 1980's, added to the internal N supply, were well in excess of growth requirements in the model.

*{Insert Fig. 1 here}*

These temporal interactions of differently-aged stands with changing $N_{dep}$ and $CO_2$ over their lifetimes therefore impact C and N budget simulations made over different time horizons. Modelled C and N budgets are represented schematically in Fig. 2 and Fig. 3, respectively, as «Sankey» diagrams (Matlab «*drawSankey.m*» function; Spelling, 2009) for three example forest sites (DB5, EN3, EN16), and in Fig. S3–S8 of the Supplement for all sites of the study. Each diagram represents the input, output and internal flows in the ecosystem, with arrow width within each diagram being proportional to flow. For carbon (Fig. 2 and S3–S5), the largest (horizontal) arrows indicate exchange fluxes with the atmosphere (GPP, $R_{eco}$), while the smaller (vertical) arrows indicate gains (green) or losses (red) in internal ecosystem C pools (CSOM, CBS, CR, CL, CLITT),

as well as any exported wood products (CEXP, orange). NEP is the balance of the two horizontal arrows, as well as the balance of all vertical arrows.

In the 5-yr simulations with no thinning occurring (Fig. 2-left; Fig. S3), NEP is equal to NECB, which is the sum of ecosystem C pool changes over time (= C sequestration if positive). By contrast, in the lifetime (since planting) simulations (Fig. 2-center; Fig. S4), the long-term impact of thinning is shown by the additional orange lateral arrow for C exported as woody biomass (CEXP). In this case, C sequestration or NECB no longer equals NEP, the difference being CEXP, the C contained in exported stems from thinned trees. By contrast, in the model, upon thinning the C from leaves, branches and roots join the litter layers or soil pools and is ultimately respired or sequestered. To compare between sites with different productivity levels, the lifetime data are also normalized as a percentage of GPP (Fig. 2-right; Fig. S5). The clear differences between 5-yr and lifetime C-budget simulations were: i) systematically larger GPP in recent 5-yr horizon (combined effects of age as well as $CO_2$ and $N_{dep}$ changes over time); ii) C storage in branches and stems (CBS) dominated in both cases, but CBS fractions were larger in the 5-yr horizon; iii) larger relative storage in soil organic matter (CSOM) when calculated over lifetime.

For nitrogen, by contrast to carbon, the focus of the budget diagrams is not on changes over time of the total ecosystem (tree + soil, organic + mineral) N pools. Rather, we examine in Fig. 3 and S6–S8 the extent to which $N_r$ deposition contributes to the mineral N pool (NMIN), which in the model is considered to be the only source of N available to the trees and therefore acts as a control of C assimilation and ultimately sequestration. In these diagrams for NMIN, the largest (horizontal) arrows indicate the modelled internal ecosystem N cycling terms ($N_{miner}$ from SOM mineralisation, $N_{upt}$ uptake by trees) and the secondary (vertical) arrows represent external exchange (inputs and losses) fluxes as $N_{dep}$, $N_{leach}$ and $N_{emission}$ (unit: g (N) m$^{-2}$ yr$^{-1}$). The variable NMIN describes the transient soil inorganic N pool in the soil solution and adsorbed on the soil matrix (NMIN = $NO_3^- + NH_4^+$; units g (N) m$^{-2}$). Since the modelled long term (multi-annual) changes in the transient NMIN pool are negligible compared with the magnitudes of the N input and output fluxes, the dNMIN/dt term is not represented as an arrow in the budget plots, and the total mineral $N_{supply}$ (defined as $N_{miner} + N_{dep}$) is basically balanced by N uptake ($N_{upt}$) and losses ($N_{leach} + N_{emission}$) (Eq. (8)). Modelled N budgets were calculated for a 5-yr time horizon (Fig. 3-left; Fig. S6) and for the whole time period since the forest was established (lifetime, Fig. 3-center; Fig. S7). Lifetime data were also normalized as a percentage of $N_{supply}$ (Fig. 3-right; Fig. S8). The clear differences between 5-yr and lifetime N-budget simulations are: i) $N_{loss}$ and especially $N_{leach}$ were significantly larger over the stand lifetime since planting; ii) $N_{upt}$ was a larger fraction of total $N_{supply}$ over the recent 5-yr period.

*{Insert Fig. 2 here}*

*{Insert Fig. 3 here}*

## 3.2 Contrasted efficiencies of carbon sequestration and nitrogen uptake

Collectively, the changes in the ecosystem C pools, especially the increases in stems and branches (CBS), roots (CR) and soil organic matter (CSOM) represent roughly 20–30% of GPP for both 5-yr and lifetime simulations (Fig. 2, S3–S5). By contrast, the analogous term for nitrogen, the $N_{upt}$ fraction of total $N_{supply}$, is a much more variable term, both between sites of the network and between the 5-yr and lifetime simulations (Fig. 3, S6–S8). Modelled lifetime CSE and NUPE values are compared in Fig. 4 with the 5-yr values, as a function of stand age, indicating that (i) the older forests of the network (age range ~80–190 yrs) tend to have larger NUPE than younger or middle aged forests (~30–60 yrs), but (ii) the difference in NUPE between the two age groups is much clearer if NUPE is calculated over the whole period since planting (lifetime). As shown in Fig. 1, BASFOR predicts large N losses in young stands (<20-25 years), in which lower N demand by a smaller living biomass, combined in the early years with enhanced $N_{miner}$ from higher soil temperature (canopy not yet closed) and with a larger drainage rate (smaller canopy interception of incident rainfall), all lead to larger NMIN losses. The 22 forests sites of this study were past this juvenile stage, but observation (ii) is a mathematical consequence of high N losses during the forest's early years having a larger impact on lifetime calculations in middle-aged than mature forests. NUPE tends to reach

70-80% on average after 100 years and is smaller calculated from lifetime than from a 5-yr thinning-free period. For forests younger than 60 years, lifetime NUPE is only around 60%.

Modelled carbon sequestration efficiency is less affected than NUPE by forest age (CSE range ~15–30%) (Fig. 4). There is a tendency for 5-yr (thinning-free) CSE to decrease from ~30% to ~20% between the ages of 30 and 190 years. This means that, in the model, $R_{eco}$ in 30 to 60-yr old stands represents a smaller fraction of GPP than in mature stands. From Eq. (1) it can readily be shown that CSE = 1 - $R_{aut}$/GPP - $R_{het}$/GPP, which is roughly equivalent to 0.5 – $R_{het}$/GPP, since in the model $R_{aut}$ is constant and approximately 0.5 for all species. By contrast, BASFOR predicts that the $R_{het}$/GPP ratio increases steadily with age at each site, after the initial establishment phase (Fig. S12a). This induces a decline in modelled CSE from 25-35% in the age class 30-60 yrs down to around 20-25% for the older forests (Fig. S12b). This also implies a non-linearity developing over time of GPP versus soil and litter layers C pools, since $R_{het}$ is assumed to a linear function of fast and slow C pools in litter layers and SOM. Lifetime CSE values are slightly smaller than 5-yr values: the difference corresponds to cumulative CEXP over time, but the trend with age is weaker than for 5-yr CSE. The relatively narrow range of modelled 5-yr CSE values (20–30%) is in sharp contrast to the much wider range of observation-based $CSE_{obs}$ values (from -9% to 61%), likely reflecting some limitations of the model and possibly also measurement uncertainties, as discussed in Flechard et al. (2020).

*{Insert Fig. 4 here}*

Beyond the overall capacity of the forest to retain assimilated C (as quantified by CSE), the modelled fate of sequestered C, the simulated ultimate destination of the C sink, is also a function of forest age and of the time horizon considered (Fig. 5). The fraction of NECB sequestered in above-ground biomass (CLBS) over a recent 5-yr horizon is on average around 80% (*versus* around 10% each for CR and CSOM) and not clearly linked to forest age, i.e. the model does not simulate any slowing down with age of the annual growth of above-ground biomass. Calculated over lifetimes, the dominant ultimate destination of sequestered C remains CLBS. However, this fraction is smaller (50–60%) in old-growth forests than in younger stands (60–80%), since a larger cumulative fraction of above-ground biomass (timber) will have been removed (CEXP) by a lifetime of thinnings in a mature forest, while the cumulative gain in CSOM is not repeatedly depleted, but on the contrary enhanced, by thinnings (since the model assumes bole removal only, not total tree harvest). Modelled annual C storage to the rooting system clearly declines with age and is an increasingly marginal term over time (although the absolute CR stock itself keeps increasing over time, except when thinning transfers C from roots to SOM).

*{Insert Fig. 5 here}*

### 3.3 Standardization of observation-based GPP through meta-modelling

The purpose of meta-modelling was to standardize observation-based $GPP_{obs}$ into GPP* through model-derived factors that separate out the effects of climate, soil and age between monitoring sites (Eq. (10)), so that the importance of $N_r$ can be isolated. The sensitivity of modelled GPP to climate and soil physical properties was tested through various model input and parameter scenarios, allowing standardization factors $f_{CLIM}$ and $f_{SOIL}$ to be calculated as described in Methods (Eq. (11)–(15)) and Table S2 in the Supplement. The resulting distributions of all simulations for all sites were represented in Fig. 6 as «violin» plots (Matlab «*distributionPlot*.m» function; Dorn, 2008) for the climate-only and soil-only scenarios ($n^2 = 484$ simulations each), and also combined climate*soil scenarios ($n^3 = 10648$ simulations). For each site, the scenarios explore the modelled response of ecosystem C dynamics to a range of climate and soil forcings different from their own. The size and position of the violin distribution indicate, respectively, the degrees of sensitivity to- and limitation by- climate, soil, or both; a site is especially limited by either factor (relative to the other sites of the network) when the baseline/default run ($GPP_{base}$) is located in the lower part of the distribution.

Similarly, to account for the effect of tree age, the $f_{AGE}$ factor was calculated following Eq. (17), whereby the time series for the ratio of modelled $GPP_{base}$(yr) to $GPP_{base}$(80) (Eq. (16)) followed broadly similar patterns for the different sites (Fig. 7), with values mostly in the range 0.6–0.8 at around age 40, crossing unity at 80 and levelling off around 1.2–1.4 after a century.

Some of the older sites (e.g. EN2, EN6, EN15) showed a peak followed by a slight decrease in modelled GPP, but not at the same age. This was due to the peak in $N_{dep}$ in the early 1980's in Europe (Fig. S1), with the $N_{dep}$ peak occurring at different ontogenetic stages in the differently aged stands. By calculating a mean $f_{AGE}$ factor across sites the peak $N_{dep}$ effect was

smoothed out (Fig. 7). Thus, for a younger forest, the multiplication of $GPP_{obs}$ by $f_{AGE}$ (>1) simulated the larger GPP* that one could expect for the same site at 80 yr; conversely, the GPP* a mature forest (>100 yr) would be reduced compared with $GPP_{obs}$.

*{Insert Fig. 6 here}*

*{Insert Fig. 7 here}*

The combined modelled effects of climate, soil, and stand age on GPP are summarized in Fig. 8. Values for both $f_{CLIM}$ and $f_{SOIL}$ are mostly in the range 0.7–1.5, and are predictably negatively correlated to mean annual temperature (MAT) and soil water holding capacity (SWHC), respectively (Fig. 8a). A value well above 1 implies that $GPP_{obs}$ for one site lies below the value one might have observed if climate or SWHC had been similar to the average of all other sites of the network. In other words this particular site was significantly limited by climate, SWHC, or both, relative to the other sites. Conversely, a value

below 1 means that GPP at the site was particularly favoured by weather and soil. Climate or soil conditions at some sites have therefore the potential to restrict GPP by around one third, while other climates or soil conditions may enhance GPP by around one third, compared with the average conditions of the whole network. Applying the $f_{CLIM}$, $f_{SOIL}$ and $f_{AGE}$ multipliers to $GPP_{obs}$ (Eq. (10)) provides a level playing field (GPP*) for later comparing sites with respect to $N_r$ deposition, but also increases the scatter and noise in the relationship of GPP* to $N_{dep}$, particularly with the introduction of $f_{AGE}$ (Fig. 8b).

*{Insert Fig. 8 here}*

## 3.4 Response of gross primary productivity to $N_r$ deposition

The standardized forest GPP* values, i.e. GPP*($f_{CLIM}$), GPP*($f_{CLIM}$ x $f_{SOIL}$) and GPP*($f_{CLIM}$ x $f_{SOIL}$ x $f_{AGE}$), show in the $N_{dep}$ range 0–1 g (N) m$^{-2}$ yr$^{-1}$ a much less steep relationship to $N_{dep}$ than the original $GPP_{obs}$ (Fig. 8b). This supports the hypothesis that GPP at the lower $N_{dep}$ sites is also limited by climate and/or soil water availability. In Fig. 8b, 2$^{nd}$-order polynomials are

fitted to the data to reflect the strong non-linearity present in $GPP_{obs}$, driven especially by the 4 highest $N_{dep}$ sites (>2.5 g (N) m$^{-2}$ yr$^{-1}$ at EN2, EN8, EN15 and EN16). The non-linearity (magnitude of the 2$^{nd}$-order coefficient) is reduced by the introduction of $f_{CLIM}$ and $f_{SOIL}$, while $f_{AGE}$ has a small residual impact on the shape of the regression. Due to this non-linear behaviour, the $dGPP/dN_{dep}$ responses decrease with $N_{dep}$ for the observation-based GPP, but less so for the standardized GPP* estimates (Fig. 8c). Values of $dGPP_{obs}/dN_{dep}$ (calculated for each $N_{dep}$ level by the slope of the tangent line to the quadratic

fits of Fig. 8b) range from around 800 g (C) g$^{-1}$ (N) at the lowest $N_{dep}$ level down to negative values at the highest $N_{dep}$ sites; for the standardized GPP* accounting for all climate, soil and age effects, this range is much narrower, from around 350 down to near 0 g (C) g$^{-1}$ (N).

Average $dGPP/dN_{dep}$ figures that are representative of this set of forest sites are given in the upper part of Table 2, either calculated over the whole range of 22 sites, or for a subset of 18 sites that excludes the four highest deposition sites (>2.5 g

(N) m$^{-2}$ yr$^{-1}$ ). If all modelled sites are considered, the mean $dGPP/dN_{dep}$ regression slopes are smaller (190–260 g (C) g$^{-1}$ (N)), being influenced by the reductions in GPP at very high $N_{dep}$ levels, possibly induced by the negative side effects of N saturation. If these four sites are excluded, the mean $dGPP/dN_{dep}$ is larger (234–425 g (C) g$^{-1}$ (N)), reflecting the fact that healthier, N-limited forests are more responsive to N additions. In this subset of 18 sites, the effects of climate, soil and stand age account for approximately half of GPP (the mean $dGPP/dN_{dep}$ response changes from 425 to 234 g (C) g$^{-1}$ (N)). For

comparison, Table 2 also provides the values of $dGPP_{obs}/dN_{dep}$ obtained directly through simple linear regression for all forest sites and for the semi-natural vegetation sites, with values of the same order (432 and 504 g (C) g$^{-1}$ (N), respectively) if the high N deposition sites ($N_{dep}$ > 2.5 g (N) m$^{-2}$ yr$^{-1}$) are removed.

As a further comparison, an additional BASFOR modelling experiment is shown in Fig. 9a, in which GPP at all sites is simulated in a range of $N_{dep}$ scenarios (0, 0.1, 0.2, 0.5, 1, 1.5, 2, 2.5, 3, 3.5, 4 and 4.5 g (N) m$^{-2}$ yr$^{-1}$, constant over lifetime) to

substitute for the actual $N_{dep}$ levels of each site. Around half the sites show a steadily increasing (modelled) GPP as $N_{dep}$ increases over the whole range 0–4.5 g (N) $m^{-2}$ $yr^{-1}$, with broadly similar slopes between sites; while the other half levels off and reaches a plateau at various $N_{dep}$ thresholds, indicating that beyond a certain level $N_{dep}$ is no longer limiting, according to the model. For comparison with the dC/dN responses calculated previously for $GPP_{obs}$ and GPP* in Fig. 8b-c and Table 2, we derive a mean modelled $dGPP/dN_{dep}$ response from a linear regression of Fig. 9a data over the range 0–2.5 g (N) $m^{-2}$ $yr^{-1}$ (i.e. excluding the highest deposition levels). This yields a mean $dGPP/dN_{dep}$ slope across all sites of 297 (273–322) g (C) $g^{-1}$ (N) for the $N_{dep}$ model experiment, only marginally larger than the three GPP* average slopes of Table 2. Note that in Fig. 8b, the response of GPP* to $N_{dep}$ is calculated *between sites* of the network, while in Fig. 9a the GPP to $N_{dep}$ response is calculated *within each site* from the model scenarios, then averaged across all sites.

*{Insert Fig. 9 here}*

### 3.5 Response of net ecosystem productivity to $N_r$ deposition

Similarly to GPP, the NEP and NECB responses to $N_{dep}$ cannot be reliably inferred directly from EC-flux network data given the large variability between sites in climate, soil type, age and other constraints to photosynthesis and ecosystem respiration. However, plausible estimates can be obtained by applying a range of mean CSE indicators (as defined previously) to project the normalized GPP* responses to $N_{dep}$ (Table 2). Carbon sequestration efficiencies for forests are confined to a narrow range (17–31% of GPP, average $\mu$=22%, standard deviation $\sigma$=4%) in model simulations over 5-yr (no thinning) time horizons (Fig. 4); they vary considerably more in EC-based observations (range -9 to 61%, $\sigma$=17%), but with a similar mean ($\mu$=25%). CSE metrics express the GPP fraction not being respired ($R_{eco}$) or exported (CEXP) out of the ecosystem. Multiplied by the $dGPP/dN_{dep}$ slope they provide estimates of the net ecosystem C gain per unit N deposited (Table 2).

Short-term (5-yr) mean estimates for NEP responses, based on average CSE from both observations ($CSE_{obs}$) and modelling ($CSE_{5-yr}$), and accounting for GPP climate/soil/age normalization, range from 41 to 47 g (C) $g^{-1}$ (N), averaged over all sites, or 51 to 57 g (C) $g^{-1}$ (N) removing the four highest $N_{dep}$ sites (middle part of Table 2). Predictably, lifetime estimates for $dNECB/dN_{dep}$ responses are about 20% smaller, on the order of 34–42 g (C) $g^{-1}$ (N). For comparison, the mean 5-yr $dNEP/dN_{dep}$ obtained directly by BASFOR modelling of $N_{dep}$ scenarios for all sites (Fig. 9b) was larger (76 ± 7 g (C) $g^{-1}$ (N) ) than the measurement-based, model-corrected estimates of Table 2.

If the forest NEP response to $N_{dep}$ is calculated directly through simple linear or quadratic regression of $NEP_{obs}$ *vs.* $N_{dep}$ (bottom part of Table 2), therefore not including any standardization of the data, the dC/dN slope is much larger (178–224 g (C) $g^{-1}$ (N) ) within the $N_{dep}$ range 0–2.5 g (N) $m^{-2}$ $yr^{-1}$. If all forest sites are considered (including N-saturated sites with $N_{dep}$ up to 4.3 g (N) $m^{-2}$ $yr^{-1}$), the dC/dN slope is much smaller (71–108 g (C) $g^{-1}$ (N)), but this only reflects the reduced NEP observed at those elevated $N_{dep}$ sites (see Fig. 4c in Flechard et al., 2020), with altogether very large scatter and very small $R^2$. Equivalent figures for (not standardized) semi-natural NEP *vs.* $N_{dep}$ appear to be significantly smaller (34–89 g (C) $g^{-1}$ (N)) than in forests.

If the meta-modelling standardization procedure for climate, soil and age is attempted (for comparison only) directly on NEP, as opposed to the preferred procedure using GPP (Eq. (10)–(17)), the simulated $f_{CLIM}$, $f_{SOIL}$ and $f_{AGE}$ reduce the NEP response to $N_{dep}$ by only 18%, from 178 down to 146 g (C) $g^{-1}$ (N) (bottom part of Table 2), while the equivalent reduction for GPP was 45%. The resulting figure (112–146 g (C) $g^{-1}$ (N) ) is likely much over-estimated, around factor of 2–3 larger than those obtained through the stepwise method using CSE * $dGPP/dN_{dep}$. Standardization factors derived from BASFOR meta-modelling are more reliable for GPP than for NEP, since model performance is significantly better for GPP than for $R_{eco}$ and hence NEP (Fig. 6 in Flechard et al., 2020).

*{Insert Table 2 here}*

## 4 Discussion

### 4.1 A moderate non-linear response of forest productivity to $N_r$ deposition

The C sequestration response to $N_{dep}$ in European forests was derived using a combination of flux tower-based C and N exchange data and process-based modelling, while a number of previous studies have been based on forest inventory methods and stem growth rates (e.g. de Vries et al.2009; Etzold et al., 2014). The main differences with previous meta-analyses that were also based on EC-flux datasets (e.g. Magnani et al., 2007; Fleischer et al., 2013; Fernández-Martínez et al., 2014, 2017), were that i) we derived total $N_{dep}$ from local measurements of the wet and dry fractions as opposed to regional/global CTM outputs; ii) we untangled the $N_{dep}$ effect from climatic, soil and other influences by means of a mechanistic model, not through statistical methods; and iii) in Flechard et al. (2020) we estimated ecosystem-level N, C and GHG budgets calculated through a combination of local measurements, mechanistic and empirical models, and database and literature data mining.

Our most plausible estimates of the dC/dN response of net productivity over the lifetime of a forest are of the order of 40–50 g (C) g$^{-1}$ (N) on average over the network of sites included in the study (Table 2). Such values are broadly in line with the recent reviews by Erisman et al. (2011) and by Butterbach-Bahl et al. (2011) (range 35–65 g (C) g$^{-1}$ (N) ), but slightly larger than estimates given in a number of other studies (e.g. Liu and Greaver, 2009; de Vries et al., 2009, 2014a). Given the considerable uncertainty attached to these numbers (Table 2), they cannot be considered significantly different from any of those earlier studies. The meta-modelling-based approach we describe for normalizing forest productivity data to account for differences in climate, soil and age among sites, reduces the net productivity response to $N_{dep}$ by roughly 50%, which is of the same order as the results (factor of 2–3 reduction) of a similar climate normalization exercise by Sutton et al. (2008). This means that not accounting for inter-site differences would have led to an over-estimation of the dC/dN slope by a factor of 2.

Observations and model simulations both indicate that the $N_{loss}$ fraction of $N_{supply}$ increases with $N_{dep}$, consistent with widespread observations of increasing $NO_3^-$ leaching above $N_{dep}$ thresholds as low as 1.0 g (N) m$^{-2}$ yr$^{-1}$ in European forests (Dise and Wright, 1995; De Vries et al, 2007; Dise et al., 2009), and exacerbated by large C/N ratios (> 25) in the organic horizons (Gundersen et al., 1998; MacDonald et al., 2002). Higher thresholds for $N_{dep}$ around 2.5 g (N) m$^{-2}$ yr$^{-1}$ (Dise and Wright, 1995; Van der Salm et al., 2007) typically indicate advanced saturation stages.

Thus, at many sites but especially those with $N_{dep}$ > 1.5–2 g (N) m$^{-2}$ yr$^{-1}$, N availability is not limiting forest growth. In such cases it becomes meaningless to try to quantify a N fertilisation effect. Indeed, despite large uncertainties in measured data and in model-derived normalization factors, the non-linear trend is robust, with dC/dN values tending to zero in N-saturated forests (>2.5–3 g (N) m$^{-2}$ yr$^{-1}$). In their review paper De Vries et al. (2014a) gave a range of $N_{dep}$ levels varying between 1.5–3 g (N) m$^{-2}$ yr$^{-1}$ beyond which growth and C sequestration were not further increased or even reversed, as predicted in classical N saturation theory by Aber et al. (1989, 1998). These findings suggest that in areas of the world where $N_{dep}$ levels are larger than 2.5–3 g (N) m$^{-2}$ yr$^{-1}$, which now occur increasingly in Asia, specifically in parts of China, Japan, Indonesia, and India (Schwede et al., 2018), the forecast increased $N_r$ emissions and increased $N_{dep}$ levels may thus not have a positive impact on the continent's land based $CO_2$ sink. Data treatment and selection in our dataset (e.g. removal of N-saturated forests) strongly impacted the plausible range of dC/dN responses (Table 2) derived from the original data. The non-linearity of ecosystem productivity relationships to $N_{dep}$ (Butterbach-Bahl and Gundersen, 2011; Etzold et al., 2014) limits the usefulness and significance of simple linear approaches. These data suggest that there is no single dC/dN figure applicable to all ecosystems, that the highly non-linear response depends on current and historical $N_{dep}$ exposure levels, and on the degree of N saturation (Aber et al., 1989, 1998), although other factors than N, discussed later, may also be involved.

For the short semi-natural vegetation sites, included in the study as a non-fertilised, non-woody contrast to forests, the apparent impact of $N_{dep}$ on $GPP_{obs}$ was of the same order as in forests, but likely much smaller than in forests when considering $NEP_{obs}$ (Table 2). This is in principle consistent with the hypothesis (de Vries et al., 2009) that the ecosystem dC/dN response may be larger in forests due to the large C/N ratio (200–500) of above-ground biomass (stems and branches), where much of the C storage occurs (up to 60–80% according to BASFOR, Fig. 5); whereas in semi-natural ecosystems C

storage in SOM dominates, with a much lower C/N ratio (10–40). However, this comparison of semi-natural *versus* forests is based on NEP$_{obs}$ that was not normalized for inter-site climatic and edaphic differences, since no single model was available to carry out a meta-modelling standardization for all the different semi-natural ecosystem types (peatland, moorland, fen, grassland), and therefore these values must be regarded as highly uncertain.

## 4.2 Limitations and uncertainties in the approach for quantifying the dC/dN response

Monitoring atmospheric gas-phase and aerosol N$_r$ contributed to reducing the large uncertainty in total N$_r$ deposition at individual sites, because dry deposition dominates over wet deposition in most forests (Flechard et al., 2020), except at sites a long way from sources of atmospheric pollution, and because the uncertainty in dry deposition and its modelling is much larger (Flechard et al., 2011; Simpson et al., 2014). However, despite the considerable effort involved in coordinating the continental-scale measurement network (Tang et al., 2009), the number of forest sites in this study (31) was relatively small compared with other studies based on ICP (de Vries et al., 2009; ICP, 2019) or other forest growth databases, or global-scale FLUXNET data (hundreds of sites worldwide; see Burba, 2019). Thus, the gain in precision of N$_{dep}$ estimates from local measurements was offset by the smaller population sample size. Nonetheless this study does show the added value of the N$_r$ concentration monitoring exercise and the need to repeat and extend such initiatives.

Understanding, quantifying and reducing all uncertainties leading up to dC/dN estimates are key issues to explore. Apart from measurement uncertainties in N$_r$ deposition and losses, and in the C balance based on EC measurements, analysed in the companion paper, the major difficulties that arose when assessing the response to N$_{dep}$ of forest productivity included:

i) The heterogeneity of the population of forests, climates and soils in the network, and the large number of potential drivers relative to the limited number of sites, hindered the use of a straightforward, regression-based analysis of observational data without a preliminary (model-based) harmonization;

ii) The model-based normalization procedure for GPP, used to factor out differences in climate, soil and age among sites, significantly amplified the noise in C/N relationships, an indication that the generalized modelled effects may not apply to all individual sites and that other important ecological determinants affecting forest productivity are missing in the BASFOR model;

iii) The EC measurement-based ratio of R$_{eco}$ to GPP (=1-CSE) was very variable among forests (Flechard et al., 2020) and this high variability cannot be explained or simulated by the ecosystem model we used, i.e. more complex model parameterizations of R$_{aut}$ and R$_{het}$ may be required to better represent the diversity of situations and processes;

iv) Nitrogen deposition likely contributes a minor fraction (on average 20% according to the model) of total ecosystem N supply (heavily dominated by soil organic N mineralization), except for the very high deposition sites (up to 40%). The fraction of N$_{dep}$/N$_{supply}$ may even be smaller considering the pool of DON (not included in BASFOR), from which bio-available organic N forms may be taken up by trees in significant quantities in non-fertile, acidic organic soils (Jones and Kielland, 2002; Warren, 2014; Moreau et al., 2019). Thus, in many cases the N$_{dep}$ fertilisation effect may be marginal and difficult to detect, because it may be smaller than typical measurement uncertainties and noise in C and N budgets. Conversely, the effect may be delayed and may manifest even after N$_r$ deposition levels have decreased, as the past N accumulation in soil may support later growth through enhanced N supply.

v) Non-linear biological controls that affect C/N relations but are not explicitly considered in the model. For example, BASFOR does consider that N addition can reduce below-ground C allocation (e.g. Högberg et al., 2010), resulting in decreased soil R$_{aut}$ and R$_{het}$ (Janssens et al., 2010), but does not account for the possible consequences of a stimulation of wood cell formation from mid-summer onwards and a delay in the cessation of tracheid production in late season (Kalliokoski et al., 2013).

A further limitation to our estimates of the dC/dN response, based on the analysis of the spatial (inter-site) variability in C and N fluxes, is that these forests are not in steady state with respect to $N_r$ deposition and ambient $CO_2$. Some stands have been affected by, and may be slowly recovering from, excess $N_r$ deposition in the second half of the 20[th] century; while the more remote sites may always have been N-limited. Figure 1 showed that the modelled GPP of the older forests increased through most of the 20[th] century, but stabilized when $N_{dep}$ started to decrease after the 1980's, while total N losses also

declined over the last 2-3 decades. This is consistent with observations of decreasing N (nitrate) leaching at long term study sites in N-E USA (Goodale et al., 2003; Bernal et al., 2012) and N Europe (Verstraeten et al., 2012; Johnson et al., 2018; Schmitz et al., 2019).

In our model analysis, the declining trend in $N_r$ deposition appears to be the primary driver for the modelled reduced N losses since the 1980's. This can be inferred from model input-sensitivity scenario runs shown in Fig. S9-S11 of the Supplement. In

Fig. S9, a constant $CO_2$ mixing ratio of 310 ppm (i.e. the mean value over the period 1900-2010), used instead of the exponential increase since the 19[th] century, does not greatly alter overall productivity patterns, nor the decreasing trend in N losses over the period 1980-2010 (Fig S9e-f), compared with the baseline run (Fig. 1). By contrast, in scenarios shown in Fig. S10-S11, the assumed constant $N_{dep}$ levels at all sites of 1.5 and 3.0 g (N) $m^{-2}$ $yr^{-1}$, respectively, together with the exponential $CO_2$ increase, remove the decreasing trends in $N_r$ losses over the period 1980-2010. Meanwhile, in constant $N_{dep}$ scenarios the

increase in GPP over the whole period is fairly monotonous, in response to a steadily increasing $CO_2$ (Fig. S10b-c), without the inflexion point around 1980 simulated in the baseline run (Fig 1b-d). In real-life stands, however, decadal decreases in N losses or exports have been observed without any significant reductions in $N_{dep}$ (Goodale et al., 2003). Other potential factors such as increased denitrification, longer growing season, plant N accumulation, changes in soil hydrological properties or temperature, historical disturbances, may also play a role (Bernal et al., 2012). Many such factors are not considered in our

model, and neither is long term climate change.

The EC-based flux data suggest that the $N_{dep}$ response of forest productivity is clearer at the gross photosynthesis level, in patterns of (normalized) GPP differences among sites, than at the NEP level, where very large differences in CSE among sites lead to a de-coupling of $N_{dep}$ and NEP. The response of GPP to $N_{dep}$ appeared to be reasonably well constrained by both EC flux measurements and BASFOR modelling, which is why we chose to normalize GPP, not NEP. The significantly better

model performance obtained for GPP than for $R_{eco}$ and NEP (Fig. 6 in Flechard et al., 2020) likely reveals a relatively poor understanding and mathematical representation of $R_{eco}$ (especially for the soil heterotrophic and autotrophic components)t, and the factors controlling their variability among sites. The large unexplained variability in CSE and C sequestration potentials may also involve other limiting factors that could not be accounted for in our measurement/model analysis, since they are not treated in BASFOR. Such factors may be related to soil fertility, internal N supply, ecosystem health, tree

mortality, insect or wind damages in the previous decade, incorrect assumptions on historical forest thinning, all affecting general productivity patterns. Since the observed variability in CSE is key to understanding and quantifying the real-world NEP response to $N_{dep}$ (beyond the relatively well constrained response of GPP in the model world), we explore some of the main issues in the following sections.

### 4.3 What drives the large variability in carbon sequestration efficiency?

Carbon sequestration efficiency metrics are directly and negatively related to the ratio of $R_{eco}$ to GPP, expressing the likelihood that one C atom fixed by photosynthesis will be sequestered in the ecosystem. Earlier FLUXNET-based statistical meta-analyses have demonstrated that although $R_{eco}$ is strongly dependent on temperature on synoptic or seasonal scales (Mahecha et al., 2010; Migliavacca et al., 2011), GPP is the key determinant of spatial variations in $R_{eco}$ (Janssens et al., 2001; Migliavacca et al., 2011; Chen et al., 2015), and further, that the fraction of GPP that is respired by the ecosystem is

highly variable (Fernández-Martínez et al., 2014), and more variable than in current model representations. We have used three different CSE indicators, averaged across all sites, to derive a NEP/$N_{dep}$ response from model-standardized GPP* data (Table 2). Values of $CSE_{obs}$ varied over a large range among sites (-9 to 61%, Fig. 10). Some of the variability might be due

to measurement errors, but small (<10%) or large (>40%) CSE$_{obs}$ values could also genuinely reflect the influence or the absence of ecological limitations related to nutrient availability or vegetation health.

### 4.3.1 From nutrient limitation to nitrogen saturation

Can nutrient limitation (nitrogen or otherwise) impact ecosystem carbon sequestration efficiency? Soil fertility has been suggested to be a strong driver at least of the forest biomass production efficiency (BPE), defined by Vicca et al. (2012) as the ratio of biomass production to GPP, with BPE increasing in their global dataset of 49 forests from 42% to 58% in soils with low- to high-nutrient availability, respectively. The study by Fernández-Martínez et al. (2014) of 92 forest sites around the globe reported a large variability in CSE (=NEP/GPP calculated from FLUXNET flux data), which they suggest is strongly driven by ecosystem nutrient availability (ENA), with CSE levels below 10% in nutrient-poor forests and above 30% in nutrient-rich forests. The range of CSE values derived from flux measurements in our study (CSE$_{obs}$ in Table 2) was similarly large, even though all of our sites were European and our dataset size was one third of theirs (N=31, of which 26 sites in common with Fernández-Martínez et al., 2014). We did not attempt in this study to characterize a general indicator of ENA beyond total N$_r$ deposition; but if we use the high, medium or low (H, M, L) scores of ENA attributed to each site through factor analysis of nutrient indicators by Fernández-Martínez et al. (2014), we find that the H group (7 sites) has a mean CSE$_{obs}$ of 32% (range 16–48%), the M group is slightly higher (7 sites, mean 39%, range 21–61%), while the L group has indeed a significantly smaller mean CSE$_{obs}$ of 14% (12 sites, range -9 to 38 %). Interestingly, the mean N$_{dep}$ levels for each group are H = 1.5 (range 0.5–2.3) g (N) m$^{-2}$ yr$^{-1}$ , M = 2.1 (range 1.1–4.2) g (N) m$^{-2}$ yr$^{-1}$ and L = 1.3 (range 0.3–4.1) g (N) m$^{-2}$ yr$^{-1}$, i.e. the highest mean CSE$_{obs}$ of the three groups is found in the group with the highest mean N$_{dep}$ (M).

The nutrients and other indicators of fertility considered by Fernández-Martínez et al. (2014) included, in addition to N, P, soil pH, C/N ratios and cation exchange capacity, as well as soil texture and soil type. However, very few sites were fully documented (see their Supplement Table S1), data were often qualitative, and other key nutrients were not included in the analysis (K, Mg and other cations; S also has been suggested to have become a limiting factor for forest growth following emission reductions, see Fernández-Martínez et al., 2017). The extent to which the overall fertility indicator quantified by ENA was driven by nitrogen in the Fernández-Martínez et al. (2014) factor analysis is not evident. At sites where other nutrients are limiting, the response to N additions would be small or negligible regardless of whether N itself is limiting. This places severe constraints on the interpretation of productivity data in response to N$_{dep}$, since most current models, which do not account for other nutrient limitations, cannot be called upon to normalize for differences between sites.

The impact of the fertility classification on CSE of the sites included in Fernández-Martínez et al. (2014) was questioned by Kutsch and Kolari (2015) on the basis of unequal quality of the EC-flux datasets found in FLUXNET and other databases. By excluding complex terrain sites (and young forests) from the Fernández-Martínez et al. (2014) dataset, Kutsch and Kolari (2015) calculated a much reduced variability in CSE, with a «reasonable» mean value of 15% (range 0–30%), and suggesting a much lower influence of nutrient status than claimed by Fernández-Martínez et al. (2014). In their reply, Fernández-Martínez et al. (2015) re-analyzed the same subset of sites selected by Kutsch and Kolari (2015), but using the same generalized linear model as used in their original analysis of the whole dataset, as opposed to the linear model used by Kutsch and Kolari (2015). Fernández-Martínez et al. (2015) then maintained that the findings of the original study were still valid for the restricted dataset, i.e. that the nutrient status had a significant influence on CSE.

The smaller European dataset of our study poses a similar dilemma. The much wider variation in CSE$_{obs}$ than modelled CSE$_{5-yr}$ may both point to possible measurement issues if CSE$_{obs}$ values (especially the larger ones) are considered ecologically implausible, and/or inform on important ecological processes that are not accounted for in the model. Among the forests in our study that seemed particularly inefficient (CSE$_{obs}$ <10%) at retaining photosynthesized carbon (EN4, EN6, EN8, EN11, EN17, EB5), all were classified as L (low ENA) in Fernández-Martínez et al. (2014) and two (EN6, EN11) were even net C sources (R$_{eco}$ > GPP). The EN4, EN6, EN17 sites had the three largest soil organic contents (SOC, Fig. 10a), which may either have induced larger rates of heterotrophic respiration, or may instead indicate low-fertility wet soils where both

assimilation and respiration are suppressed. However, EN4 has also been reported as having unrealistically large ecosystem respiration rates (Anthoni et al., 2004). The EN8 site (mature pine-dominated forest in Belgium) was very unlikely to be N- or S-limited, having been under the high deposition footprint of Antwerp petrochemical harbour and local intensive agriculture for decades, even if emissions have declined over the last 20 years (Neirynck et al., 2007, 2011). However, the comparatively low LAI, GPP and CSE (Fig. 4 in Flechard et al., 2020) at this site are likely not independent of the historical, N- and S-induced soil acidification, which has worsened the already low P and Mg availabilities (Janssens et al., 1999), and from which the forest is only slowly recovering (Neirynck et al., 2002; Holmberg et al., 2018). This site is actually an excellent example to illustrate the complex web of biogeochemical and ecological interactions, which further complicate the quantification of the (single-factor) $N_{dep}$ impact on C fluxes. By not accounting for the low Mg and P availabilities and the poor ecosystem health, the BASFOR model massively over-estimated GPP, $R_{eco}$ and NEP at EN8 (Fig. 6 in Flechard et al., 2020). In fact, based on prior knowledge of this site's acidification history, and since such mechanisms and impacts are not mathematically represented in BASFOR, EN8 was from the start discarded from the calibration dataset for the Bayesian procedure (Cameron et al., 2018). The four lowest $CSE_{obs}$ values were found at sites with topsoil pH < 4 (Fig. 10c), although other forests growing on acidic soils had reasonably large $CSE_{obs}$ ratios.

The large variability in $CSE_{obs}$ cannot be explained by any single edaphic factor (Fig. 10a-c), more likely by a combination of many factors that may include $N_{dep}$ (Fig. 10e). As noted previously, C flux measurements at all four forest sites with $N_{dep}$ > 2.5 g (N) m$^{-2}$ yr$^{-1}$ (EN2, EN8, EN15, EN16) indicated lower productivity estimates than those in the intermediate $N_{dep}$ range, or at least smaller than might have been expected from a linear N fertilisation effect (Fig. 4 in Flechard et al., 2020). EN2 (spruce forest in southern Germany) is also well-documented as an N-saturated spruce forest with large total N losses (~3 g (N) m$^{-2}$ yr$^{-1}$) as NO, $N_2O$ and $NO_3^-$ (Kreutzer et al., 2009), but its productivity and CSE are not affected to the same extent as EN8. Not all the difference is necessarily attributable to the deleterious impacts of excess $N_r$ deposition, as suggested by the GPP normalization exercise (Fig. 8). For example, EN15 and EN16, planted on sandy soils, appear from meta-modelling to suffer from water stress comparatively more than the average of all sites (Fig. 6-Soil), if indicators of soil water retention based on estimates of soil depth, field capacity and wilting point can be considered reliable.

*{Insert Fig. 10 here}*

### 4.3.2 Forest age

Forest age is expected to affect photosynthesis (GPP), growth (NPP), carbon sequestration (NEP) and CSE for many reasons. A traditional view of the effect of stand age on forest NPP (Odum, 1969) postulated that $R_{aut}$ increases with age and eventually nearly balances a stabilized GPP, such that NPP approaches zero upon reaching a dynamic steady state. Revisiting the paradigm, Tang et al. (2014) found that NPP did decrease with age (> 100 yr) in boreal and temperate forests, but the reason was that both GPP and $R_{aut}$ declined, with the reduction in forest growth being primarily driven by GPP, which decreased more rapidly with age than $R_{aut}$ after 100 years. However, the ratio NPP/GPP remained approximately constant within each biome.

The effect of age on NEP and CSE is even more complex since this involves not only changing successional patterns of GPP and $R_{aut}$, but also of $R_{het}$ over a stand rotation of typically one century or more, which is much longer than the longest available flux datasets. Therefore age effects are often studied by comparing differently aged forest sites across the world, which introduces many additional factors of variation, including differences in water availability, soil fertility, or even tree species, genera, or PFTs. Forest and tree ages should in theory be normalized to account for species-specific ontogeny patterns, i.e. the age of 80 years may be relatively young for some species, and quite old for others, and therefore population dynamics may be very different for the same age. Nevertheless, forest age has been suggested to be a dominant factor controlling the spatial and temporal variability in forest NEP at the global scale, compared with abiotic factors such as climate, soil characteristics and nutrient availability (Besnard et al., 2018). In that study, the multivariate statistical model of NEP, using data from 126 forest eddy-covariance flux sites worldwide, postulated a non-linear empirical relationship of NEP

to age, adapted from Amiro et al. (2010), whereby NEP was negative (a net C source) for only a few years after forest establishment, then increased sharply above 0 (a net C sink), stabilized after around 30 years and remained at that level thereafter for mature forests (> 100 years). This model, therefore, did not assume any significant reduction in forest net productivity after maturiy, up to 300 years, consistent with several synthesis studies that have reported significant NEP of centuries-old forest stands (Buchmann and Schulze, 1999; Kolari et al., 2004; Luyssaert et al., 2008).

By analogy, our approach for accounting for the age effect was based on the modelled time course of GPP (Eq. (16)-(17)), which in the BASFOR model tended to stabilize after 100 years, and subsequently using a mean CSE that did not depend on stand age. However, the variability in $CSE_{obs}$ appeared to be much larger in mature forests (>80 years) than in the younger stands (Fig. 10d). For the younger forests (<60 years, all sites probably still in an aggrading phase), the $CSE_{obs}$ values were in a narrow band of 15–30% and were well represented by model simulations, with the exceptions of EN1, EB3 at around 50% and of EN4 being near 0% (all three locations being high elevation sites with complex terrain and potential EC measurement issues, see Flechard et al., 2020). By contrast, values for mature forests were either below 15% or above 30%. For some cold sites such as EN6 and EN11, growing in low nutrient environments (e.g. peat at EN6) with high SOC (Fig. 10a) and/or high soil C/N ratio (Fig. 10b) and low soil pH (Fig. 10c), or for the N-saturated and acidified EN8 site, the low CSE is not necessarily linked to age. Aging, senescence and acidification may at some point curb sequestration efficiency in older forests, but even excluding the complex terrain sites, there remain a good number of productive mature sites with $CSE_{obs}$ in the range 30–40%, which questions the Odum (1969) paradigm of declining net productivity and C equilibirum in old forests.

### 4.3.3 Does nitrogen deposition impact soil respiration?

The overall net effect of $N_r$ deposition on carbon sequestration must include not only productivity gains, but also indirect, positive or negative impacts on soil C losses, which all affect CSE. Carbon sequestration efficiency reflects the combined magnitudes of soil heterotrophic ($R_{het}$) and autotrophic ($R_{aut}$, both below- and above-ground) respiration components, relative to GPP. We postulated that the primary effect of $N_{dep}$ and $N_{supply}$ is on GPP, but potential side effects of $N_{dep}$ or N additions on ecosystem and soil carbon cycling have been postulated. The traditional theory of the role of N on microbial decomposition of SOM was that, above a certain C/N threshold value, the lack of N inhibits microbial activity compared with lower C/N ratios (Alexander, 1977). However, reviews by Fog (1988) and Berg and Matzner (1997) found that microbial activity was often unaffected, or even negatively affected, by the addition of N to low-N decomposing organic material. The negative effects were mostly found for recalcitrant organic matter (high lignin content) with a high C/N ratio (e.g. wood or straw); while N addition to easily degradable organic matter with a low C/N ratio (e.g. leaf litter with low lignin content) actually boosted microbial activity. The meta-analysis by Janssens et al. (2010) of N manipulation experiments in forests suggests that excess $N_r$ deposition reduces soil – especially heterotrophic – respiration in many temperate forests. They argue that the mechanisms include i) a decrease in below-ground C allocation and the resulting root respiration, permitted by a lesser need to develop the rooting system when more N is available (see also Alberti et al., 2015); ii) a reduction in the activity, diversity and biomass of rhizospheric mycorrhizal communities (see also Treseder, 2008); iii) a reduction in the priming effect, the stimulation of SOM decomposition by saprotrophic organisms through root and mycorrhizal release of energy-rich organic compounds; iv) N-induced shifts in saprotrophic microbial communities, leading to reduced saprotrophic respiration; and v) increased chemical stabilization of SOM into more recalcitrant compounds. The authors point out that in N-saturated forests different processes and adverse effects are at play (e.g. base cation leaching and soil acidification). Of the five afore-mentioned mechanisms potentially involved in the suppression of soil respiration by N addition, only the first one (control by N availability of the root/shoot allocation ratio) is functional in BASFOR, and therefore our simulations do not include the other inhibitory effects of excess N on mycorrhizal, fungal and bacterial respiration.

An important implication of the negative impact of $N_r$ on soil respiration is that the nitrogen fertilisation effect on gross photosynthesis would be roughly doubled, in terms of C sequestration, by the concomitant decrease in soil respiration. In their meta-analyses of N addition experiments in forests and comparison of sites exposed to low *vs* elevated $N_{dep}$, Janssens et

al. (2010) show that both $R_{het}$ and soil carbon efflux (SCE), a proxy for total $R_{soil}$ (= $R_{het}$ + $R_{aut,soil}$), tend to decline with N addition, be it through fertilisation or atmospheric deposition, although the effect is far from universal. The negative $N_{dep}$ response of $R_{het}$ was much more pronounced for SOM than for leaf litter, and stronger at highly productive sites than at less productive sites. The negative impact on SCE was mostly found at sites where N was not limiting for photosynthesis. When N is strongly limiting, and in young forests, $N_r$ deposition may well favour SOM decomposition.

To examine the potential impact of $N_{dep}$ on $R_{soil}$, we compiled the soil respiration data available from the literature and databases for the collection of forest sites in our study, which covers the whole N limitation to N saturation spectrum. Sites ranged from highly N-limited boreal systems, where an N addition might trigger enhanced tree growth, increased microbial biomass and heterotrophic respiration, to N-saturated, acidified systems (EN2, EN8, possibly also EN15, EN16), in which poor ecosystem and soil health may lead to different ecological responses than those of the below-ground carbon cycling scheme in Janssens et al. (2010).

Since the below-ground autotrophic (root and rhizosphere) respiration component is regulated to a large extent by photosynthetic activity (Collalti and Prentice, 2019), as well as seasonality in below-ground C allocation (Högberg et al., 2010), and contributes a large part of $R_{soil}$ on an annual basis (Korhonen et al., 2009), the relationship of $R_{soil}$ to $N_{dep}$ is examined by first normalizing to GPP (Fig. 11a), yielding a soil respiration metric that is comparable between sites (for $R_{soil}$ data, see Table S7 in the Supplement to Flechard et al., 2020). Similarly, the ratio $R_{soil}/R_{eco}$ shows the relative contribution of below-ground to total (ecosystem) respiration (Fig. 11c). Note that caution is needed when considering both $R_{soil}/GPP$ and $R_{soil}/R_{eco}$ ratios, since significant uncertainty may arise from i) methodological flaws in comparing chamber *versus* eddy covariance measurements (e.g. considerations over tower footprint, spatial heterogeneity and representativeness of soil collars), ii) uncertainty in deriving GPP and $R_{eco}$ estimates from EC-NEE measurements, and iii) different time spans for the EC and soil chamber measurements, affected by inter-annual flux variability. Thus, values of $R_{soil}/R_{eco}$ above unity (Fig. 11c), although physically non-sensical, do not necessarily imply large measurement errors, but possibly also that there may be no spatial or temporal coherence in EC and chamber data (Luyssaert et al., 2009).

Either ignoring such outliers, or judging that a measurement bias by soil chambers affects all sites the same way (e.g. systematic over-estimation of soil respiration in low turbulence conditions when using static chambers, Brændholt et al., 2017), we may argue that the apparent decrease of both chamber/EC ratios $R_{soil}/GPP$ and $R_{soil}/R_{eco}$ with $N_{dep}$ (Fig. 11a, 11c) has some reality, even if their absolute values are biased. Soil $CO_2$ efflux tends to be a larger fraction of GPP (>0.5) at the smaller $N_{dep}$ rates (<1.5 g (N) $m^{-2}$ $yr^{-1}$) than at sites with larger $N_{dep}$, where this fraction is more often in the range 0.4–0.5. It is also noteworthy that the largest $R_{soil}/GPP$ ratios (EN5, EN17) are found at sites with very large SOC compared with the other sites (Fig. 11b). The $R_{het}/R_{soil}$ ratio also tends to decrease with $N_{dep}$ (Fig. 11e), and although measured by different methods at the different sites, this is arguably a more robust metric than chamber/EC respiration ratios, because the differential respiration measurements on control and treatment plots (root exclusion, trenching, girdling) are made on the same spatial and temporal scales.

Many other factors that impact soil respiration (age, soil pH, microbial abundance and diversity, etc.) are not considered here and beyond the scope of this paper. In view of these uncertainties, if the assessment within this restricted dataset does not provide a full and incontrovertible proof of the negative impact of $N_r$ deposition on soil respiration, it at least is not in open contradiction to the prevailing paradigm that both below-ground autotrophic and heterotrophic respiration are expected to decrease as $N_r$ deposition increases. However, the decreasing trends observed in Fig. 11a, 11c, 11e are largely driven by these few high $N_{dep}$ sites (>3g (N) $m^{-2}$ $yr^{-1}$) in which the negative effects of N saturation and acidification very likely outweigh the benefits of reduced soil respiration in terms of C sequestration.

*{Insert Fig. 11 here}*

## 5 Conclusion

The magnitude of the mean $N_r$ deposition-induced fertilisation effect on forest C sequestration, derived here from eddy covariance flux data from a diverse range of European forest sites, is of the order of 40–50 g (C) g$^{-1}$ (N), and comparable with current estimates obtained from inventory data and deposition rates from continental-scale deposition modelling used in the most recent studies and reviews. The range of dC/dN values is a consequence of where in the ecosystem the $N_r$-induced carbon sequestration takes place, whether there are $N_r$ losses and how other environmental conditions affect growth. However, this mean dC/dN response should be taken with caution for several reasons. First, uncertainties in our dC/dN estimates are large, partly because of the relatively small number of sites (31) and their large diversity in terms of age, species, climate, soils, and possibly fertility and nutrient availability. Second, adopting a mean overall dC/dN response universally and regardless of the context may be misleading due to the clear non-linearity in the relationship between forest productivity and the level of $N_r$ deposition, i.e. the magnitude of the response changes with the N status of the ecosystem. Beyond a $N_r$ deposition threshold of 1–2 g (N) m$^{-2}$ yr$^{-1}$ the productivity gain per unit $N_r$ deposited from the atmosphere starts to decrease significantly. Above 2.5 g (N) m$^{-2}$ yr$^{-1}$, productivity actually decreases with further $N_r$ deposition additions, and this is accompanied by increasingly large ecosystem $N_r$ losses, especially as $NO_3^-$ leaching. Further sources of uncertainty in our forest ecosystem model involve missing – but possibly large – terms of the N cycle, such as $N_2$ fixation, $N_2$ loss by denitrification, DON uptake by trees and DON leaching.

Ecosystem meta-modelling was required to factor out the effects of climate, soil water retention and age on forest productivity, a necessary step before estimating a generalised response of C storage to $N_r$ deposition. Neglecting these effects would lead to a large over-estimation (factor of 2) of the dC/dN effect in this European dataset and possibly also in other datasets worldwide. After factoring out the effects of climate, soil water retention and forest age in the present dataset, only part of the non-linearity was removed and there was still a decline in the dC/dN response with increasing $N_{dep}$. One possible interpretation is that the remaining non-linearity may be regarded as an indicator of the impact of increasing severity of N saturation on ecosystem functioning and forest growth. However, the results also show that the large inter-site variability in carbon sequestration efficiency, here defined at the ecosystem scale and observed in flux data, cannot be entirely explained by the processes represented in model we used. This is likely due in part to an incomplete understanding and over-simplified model representation of plant carbon relations, soil heterotrophic and autotrophic respiration, the response to nitrogen deposition of physiological processes such as stomatal conductance and water-use efficiency, and possibly also because other nutrient limitations were insufficiently documented at the monitoring sites and not accounted for in the model.

**Code and data availability**

The data used in this study are publicly available from online databases and from the literature as described in the Materials and Methods section.

The codes of models and other software used in this study are publicly available online as described in the Materials and Methods section.

**Author Contributions**

CRF, MvO, DRC, WdV, MAS, AI conceived the paper; CRF performed the data analyses, ran model simulations and wrote the text; MvO, DRC wrote and provided the BASFOR model code and performed Bayesian calibration; MAS, EN, UMS, KBB, WdV conceived or designed the NEU study; AI, NB, IAJ, JN, LM, AV, DL, ALeg, KZ, MAub, MAur, BHC, JD, WE, RJ, WLK, ALoh, BL, GM, VM, JO, MJS, TV, CV, KBB, UMS provided eddy covariance and/or other field data, or contributed to data collection from external databases and literature; MvO, DRC, WdV, AI, MAS, NB, NBD, IAJ, JN, LM, AV, DL, ALeg, KZ, AJF, RJ, AN, EN, UMS contributed substantially to discussions and revisions.

**Competing Interests**

The authors declare that they have no conflict of interest

**Acknowledgements**

The authors gratefully acknowledge financial support by the European Commission through the two FP-6 integrated projects CarboEurope-IP (Project No. GOCE-CT-2003-505572) and NitroEurope-IP (Project No. 017841), the FP-7 ECLAIRE project (Grant Agreement No. 282910), and the ABBA COST Action ES0804. We are also thankful for funding from the French GIP-ECOFOR consortium under the F-ORE-T forest observation and experimentation network, and from the MDM-2017-0714 Spanish grant. We are grateful to Janne Korhonen, Mari Pihlatie and Dave Simpson for their comments on the manuscript. Finalisation of the manuscript was supported by the UK Natural Environment Research Council award number NE/R016429/1 as part of the UK-SCAPE programme delivering National Capability. We also wish to thank two anonymous referees for their constructive criticism of the manuscript.

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

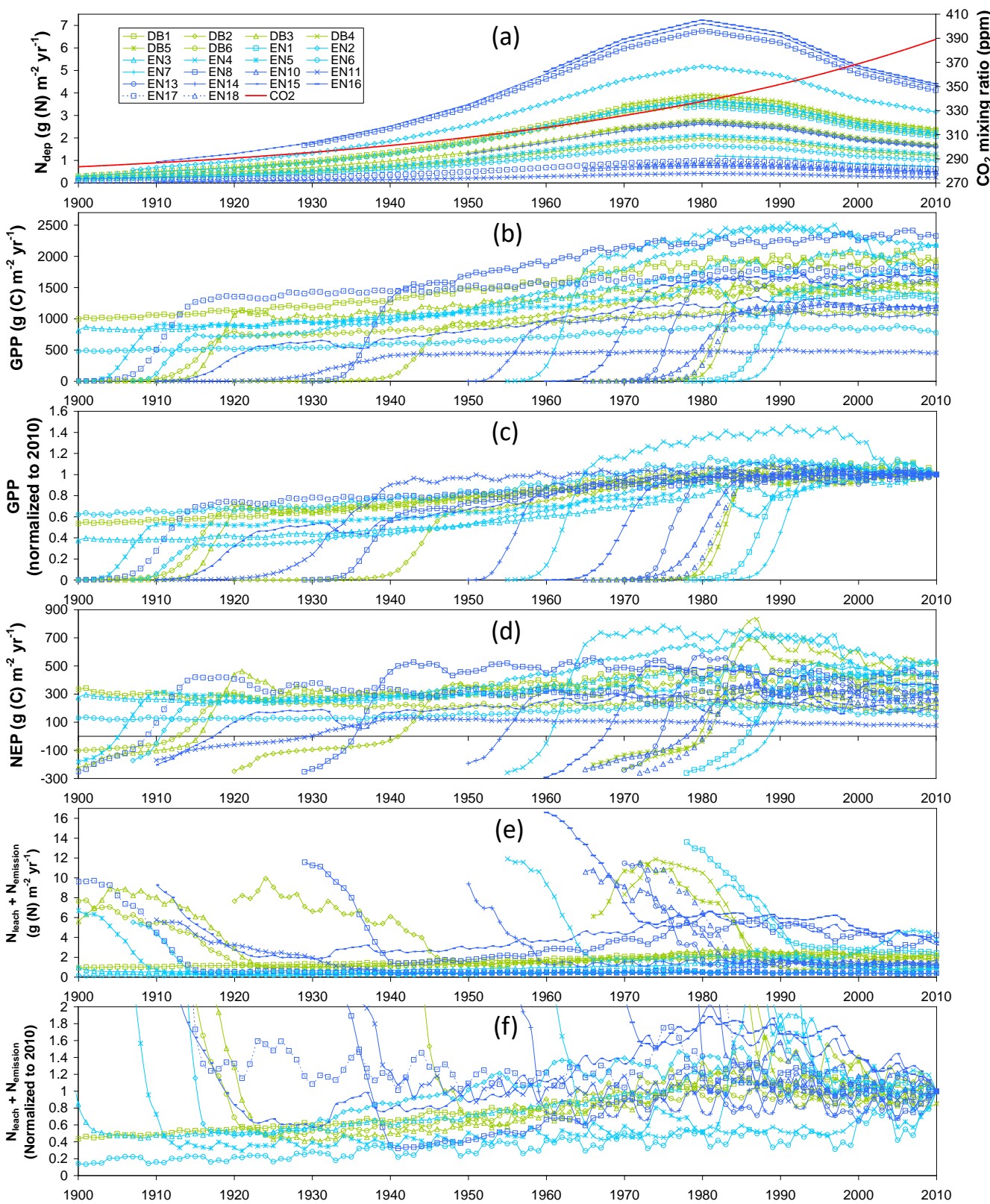

Figure 1. Time courses for 22 forest study sites (DB: deciduous broadleaf; EN: evergreen needleleaf) of (a) assumed atmospheric $N_r$ deposition ($N_{dep}$) and $CO_2$ mixing ratio, and baseline model simulations of (b) gross primary productivity (GPP), (c) GPP normalized to the 2010 value, (d) net ecosystem productivity (NEP), (e) total N losses by leaching ($N_{leach}$) and gaseous emissions ($N_{emission}$), and (f) total N losses normalized to 2010.

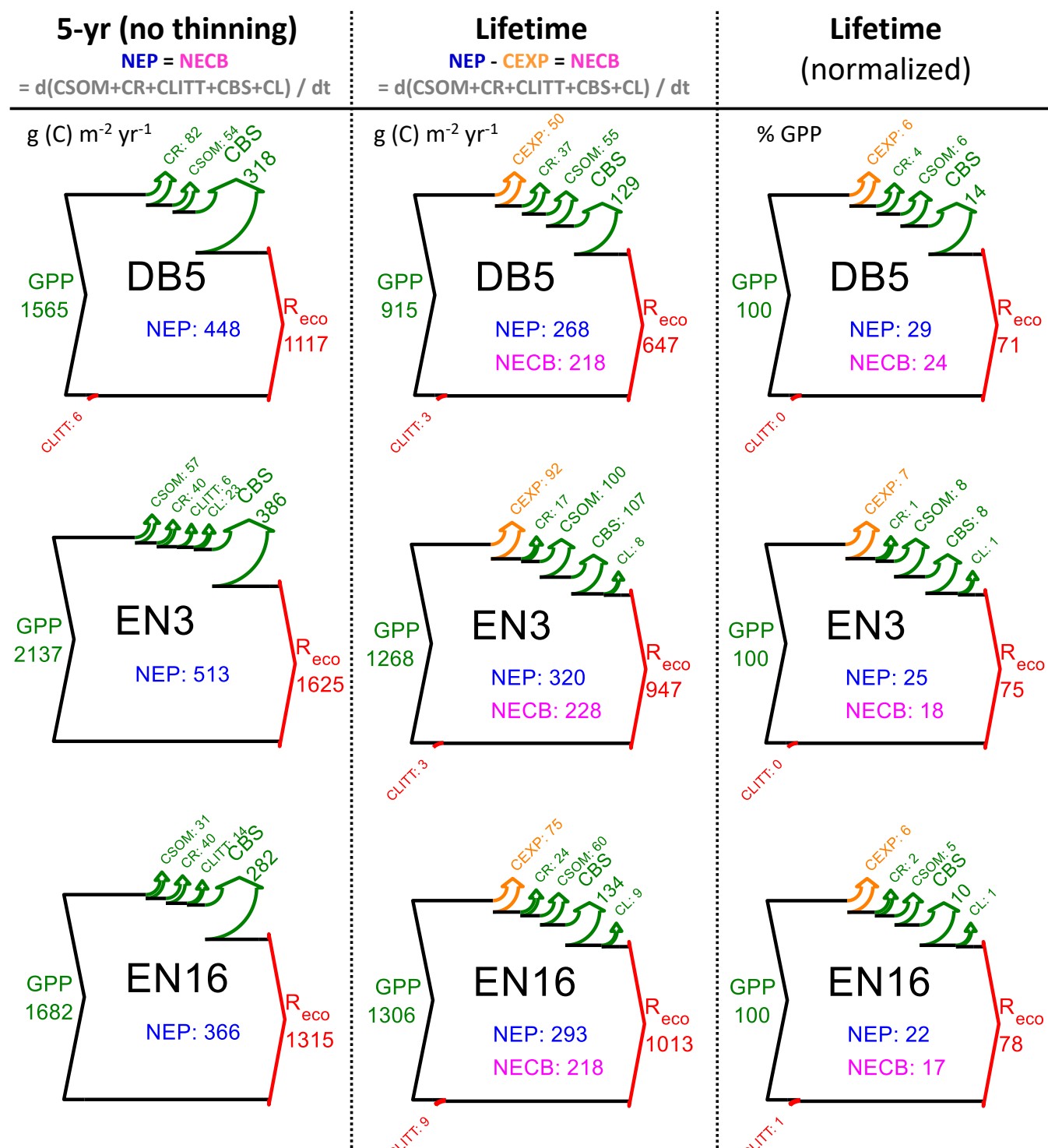

Figure 2. Modelled (BASFOR) budgets and partitioning of gross primary productivity (GPP), ecosystem respiration ($R_{eco}$), net ecosystem productivity (NEP), net ecosystem carbon balance (NECB), at three example forest sites (DB5: 45-yr old *Fagus sylvatica*; EN3: 120-yr old *Picea abies*; EN16: 51-yr old *Pseudotsuga menziesii*), and associated modelled changes in C pools in soil organic matter (CSOM), roots (CR), litter layers (CLITT), branches and stems (CBS) and leaves (CL) (units: g (C) m⁻² yr⁻¹ left and center; normalized to % lifetime GPP on the right). Simulations were run either over the most recent 5-year period which did not include any thinning event («5-yr» in the text), or over the whole time period since the forest was established («lifetime»). Green indicates ecosystem C gain (photosynthesis and C pool increase); red denotes ecosystem C loss (respiration and C pool decrease); the orange arrows indicate C export through thinning (CEXP). The NECB percentage value (right) corresponds to the lifetime carbon sequestration efficiency (CSE). The sizes of the Sankey plots are not proportional to the C fluxes of the different study sites.

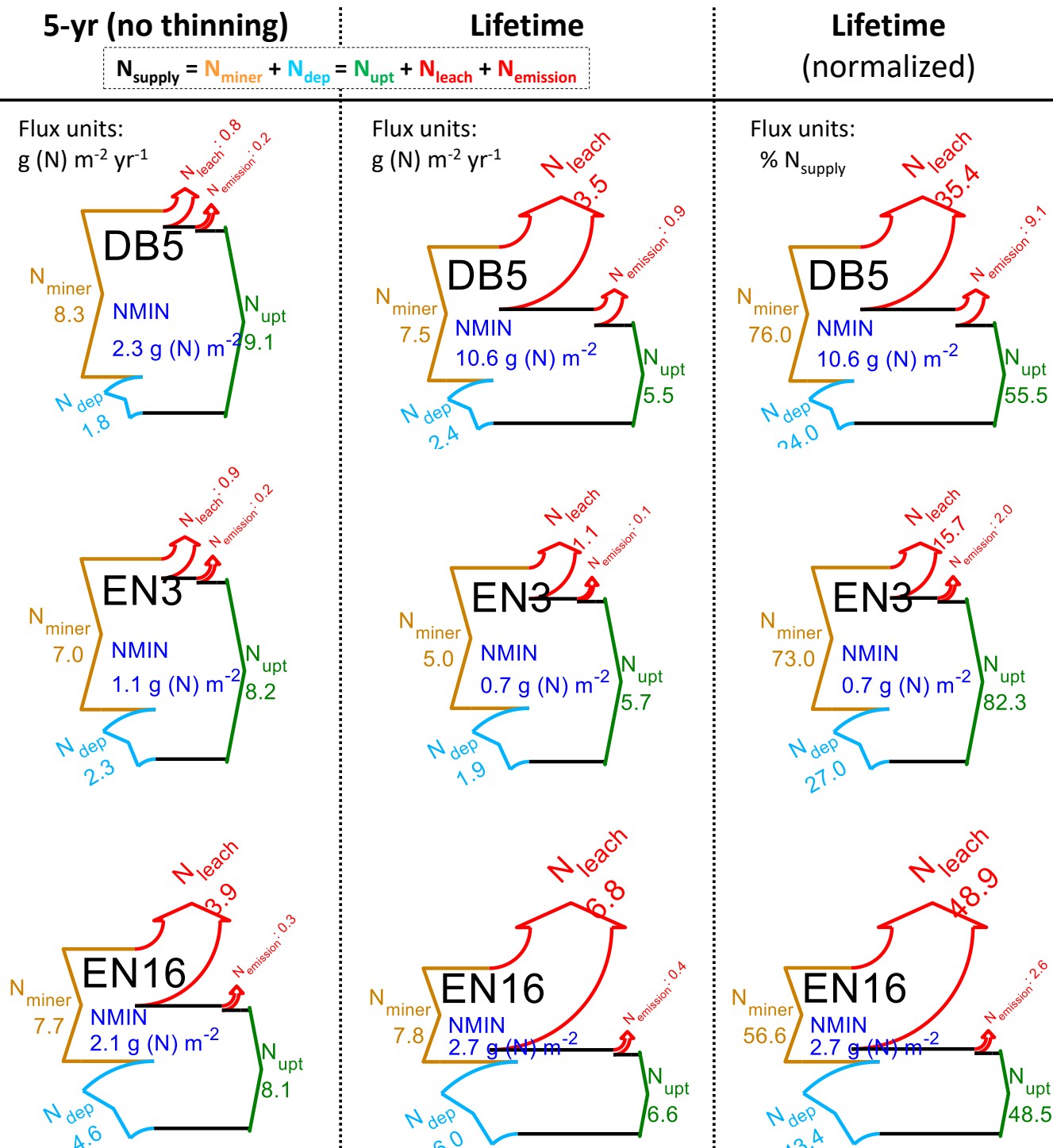

**Figure 3. Modelled (BASFOR) inorganic nitrogen budgets at three example forest sites (DB5: 45-yr old *Fagus sylvatica*; EN3: 120-yr old *Picea abies*; EN16: 51-yr old *Pseudotsuga menziesii*). Simulations were run either over the most recent 5-year period which did not include any thinning event («5-yr» in the text), or over the whole time period since the forest was established («lifetime»). The data show ecosystem SOM mineralisation ($N_{miner}$) and atmospheric $N_r$ deposition ($N_{dep}$), balanced by vegetation uptake ($N_{upt}$) and the sum of losses as dissolved N ($N_{leach}$) and gaseous NO + $N_2O$ ($N_{emission}$) (units: g (N) m$^{-2}$ yr$^{-1}$ left and center; % of lifetime $N_{supply}$ on the right, with $N_{supply}$ defined as $N_{miner} + N_{dep}$). NMIN indicates the mean size of the soil inorganic N pool (g (N) m$^{-2}$) over the modelling period. The N uptake percentage value (right) corresponds to the lifetime nitrogen uptake efficiency (NUPE). The sizes of the Sankey plots are not proportional to the N fluxes of the different study sites.**

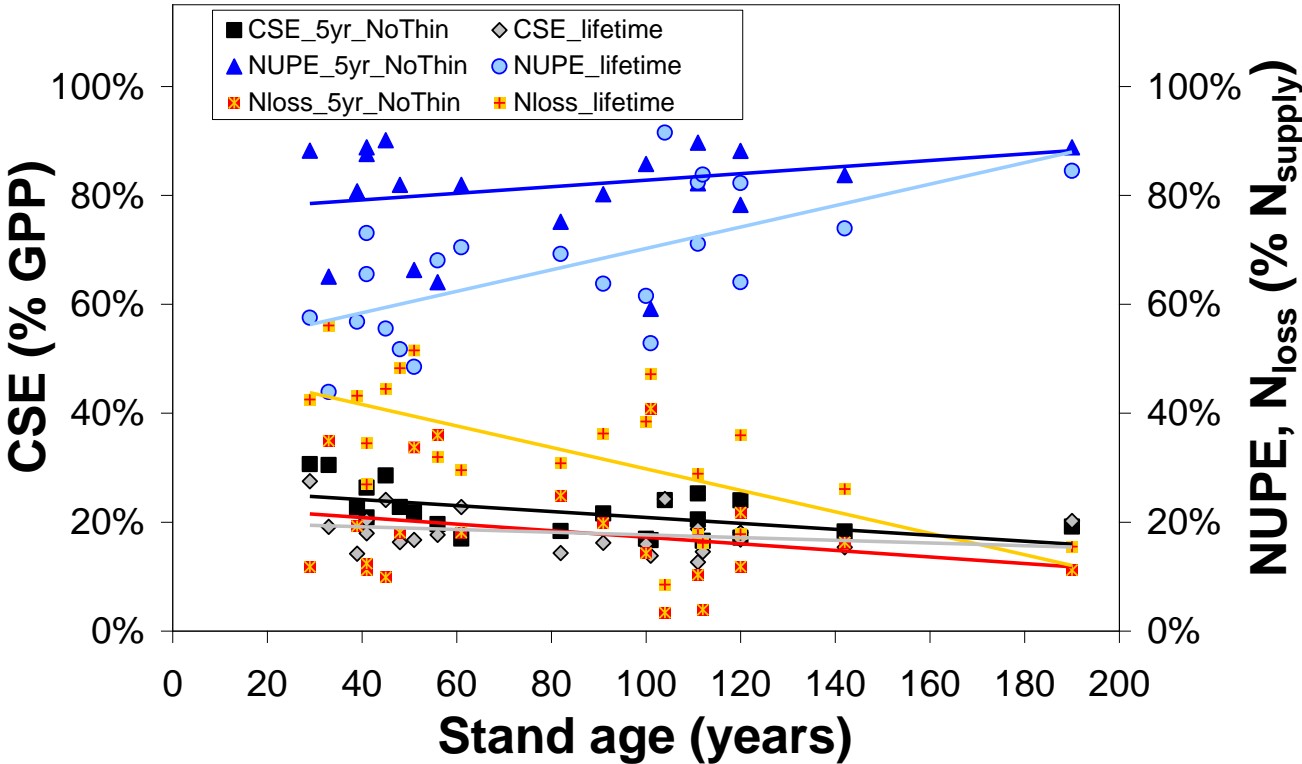

**Figure 4. Influence of forest stand age on modelled (BASFOR) C sequestration efficiency (CSE, expressed as % gross primary productivity GPP), N uptake efficiency (NUPE) and the $N_{loss}$ fraction (expressed as % $N_{supply}$). Each data point represents one of 22 modelled forest sites. CSE and NUPE values are calculated either i) over the most recent 5-yr period including no thinning event around the time frame of the CEIP-NEU integrated projects, or ii) over the whole lifetime of the stands (including all thinning events). See Eq. (1)-(9) for definitions and calculations of the indicators.**

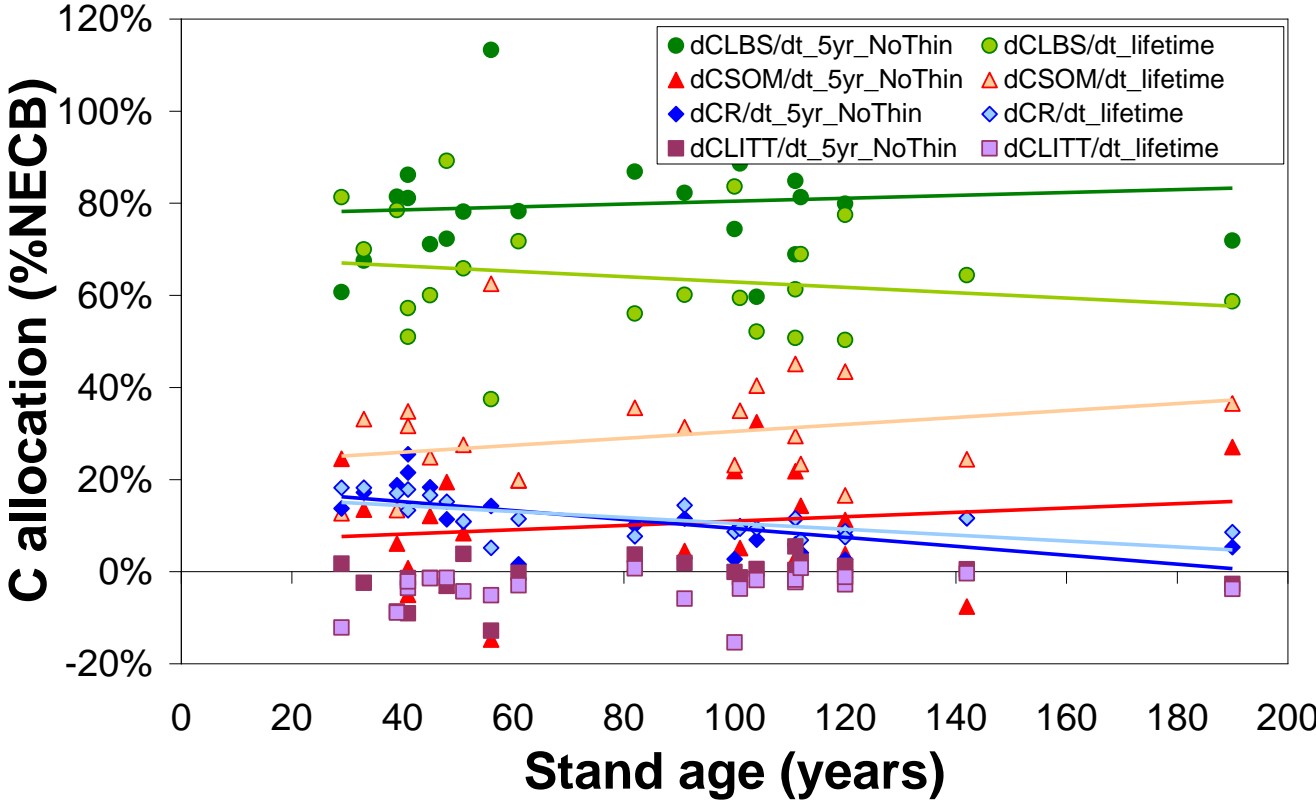

**Figure 5. Modelled (BASFOR) ultimate allocation of sequestered C (expressed as % net ecosystem carbon balance NECB) into ecosystem pools in soil organic matter (CSOM), roots (CR), litter layers (CLITT), leaves, branches and stems (CLBS). Each data point represents one of 22 modelled forest sites, plotted as a function of stand age. At each site, the net ecosystem carbon balance equals the sum of all individual storage (or loss) terms, i.e. NECB = dCLBS/dt + dCSOM/dt + dCR/dt + dCLITT/dt, shown here as fractions of the total to indicate the relative importance of the different ecosystem sinks. Values are calculated either i) over the most recent 5-year period including no thinning event around the time frame of the CEIP-NEU integrated projects, or ii) over the whole lifetime of the stands (including all thinning events).**

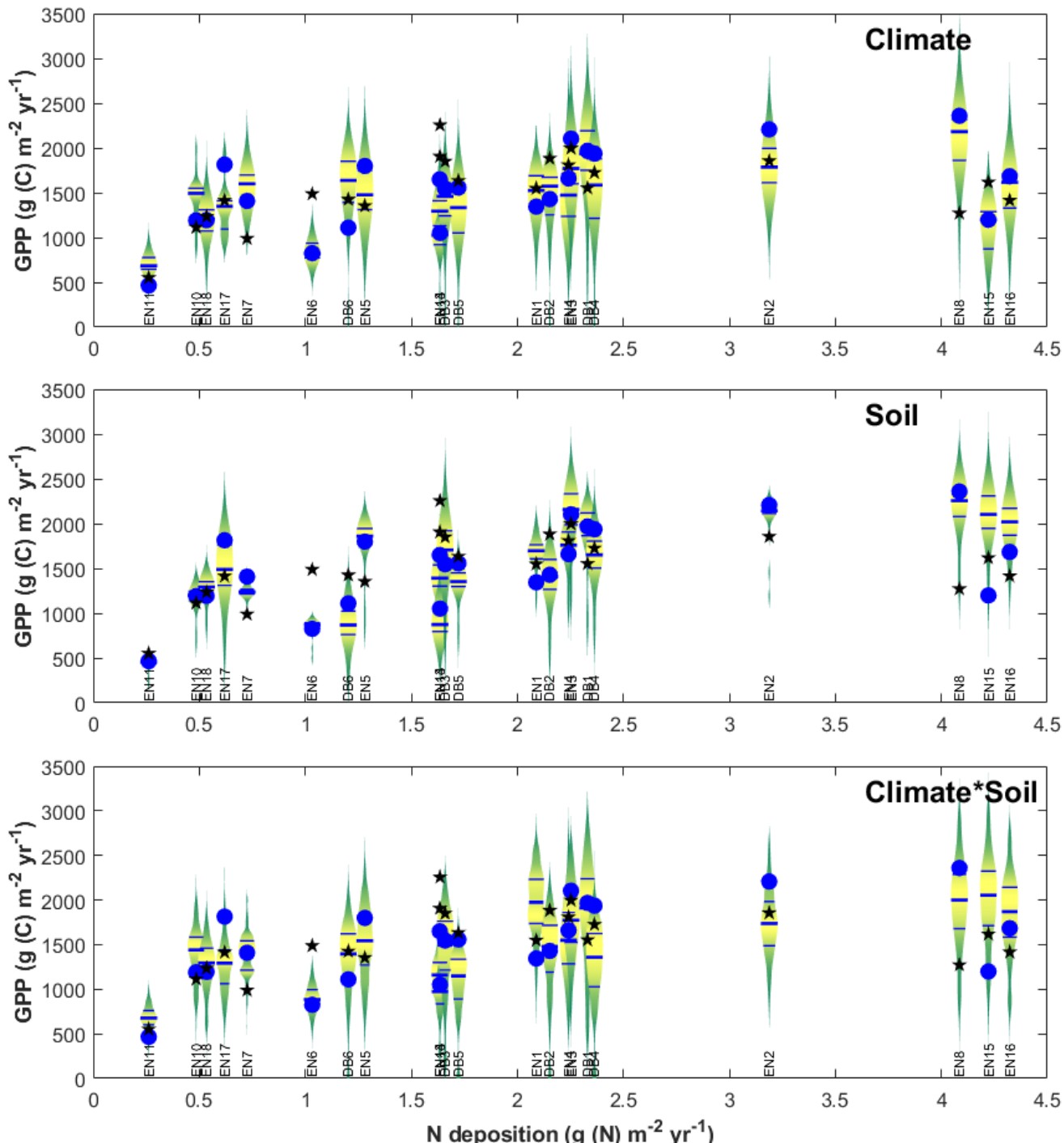

**Figure 6. Input sensitivity study for gross primary productivity (GPP) modelled at each forest monitoring site for different soil/climate scenarios (vertical «violin» plots), compared with model base runs GPP_base (blue circles) and EC-derived GPP_obs (black stars). The data are displayed as a function of $N_r$ deposition over the CEIP-NEU measurement periods, for n=22 deciduous broadleaf (DB) and coniferous evergreen needleleaf (EN) forest ecosystems. For each site, the violin plot shows the range and distribution (median, quartiles) of GPP modelled at the site using climate and/or soil input data from all 22 sites, showing the sensitivity to model inputs other than N deposition. See text for details.**

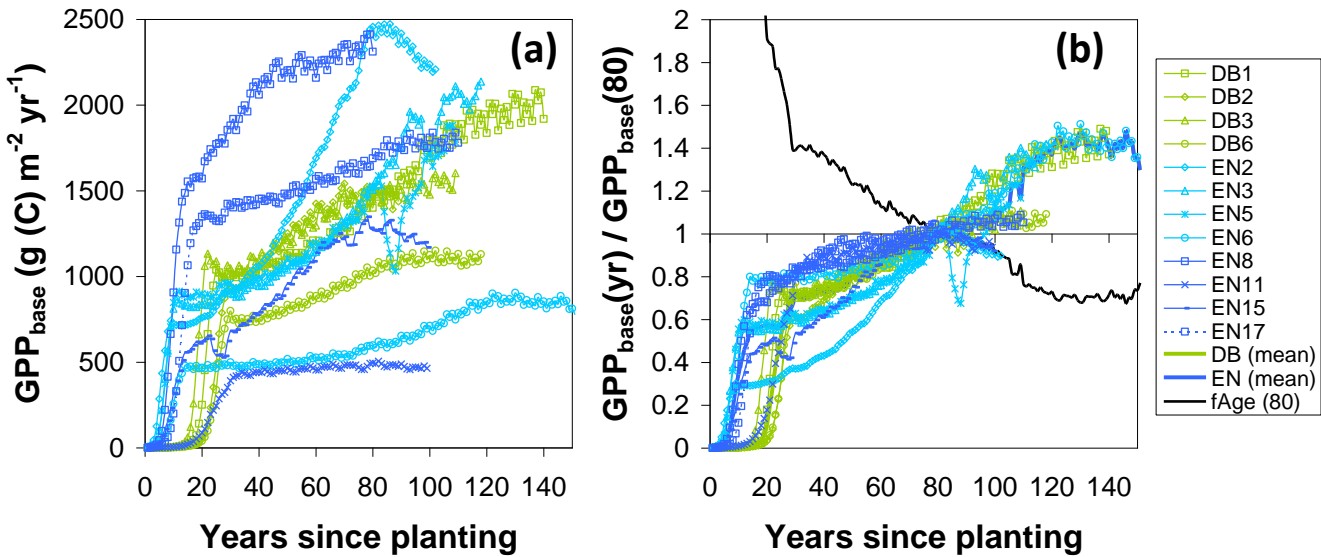

**Figure 7. Steps in the calculation of a normalization factor for forest age (f$_{AGE}$, normalized to 80 yr) from modelled BASFOR growth curves for mature forests (12 sites older than 80 yr). (a) Modelled time course for baseline gross primary productivity (GPP$_{base}$); (b) Each site's GPP$_{base}$ curve is normalized to the value at age 80 yr. A single f$_{AGE}$ curve is then calculated as the mean of all sites after normalization to GPP$_{base}$(80). The f$_{AGE}$ curve is subsequently used as a scaling function to standardize all sites' measured GPP to a notional age of 80 (see Eq. (10), (16), (17)). DB: deciduous broadleaf; EN: coniferous evergreen needleleaf.**

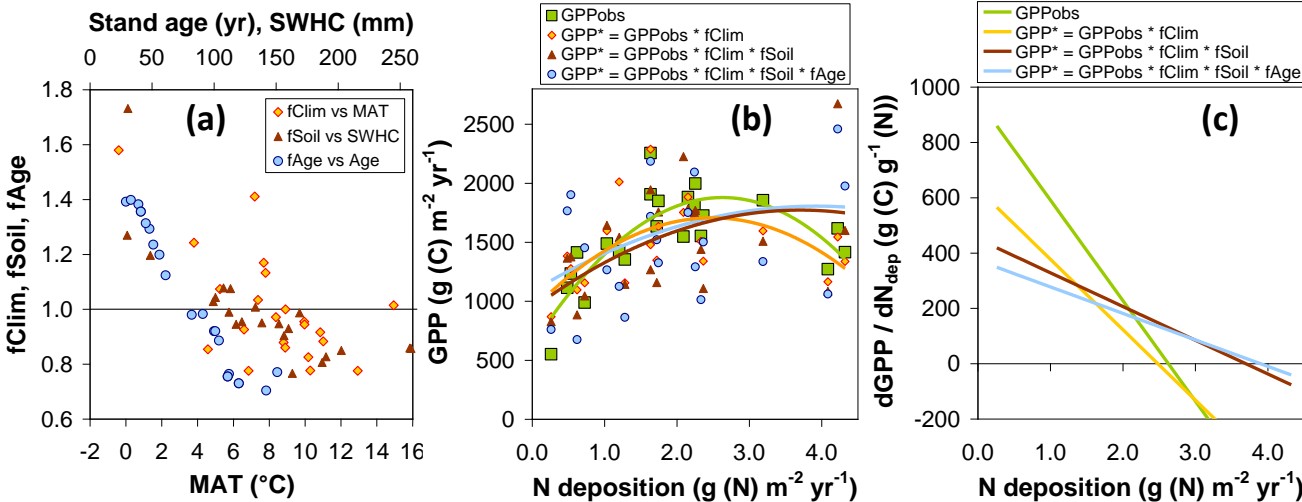

**Figure 8. Model-based assessment of the sensitivity of gross primary productivity (GPP) to climate, soil, age and N_r deposition. (a) GPP standardization factors for climate (f_CLIM), soil (f_SOIL) and age (f_AGE) for observational (EC-based) data as a function of the dominant climatic and soil drivers (MAT: mean annual temperature; SWHC: soil water holding capacity; see text for details); (b) the resulting standardized GPP\* compared with the original GPP_obs as a function of N_dep (one data point for each of 22 sites), with 2^nd-order polynomial fits; (c) estimates of the GPP response to N_dep, calculated as the slope of the tangent line to the quadratic fits and plotted as a function of N_dep.**

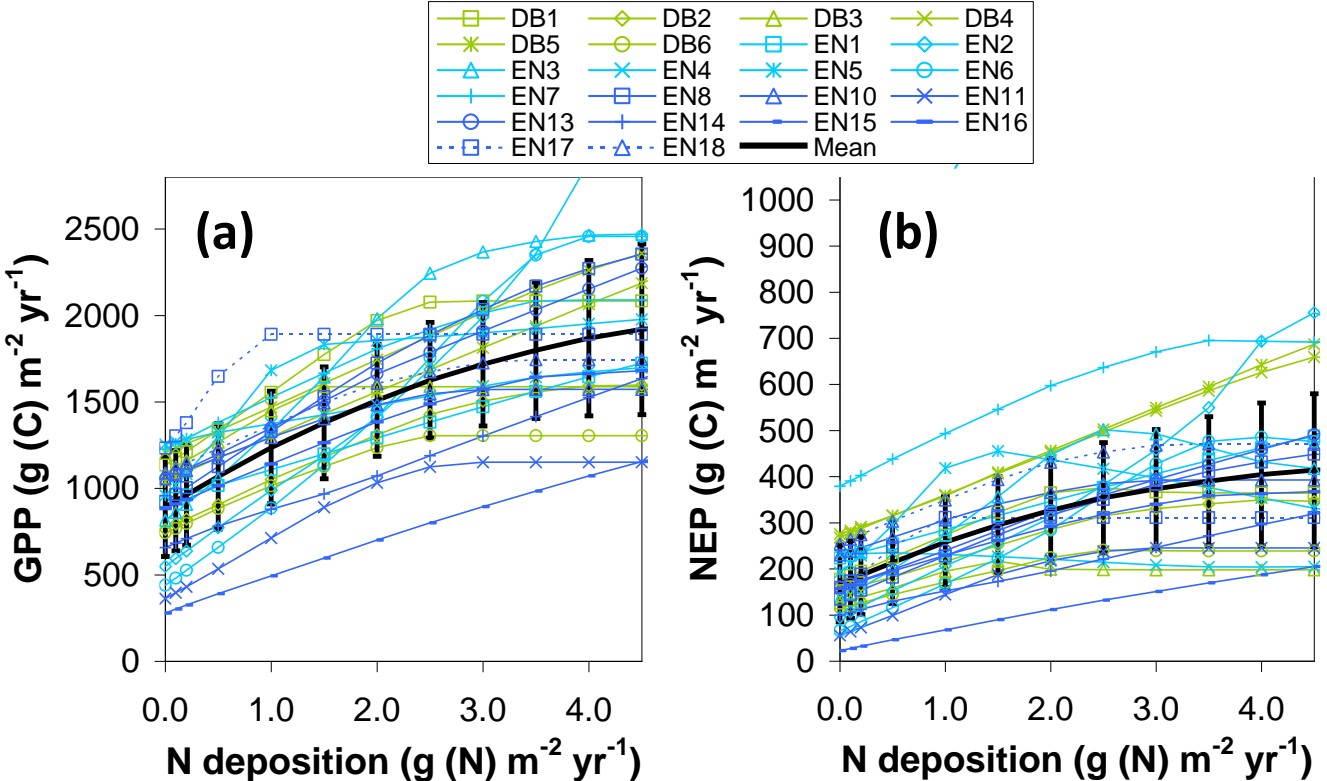

**Figure 9.** Simulated BASFOR model sensitivity to N deposition of (a) gross primary productivity (GPP) and (b) net ecosystem productivity (NEP) for 22 forest sites (with mean +/- standard deviation), derived from a purely modelled approach (not involving measured EC flux data). Each site was modelled using a range of $N_{dep}$ values from 0 to 4.5 g (N) m$^{-2}$ yr$^{-1}$ (constant $N_{dep}$ over the lifetime of the stands). DB: deciduous broadleaf; EN: coniferous evergreen needleleaf.

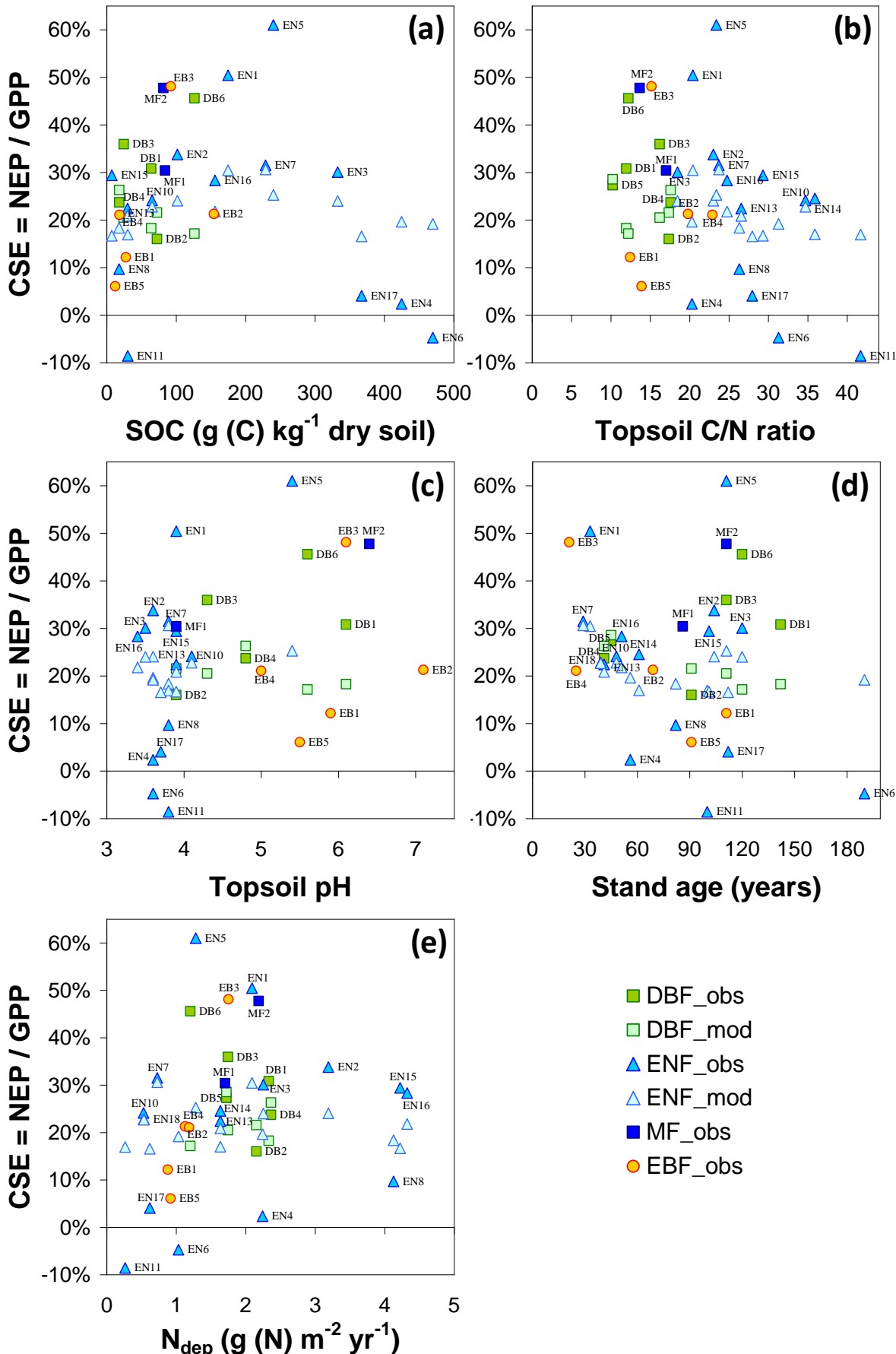

**Figure 10. Variability of observation-based (obs) and modelled (mod) carbon sequestration efficiency (CSE) defined as the ratio of net ecosystem productivity (NEP) to gross primary productivity (GPP), calculated over a ~5-yr measurement period. The data are plotted versus (a) topsoil organic carbon content (SOC), (b) topsoil C/N ratio, (c) topsoil pH, (d) forest stand age, and (e) nitrogen deposition ($N_{dep}$). DBF: deciduous broadleaf forests; ENF: coniferous evergreen needleleaf forests; MF: mixed needleleaf/broadleaf forests; EBF: Mediterranean evergreen broadleaf forests.**

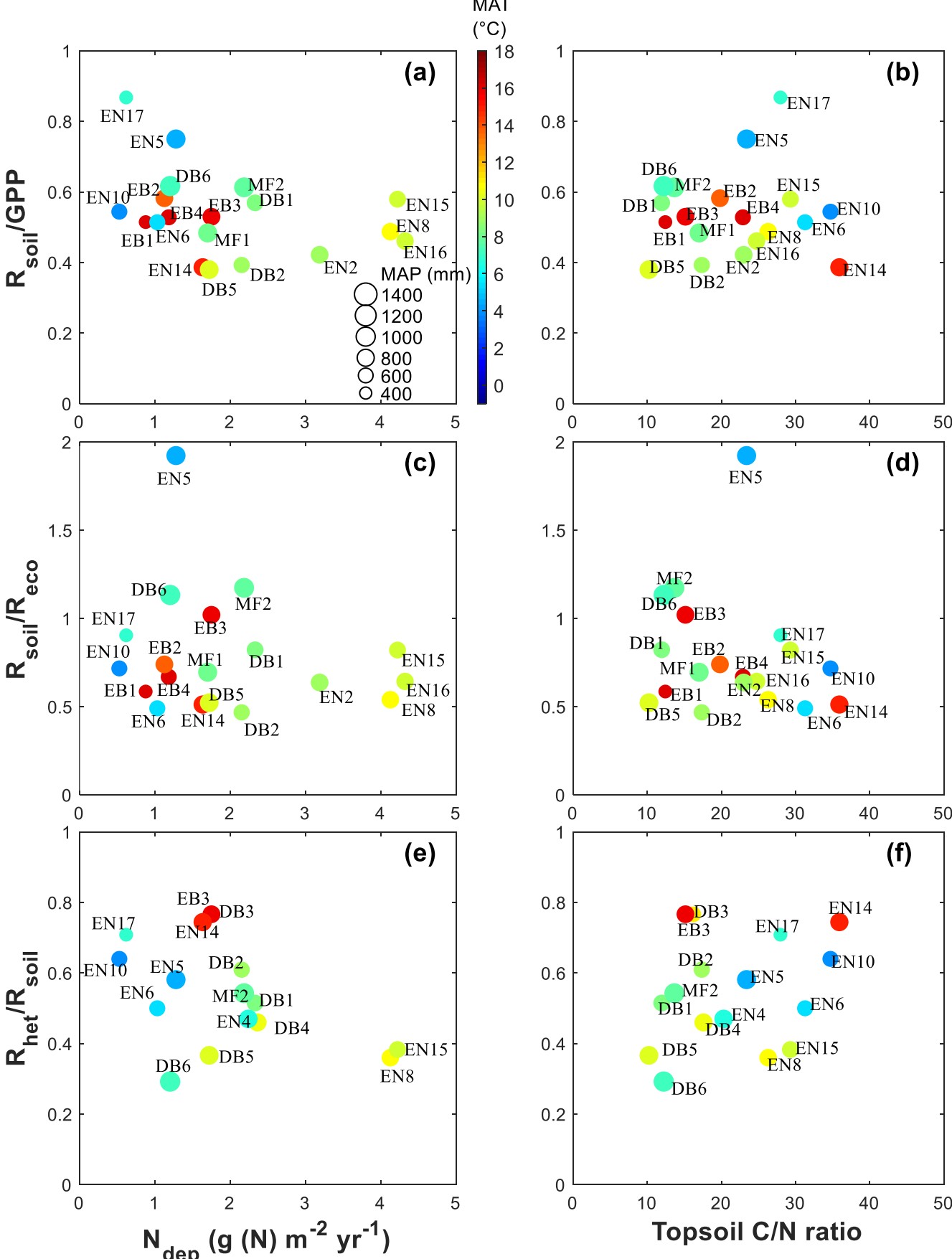

**Figure 11. Variability of normalized soil respiration metrics as a function of nitrogen deposition (a, c, e) and soil organic carbon (b, d, f). In all plots, the color scale indicates mean annual temperature (MAT), and the symbol size is proportional to mean annual precipitation (MAP). $R_{soil}$: total soil respiration; $R_{eco}$: total ecosystem respiration; $R_{het}$: heterotrophic component of $R_{soil}$; GPP: gross primary productivity. DB: deciduous broadleaf; EN: coniferous evergreen needleleaf; EB: Mediterranean evergreen broadleaf; MF: mixed needleleaf/broadleaf forests.**

Table 1. BASFOR model state variables, inputs and outputs, and other acronyms used in the study.

| BASFOR variables | Description |
| --- | --- |
| *Tree state variables* | |
| CL | Carbon pool in Leaves |
| CB | Carbon pool in Branches |
| CS | Carbon pool in Stems |
| CLBS | Carbon pool in Leaves, Branches and Stems |
| CR | Carbon pool in Roots |
| CRES | Carbon pool in Reserves |
| NL | Nitrogen pool in Leaves |
| SD | Forest stand density |
| *Soil state variables* | |
| CLITT | Carbon pool in Litter layers |
| CSOMF | Carbon pool in Soil Organic Matter (Fast turnover) |
| CSOMS | Carbon pool in Soil Organic Matter (Slow turnover) |
| NLITT | Nitrogen pool in Litter layers |
| NSOMF | Nitrogen pool in Soil Organic Matter (Fast turnover) |
| NSOMS | Nitrogen pool in Soil Organic Matter (Slow turnover) |
| NMIN | Soil Mineral (inorganic) Nitrogen pool |
| WA | Water pool in the root zone |
| *Soil parameters* | |
| $\Phi_{SAT}$ | Saturation soil water content |
| $\Phi_{FC}$ | Field capacity |
| $\Phi_{WP}$ | Wilting point |
| ROOTD | Root depth |
| *Model inputs (daily time step)* | |
| $R_g$ | Daily global radiation |
| $T_a$ | Daily average air temperature |
| P | Daily accumulated rain |
| WS | Daily average wind velocity |
| RH | Water vapour pressure |
| $CO_2$ | Annual mean $CO_2$ mixing ratio |
| $N_{dep}$ | Annual atmospheric nitrogen deposition |
| thinFR | Fraction of trees removed by thinning |
| *Model outputs* | |
| H | Tree height |
| DBH | Diameter at breast height |
| LAI | Leaf area index |
| LeafN | Leaf N content |
| GPP | Gross primary productivity |
| $R_{eco}$ | Ecosystem respiration |
| $R_{het}$ | Soil heterotrophic respiration |
| NPP | Net primary productivity |
| NEE | Net ecosystem exchange |
| ET | Evapotranspiration |
| $N_{miner}$ | Nitrogen supply from SOM mineralization |
| $N_{upt}$ | Root N uptake by trees |
| $N_{leach}$ | Inorganic N leaching |
| NO | Nitric oxide |
| $N_2O$ | Nitrous oxide |
| $N_{emission}$ | Gaseous soil NO + $N_2O$ emissions |

| Other variables | |
| --- | --- |
| $GPP_{obs}$, $NEP_{obs}$ | Observation-based (eddy covariance) GPP or NEP |
| $GPP_{base}$ | Baseline model run for GPP |
| GPP*, NEP* | Model-standardized observation-based GPP or NEP |
| $f_{CLIM}$, $f_{SOIL}$, $f_{AGE}$ | Model-derived standardization factors to account for climate, soil, age |
| NECB | Modelled net ecosystem carbon balance, calculated as d(CLBS+CR+CSOM+CLITT)/dt |
| $R_{aut}$ | Autotrophic respiration |
| $R_{soil}$ | Soil (heterotrophic and rhizospheric) respiration |
| SCE | Soil $CO_2$ efflux measured by chamber methods |
| $CSE_{obs}$ | Observation-based carbon sequestration efficiency ($NEP_{obs}/GPP_{obs}$) |
| $CSE_{5-yr, \ lifetime}$ | Modelled carbon sequestration efficiency; = NEP/GPP (5-yr), or NECB/GPP (lifetime) |
| NUPE | Modelled nitrogen uptake efficiency, calculated as $N_{upt}$ / $N_{supply}$ |
| $N_{supply}$ | Total mineral N supply, calculated as (modelled) $N_{miner}$ + (observation-based) $N_{dep}$ |
| $N_{loss}$ | Modelled percentage ecosystem N losses, calculated as ($N_{leach}$ + $N_{emission}$) / $N_{supply}$ |
| dC/dN, $dGPP/dN_{dep}$, $dNEP/dN_{dep}$ | Response (slope) of ecosystem C productivity versus atmospheric $N_r$ deposition |
| SWHC | Soil water holding capacity, = ($\Phi_{FC}$ - $\Phi_{WP}$) x ROOTD |
| MAT, MAP | Mean annual temperature or precipitation |
| CEXP | Carbon exported by thinning or harvest in forests |

**Table 2. Estimates of ecosystem dC/dN response for gross and net productivity, calculated under different assumptions and expressed as g (C) photosynthesized or sequestered per g (N) deposited from the atmosphere. The stepwise method described in this paper (for forests only) first calculates dGPP/dN$_{dep}$, for both raw GPP$_{obs}$ and GPP\* standardized by meta-modelling following Eq. (10)-(17); this is then multiplied by different estimates of CSE (from observations or from modelling) to**

5 **provide an NEP (5-yr) or NECB (lifetime) equivalent. Quadratic regressions (Q) are used for productivity vs N$_{dep}$, whereby the mean tangent slope is calculated either over the whole N$_{dep}$ range (0-4.3 g (N) m$^{-2}$ yr$^{-1}$) (italics), or discarding sites with N$_{dep}$ larger than 2.5 g (N) m$^{-2}$ yr$^{-1}$ (bold). Uncertainty ranges are calculated from combined standard errors in dGPP/dN$_{dep}$ and in CSE. For comparison purposes only, are also displayed i) simple linear regression (L) slopes of EC-based (not standardized) GPP$_{obs}$ and NEP$_{obs}$ versus N$_{dep}$ for both forests and semi-natural vegetation; and ii) results of the meta-modelling standardization method applied directly to NEP$_{obs}$ instead of GPP$_{obs}$.**

| | FORESTS | | | | SEMI-NATURAL |
|---|---|---|---|---|---|
| | GPP$_{obs}$ | GPP\* | | | GPP$_{obs}$ |
| | | GPP$_{obs}$ x f$_{CLIM}$ | GPP$_{obs}$ x f$_{CLIM}$ x f$_{SOIL}$ | GPP$_{obs}$ x f$_{CLIM}$ x f$_{SOIL}$ x f$_{AGE}$ | |
| **Gross primary productivity per unit N$_{dep}$** dGPP /dN$_{dep}$ (g (C) g$^{-1}$ (N)) | *260 [38, 483 ]* [M,Q] **425 [203, 648]** [M,Q] *146 [89, 203]* [A,L] **432 [355, 509]** [A,L] | *146 [-121, 412]* [M,Q] **261 [-5, 528]** [M,Q] | *218 [-174, 609]* [M,Q] **273 [-119, 664]** [M,Q] | *190 [-375, 755]* [M,Q] **234 [-331, 799]** [M,Q] | *374 [275, 474]* [A,L] **504 [331, 677]** [A,L] |
| CSE$_{obs}$ \* (dGPP /dN$_{dep}$) | *64 [8, 136]* [M,Q] **105 [43, 182]** [M,Q] | *36 [-25, 116]* [M,Q] **64 [-1, 149]** [M,Q] | *53 [-36, 172]* [M,Q] **67 [-25, 187]** [M,Q] | *47 [-79, 213]* [M,Q] **57 [-69, 225]** [M,Q] | |
| CSE$_{5-yr}$ \* (dGPP /dN$_{dep}$) | *57 [8, 110]* [M,Q] **93 [42, 147]** [M,Q] | *32 [-25, 94]* [M,Q] **57 [-1, 120]** [M,Q] | *47 [-36, 138]* [M,Q] **59 [-25, 151]** [M,Q] | *41 [-78, 172]* [M,Q] **51 [-69, 181]** [M,Q] | |
| **Net ecosystem productivity per unit N$_{dep}$** (g (C) g$^{-1}$ (N)) — CSE$_{lifetime}$ \* (dGPP /dN$_{dep}$) | *47 [7, 91]* [M,Q] **77 [35, 122]** [M,Q] | *26 [-21, 78]* [M,Q] **47 [-1, 100]** [M,Q] | *39 [-30, 115]* [M,Q] **49 [-20, 125]** [M,Q] | *34 [-65, 143]* [M,Q] **42 [-57, 151]** [M,Q] | |
| | NEP$_{obs}$ | NEP\* | | | NEP$_{obs}$ |
| | | NEP$_{obs}$ x f$_{CLIM}$ | NEP$_{obs}$ x f$_{CLIM}$ x f$_{SOIL}$ | NEP$_{obs}$ x f$_{CLIM}$ x f$_{SOIL}$ x f$_{AGE}$ | |
| dNEP /dN$_{dep}$ | *108 [-118, 333]* [M,Q] **178 [-47, 403]** [M,Q] *71 [29, 114]* [A,L] **224 [157, 292]** [A,L] | *93 [-166, 352]* [M,Q] **161 [-98, 420]** [M,Q] | *120 [-162, 403]* [M,Q] **165 [-117, 447]** [M,Q] | *112 [-146, 370]* [M,Q] **146 [-112, 404]** [M,Q] | *89 [46, 132]* [A,L] **34 [-41, 109]** [A,L] |

GPP: gross primary productivity; NEP: net ecosystem productivity; NECB: net ecosystem carbon balance
GPP$_{obs}$ , NEP$_{obs}$ : observation-based (eddy covariance) GPP or NEP
GPP\*, NEP\*: GPP or NEP standardized through meta-modelling for the effects of climate (f$_{CLIM}$), soil (f$_{SOIL}$), age (f$_{AGE}$)
CSE: carbon sequestration efficiency

15 CSE$_{obs}$ = NEP$_{obs}$ / GPP$_{obs}$ (eddy covariance-based, mean value across all sites)
CSE$_{5-yr}$ = NEP$_{5-yr}$ / GPP$_{5-yr}$ (BASFOR model-based over 5-yr period, mean value across all sites)
CSE$_{lifetime}$ = NECB$_{lieftime}$/ GPP$_{lifetime}$ (BASFOR model-based over lifetime, mean value across all sites)
[Q]: calculated by quadratic regression
[L]: calculated by simple linear regression

20 [A]: calculated on the basis of all sites in the monitoring network (31 forests, 9 semi-natural sites)
[M]: calculated on the basis of the subset of 22 forest sites included in BASFOR meta-modelling