# Peer review of "Carbon/nitrogen interactions in European forests and semi-natural vegetation. Part II: Untangling climatic, edaphic, management and nitrogen deposition effects on carbon sequestration potentials"

_Biogeosciences, 2019_

## Referee Comment (RC1) · Anonymous Referee #1 · 2 Oct 2019

General remarks: Magnani et al. (2007) reported very large responses of forest carbon sequestration to nitrogen deposition. Several authors rapidly pointed out that the response proposed was way above previous estimates and direct observations in N addition studies. This apparent discrepancy has been discussed at length for more than a decade now, but there is still a need for a more stringent analysis of how dC responds to dN. The effort made in this manuscript is, therefore, most interesting and commendable. However, this model analysis is very complex. Many hours of careful

reading is needed to get an insight into how the model is constructed and how it handles the critical assumptions involved. A reader will also need to read the companion paper. Most readers will still be left with many queries. This is not uncommon in the case of modelling papers. Vital assumptions are deeply embedded and not clearly visible although the outcome is constrained by the assumptions. In fact, trust in the many reputed authors, rather than the apparent quality of the manuscript, drove me to read it once again. Could these complex matters and their analysis be made more understandable and transparent, respectively? I am not sure how, but would like to ask the authors to do their utmost. Hopefully, the comments below will be helpful when revising the ms. The treatment of the relations between Ndep and the internal forest N cycle is pivotal. A step ahead here is the use of local data on Ndep where possible and not just regional estimates. As regards the internal N cycle, the authors do not appear (e.g., lines 264-265) to handle that organic N sources (chiefly amino acids and peptides) are used by plants and probably dominate in less fertile systems, especially boreal forests. Inorganic N sources become dominant when the N supply is large relative to the biological demand. The authors may also reflect on the trends of decreasing leaching of inorganic N from forests in NE USA and N Europe. What in their models could drive this phenomenon? It could be related to higher tree growth (more C) in response to management or environmental change? How should C be coupled to N? On p. 15 potential net effects of N supply on C sequestration efficiency are discussed. The authors mention that C sequestration in a high C/N component like wood would be one explanation (used also by Magnani et al.) among the many complex and non-linear interactions between N and C. Is it at all possible and in line with findings in N-15 tracer studies that the majority of the N added goes into wood? The answer from many studies appears to be no (e.g., Nadelhoffer et al. 1999). This calls for an analysis of the physiological processes in which interactions between the cycles of N and C are particularly important. Modelling is necessary, there is no doubt about that, but it needs to make best use of all the data available including recent findings. These are many, but the authors could perhaps consider some, which describe non-linear biological controls

(e.g., Kallokoski et al. 2013, Tree Physiol. 33, 1145- , show that wood cell formation is similar in N-limited and N-fertilized trees during early summer, but then cease in the former but continue in the latter, and Högberg et al. 2010, New Phytol. 187, 485- , show that tree belowground C allocation is greatly reduced by additions of N; such relations may be interconnected). I would suggest that the authors rethink and reword a part of the reporting of results (lines 387-393). Firstly, forests 30-60 years are not young, especially not in Central Europe, in the sense that they have a low demand for N because of a low biomass as stated by the authors. Older forests may have a larger biomass for sure, but this is because of their trunks, tissues with much less biological activity and N demand than foliage and fine roots. On the contrary, 30-60 yrs-old forest most probably have fully closed canopies and a very high demand for N. Secondly, the idea of such forests leaching more N because of less canopy interception of water (and hence greater runoff), is also unlikely. Check with hydrologists if they see more runoff from forests 30-60 yrs-old than from older forests! Moreover, foresters would describe forests < 30 years old as young; in the context of rotational forestry in Europe 30-60 yrs-old forest are middle-aged.

More specific comments: Lines 73-74: Shouldn't this be phrased the other way around: "... with no further C uptake response at high Ndep levels (Ndep > 2.2-3 g m-2 yr-1) followed by large N losses by leaching and gaseous emissions." Line 140: add fires and insect attacks here. Line 145: in some regions, e.g. in N. Europe, there are many N-fertilizer experiments. Line 216: you write even, but maybe mean seven? Line 266-268: the use of another definition of NUE is widespread; I understand that you want to use an acronym, but it is unfortunate to use one that commonly has another meaning. May I also suggest that you use Nmob = N mobilized, rather than Nmin, which means that you overlook organic N compounds as N sources. Line 373: do you have clear evidence that Nmin does not change over time? Other authors discuss N oligotrophication and report that runoff of mineral N from forest decreases. Line 402: it is interesting to learn in which direction the non-linearity develops. Line 525: BASFOR may be mechanistic, but the vital interactions are not discussed and clarified in the

description of the model. Line 595: it is unclear if internal N supply is a component of soil fertility. Line 616: more thorough discussions about optimal allocation theory, especially C-N interactions, are found in, e.g., Franklin et al. (2012, Tree Physiol. 32, 648- ). Line 716: It is OK to cite Fog here, but why not cite authors, which discuss similar phenomena in forests (like Berg & Matzner 1997 Environ. Rev. 5, 1- ). Lines 743-744: below-ground autotrophic respiration does not exactly follow photosynthesis, but is also affected by seasonality in C below-ground allocation (Högberg et al. 2010 New Phytol. 187, 485- ). Line 780: what is the difference between fertility and nutrient availability in this context? All texts to Figures and Tables should be self-explanatory. Thus, acronyms should thus always be explained in these texts. Figures 3-6 and 8: these take some time to comprehend. A reader will need some guidance. And the text in the boxes are difficult to read and understand. Figures 9 & 10: the texts by the symbols are difficult to read as sometimes they come on top of each other.

---

## Referee Comment (RC2) · Anonymous Referee #2 · 25 Oct 2019

Summary: Flechard et al. use a "meta-modelling" analysis of forest C fluxes and balance to examine the response of these processes to N deposition for ∼30 sites (22 forests). They confirm that estimates of C gain from N deposition are smaller if environmental drivers are considered first.

General comments:

Overall, the analysis seems generally reasonable, and broadly supports the past reanalysis of forest C gain from N deposition by Sutton et al. (2008), which showed a much smaller C gain than imputed from the widely-critiqued Magnani et al. (2007) study. This text seems a bit long for that main take-home, with a data set only a bit larger – though analyzed in greater detail that that earlier dataset. It would be nice to have somewhat more focus in parsing this overall NEP response (i.e., more GPP vs less Reco or Rh?) beyond the surprisingly large reported GPP response.

Extensive discussion space is used on C sequestration efficiency (CSE =NEP/GPP), though it's not apparent quite what this adds over more in-depth examination of the individual C flux responses that go into this ratio. In particular, discussion of mechanistic explanations for the N effects on GPP and Rh (or Rsoil) would seem to be more directly related here – i.e., to explain saturation of the GPP response (discussed reasonably), or suppression of decomposition as a substantial portion of the overall dC/dN response.

The direct effects of N on decomposition process appears largely restricted to the last page of the Discussion, and they merit much greater attention earlier and throughout the manuscript.

The reported response of a saturation of the growth response to N dep, coinciding with an increase in N losses, fits exactly within expectations of N saturation theory (e.g., Aber et al. 1989, 1998, BioScience), which deserves more explicit recognition and discussion.

A less central suggestion: The authors state the importance of detailed site-level N deposition values over estimated modeled ones. While believable, this point would be supported more substantively by showing it directly, e.g., by comparing estimated v measured N deposition values, and quantitatively comparing dC/dN results for these two types of N deposition estimates.

Overall, the manuscript might be revised to reduce sometimes redundant-seeming extensive discussion of CSE, and provide greater and more direct emphasis on its novel

insights (beyond Sutton et al. 2008, or classic N saturation theory).

Detailed comments:

Abstract

Line 65 – somewhere in the abstract, specify the number of sites included in this analysis

Line 67-71 – The reduction of dC/dN from considering factors other than N deposition was for GPP, not NEP, right? that should be clear in the abstract, which describes this response in terms of C sequestration, which generally aligns more closely with NEP. Similarly, be clear about which C cycle term yielded the 40-50 gC/gN response.

Line 74 – text indicates that the dC/dN response saturates above 2.5-3.0 gN/m2/yr "due to" leaching and other losses.. but the latter don't appear to be measured here? If this attribution is from the model analyses, do indicate that as a modeled result.

Main text:

Line 81-93. The cited references provide examples of experimental studies that indicate little or no increase in C sequestration from N addition, and might be presented as some of the conflicting evidence for a universal N-deposition induced C sink, rather than challenging the entire notion of this phenomenon in its entirety

Line 92. Is the Dezi et al. 2010 reference a model-based analysis of dC/dN? If so, clarify that it's different from the empirical approaches of the other studies

Line 93-97. In this review of dC/dN values – and throughout the manuscript – be clear as to which values pertain to which C pool (i.e., tree, soil, or whole ecosystem) or which specific C cycle processes.

Line 107 – 120. It's not wholly apparent why this review of basic N balance processes is needed? That is, nearly every forest N budget shows that new N deposition supplies only a small fraction of plant annual N demand compared to internal N recycling; the

key (missing?) point is that the value of N deposition is that it may be acquired directly or at little energetic cost to plants, and can accumulate over time.

Line 121. Add specific citations to this sentence critiquing "some previous estimates" for failing to account for factors other than N deposition.

Line 122-130. This section seems somewhat oversimplified in pitching its novelty: N deposition can, but does not necessarily covary with gradients of other environmental variables; this covariation often depends on the geographic region selected. The Magnani et al. (2007) simple regression analysis indeed failed to consider this covariation, but its problems seemed very effectively addressed by the Sutton et al. (2008) reanalysis, in demonstrating the need to consider variation in factors besides N deposition. Is the goal in this manuscript to do a similar analysis of tower-based C balance measurements in greater depth than that one? Other gradient analyses have also considered N deposition along with other environmental drivers, sometimes also considering nonlinear responses (e.g., Solberg et al. 2009, Thomas et al. 2010).

Line 175-176. Elaborate on what exactly is meant by "soil C pools rely on various assumptions or empirical models for their estimation." Assumed and modeled soil C can often vary markedly from measured values; how well do these assumptions work?

Line 184. Specify the minimum and/or mean number of years of EC data are used to compute the C fluxes of interest here.

Line 202-204. Does this model's soil dynamics allow it to represent the inhibitory effect of N deposition on soil decomposition? If not, this point should receive explicit attention in the Methods and/or Discussion, on how this effect of N was considered in this analysis.

Line 207-208. It's certainly difficult to reliably simulate N loss fluxes to DON and N2, and it's correspondingly understandable that this set of model-based estimates would not include them. However, when measured, these fluxes can dominate ecosystem N

loss fluxes and should thus receive more attention as to the uncertainties introduced by their omission in these calculations.

Line 246 / Section 2.2.3 – why focus so much text here and the Discussion on the ratio of C sequestration _efficiency_, CSE, (NEP / GPP and similar), rather than NEP itself and its component parts (i.e., increased GPP? Suppressed Reco or Rh)? The Introduction provides no context or central questions for focusing on questions concerning the CSE ratio, and so this emphasis seems somewhat unexpected and extraneous here, and the lengthy text in the Discussion (∼line 600-700)

Line 265 – late text (line 273) indicate N fixation wasn't considered, so be consistent with that point here

Line 314-316 – identify what is "the broad pattern of GPP vs N dep. in Flechard et al. (2019)."

Line 373-374 – in this N balance (N mineralisation + N dep – N plant uptake – N leach – N emissions), what about accumulation of N in soils or soil organic matter? Often a very large if not the largest sink.

Line 457 – suggest "at the low_er_ N dep sites..." That is, 1.0 g/N/m2 is often considered elevated.

Line 466 – clarify which "this set" is meant – higher or lower N dep group?

Line 468-485. These dC-GPP/dN values are simply _enormous_ and correspondingly difficult to fathom – even when reduced from 425 to 234 gC/gN! How do these compare to empirical NPP values? Presumably a 50% GPP to NPP efficiency would yield something like 212 to 117 gC/gN, still well beyond empirical NPP responses. How / why is it so large compared to eventual dC-NEP/dN response?

Line 495-502. These dC-NEP/dN values (∼40-60 gC/gN) seem more consistent with empirical responses: to what extent are these values due to modeled plant vs soil C sequestration? Is the soil C sink from additional litterfall inputs or from suppressed

decomposition?

Line 535. Yes, the results here seem to confirm that of Sutton et al. (2008). How does this analysis provide additional insights beyond that one?

Line 537. Provide citations for observations of N losses. Thresholds of 0.8 – 1.0 g N/m2/yr for N leaching have been reported commonly (e.g., MacDonald et al. 2003, Global Change Biology, and similar).

Line 540. These responses are exactly as expected – i.e., The saturating response of ecosystem NPP to N deposition, and corresponding increase in N losses, are standard predictions of classic N saturation theory as originally proposed (e.g., Aber et al. 1989 & 1998, BioScience). Discuss how this work provides an advance over that prior set of expectations.

Line 560-570. This paragraph states that the detailed, more-precise N deposition measurements improve calculation of dC/dN responses.. and while plausible, it should first be demonstrated how these estimates compare with the simpler alternative.

Line 580-582. This conclusion on Reco vs N dep does not seem to have been discussed in the Results?

Line 585. Per above, the annual N input is small relative to annual N demand. But its accumulation over time can support a much larger fraction of N demand.

Line ∼590. This would seem one of several places to mention the effect of N on decomposition

Line 598 – 708. It's not apparent why so much emphasis is placed on carbon sequestration efficiency (CSE = NEP / GPP) rather than the component C fluxes (GPP, NPP, NEP, Rh). It seems somewhat redundant with these other responses, and a direct outcome of individual responses. What additional insights does it provide?

Line 649. A large soil C stock doesn't necessarily indicate higher heterotrophic respiration responses – and can result from the opposite situation (i.e., lower Rhet allows more soil C to accumulate).

Line 652. The history of N and S deposition at this site (EN8) indeed might be important. What about considering cumulative N deposition across the range of sites?

Line 709 onward. This seems quite late for a first substantive mention of the direct effects of extra N on decomposition and belowground processes, often shown to be of comparable magnitude as many aboveground responses (e.g., Janssens et al. 2010, Frey et al. 2014). In addition: does the modelling approach consider these processes, or would it miss them?

Line 714. Add citation(s) for this "traditional theory of role of N..."54

Line 737-742. This content seems more appropriate to the Methods, as a general data synthesis activity part of this study. Similarly line 743-763 seem more appropriate to Results.

Line 775. NEP dC/dN of 40-50 is on the lower end of inventory? Many inventory-based assessments seem to show something in this range, with ∼20-25 for trees and 20-25 gC/gN for soil.

---

## Author Comment (AC1) · 6 Dec 2019

**"Carbon / nitrogen interactions in European forests and semi-natural vegetation. Part II: Untangling climatic, edaphic, management and nitrogen deposition effects on carbon sequestration potentials" by Chris R. Flechard et al.**

**Author's Response to Referees**

We are thankful to both referees for their interest in our study, for their constructive comments, sharp insights and challenging questions, which have helped improve the manuscript. For clarity's sake, we provide our point-by-point **responses to each comment in blue**, and provide **suggested changes to the manuscript text in green**.

**Anonymous Referee #1**

**General remarks**:

Magnani et al. (2007) reported very large responses of forest carbon sequestration to nitrogen deposition. Several authors rapidly pointed out that the response proposed was way above previous estimates and direct observations in N addition studies. This apparent discrepancy has been discussed at length for more than a decade now, but there is still a need for a more stringent analysis of how dC responds to dN. The effort made in this manuscript is, therefore, most interesting and commendable.

However, this model analysis is very complex. Many hours of careful reading is needed to get an insight into how the model is constructed and how it handles the critical assumptions involved. A reader will also need to read the companion paper. Most readers will still be left with many queries. This is not uncommon in the case of modelling papers. Vital assumptions are deeply embedded and not clearly visible although the outcome is constrained by the assumptions. In fact, trust in the many reputed authors, rather than the apparent quality of the manuscript, drove me to read it once again. Could these complex matters and their analysis be made more understandable and transparent, respectively? I am not sure how, but would like to ask the authors to do their utmost. Hopefully, the comments below will be helpful when revising the ms.

When we submitted this paper (Part II) and its companion paper (Part I) as components of the same study, a more detailed description of the BASFOR model, and of the way it was implemented in this study, were included in Part I. We only originally provided a cursory description of the model in Part II, but we now recognize that, with Part II being much more model-oriented than Part I (which is more measurements-oriented), it makes sense to move the detailed description to Part II, while keeping a short outline in Paper I and refer there to Part II for details. We will do this in the revised papers, and at the same time this will achieve a significant reduction in the length of the Part I paper, which was recommended by one of its Referees.

We do not copy to this response letter the whole model description, but this can be found in Section 2.6 of Part I (https://www.biogeosciences-discuss.net/bg-2019-333/), and will be moved to Section 2.2.1 of the Part II paper, alongside with the relevant supplementary information (Figures S4, S5 of the Part I Supplement), now moved to the Part II Supplement.

In addition, we remind the Reviewer that, as stated on line 198, we provide a link to the online BASFOR model description at https://github.com/MarcelVanOijen/BASFOR. This link provides the complete Fortran code of the model, which makes it completely transparent with respect to model structure, embedded assumptions and data flows between the model parts.

Further, to ease the reader into the flow of the paper, to better introduce the modelling background and framework that underpin the C and N budget calculations of Fig. 1-4, and to illustrate graphically the temporal dimension of all the different forest sites of this study in relation to changing $N_r$ deposition and increasing atmospheric $CO_2$ through the 20[th] century, we propose to add a figure (as the new Figure 1) to the main body of the paper. This figure is shown

hereafter as Fig. R1 and will depict the modelled (baseline) time course of GPP, NEP, N deposition and N losses for all study sites over the period 1900-2010. In the text, we will add a short paragraph to describe this new Fig. 1 at the start of Results – 3.1, just before the description of the Sankey plots for C and N budgets:

'*3.1 Short term (5-yr) versus lifetime C and N budgets from ecosystem modelling*

The time course of modelled (baseline) GPP, NEP and total leaching and gaseous N losses is shown in Fig. 1 for all forests sites over the 20[th] century and until 2010, forced by climate, increasing atmospheric $CO_2$ and by the assumed time course of $N_r$ deposition over this period (Fig. 1A). For each stand, regardless of its age and establishment date, an initial phase of around 20-25 years occurs, during which GPP increases sharply from zero to a potential value attained upon canopy closure (Fig. 1B), while NEP switches from a net C source to a net C sink after about 10 years (Fig. 1D). Initially $N_r$ losses are very large (typically of the order of 10 g (N) m$^{-2}$ yr$^{-1}$), then decrease rapidly to pseudo steady-state levels when GPP and tree N uptake reach their potential.

After this initial phase, modelled GPP increases steadily in response to increasing $N_{dep}$ and atmospheric $CO_2$, but only for the older stands established before around 1960, i.e. those stands that reach canopy closure well before the 1980's, when $N_r$ deposition is assumed to start declining. Thereafter, modelled GPP ceases to increase, except for the recently established stands that have not yet reached canopy closure. The stabilization of GPP for mature trees at the end of the 20[th] century in the model is likely a consequence of the effects of decreasing $N_{dep}$ and increasing $CO_2$ cancelling each other out to a large extent. In parallel, modelled total N losses start to decrease after the 1980's, even for sites long past canopy closure (Fig. 1E-F), but this mostly applies to stands subject to the largest $N_{dep}$ levels, i.e. where the historical high $N_{dep}$ of the 1980's, added to the internal N supply, were well in excess of growth requirements in the model.

These temporal interactions of differently-aged stands with changing $N_{dep}$ and $CO_2$ over their lifetimes therefore impact C and N budget simulations made over different time horizons. Modelled C and N budgets are represented schematically in Fig. 2 and Fig. 3, respectively, as «Sankey» diagrams …'

We believe all these changes will help make '…*the analysis … more understandable and transparent…*', as requested by the Referee. The added text and the new Figure 1 also introduce the topic of downward trends in N leaching since the end of the 20[th] century, raised in another comment by Referee 1, which will then be further discussed in the paper (see below).

[Figure]

Figure R1 *(= new Fig. 1 of revised paper)*. Time courses of (A) assumed atmospheric $N_r$ deposition and $CO_2$ mixing ratio, and baseline model simulations of (B) gross primary productivity (GPP), (C) GPP normalized to the 2010 value, (D) net ecosystem productivity (NEP), (E) total N losses by leaching and gaseous emissions, and (F) total N losses normalized to 2010.

The treatment of the relations between Ndep and the internal forest N cycle is pivotal. A step ahead here is the use of local data on Ndep where possible and not just regional estimates. As regards the internal N cycle, the authors do not appear (e.g., lines 264-265) to handle that organic N sources (chiefly amino acids and peptides) are used by plants and probably dominate in less fertile systems, especially boreal forests. Inorganic N sources become dominant when the N supply is large relative to the biological demand.

The referee is right in pointing out that we do not consider organic N sources as part of the nitrogen supply and uptake by trees. The BASFOR model does not account for dissolved organic N (DON) forms with a relatively low molecular weight (e.g. amino acids and small peptides), that may be taken up by roots alongside dissolved inorganic N (DIN, mainly ammonium and nitrate). Proteins and other N-containing molecules from the litter layers, roots and soil organic matter (with slow or fast turnover) are decomposed, in the model pools, into mineral N, $CO_2$ and $H_2O$, but the large number of intermediate organic degradation products are not explicitly simulated. We therefore cannot, through modelling, address the Referee's question about the importance of organic N uptake in boreal forests and elsewhere.

However, the role of organic N is perhaps less clear than the Referee suggests, as far as the dominance of organic over inorganic N uptake in less fertile systems and boreal forests is concerned. There is no doubt that roots can take up some forms of bio-available organic N, and that in acidic soils containing large amounts of organic matter, soluble N is dominated by organic forms, with amino acids making up to 10-20% of the total DON pool, and correspondingly low DIN concentrations (Jones and Kielland, 2002, Soil Biology and Biogeochemistry 34, 209-219). But the ecological significance of this N acquisition pathway remains controversial (Moreau et al., 2019, Functional Ecology 33:540–552). It appears that DON has very fast turnover to ammonium (Jones and Kielland, 2002), so that ultimately the trees take up much DIN as well, despite the high concentrations of DON in the soil. It is also important that a large fraction of DON is originally lost from plant roots, so that its uptake compensates this N-leak, rather than leading to an important pathway of net N-acquisition by plants (Jones et al., 2005, Soil Biol Biochem 37, 413-423). Further, many forms of DON are by-products of microbial breakdown of humic substances, that are not necessarily bio-available (Warren, 2014, Plant Soil, 375, 1–19).

We nonetheless agree that the revised manuscript should make it clear that i) the current state of knowledge is that soil plant N nutrition relies on both dissolved organic and inorganic N forms, albeit with uncertain partitioning, and ii) our ecosystem model does not consider organic N supply and uptake by trees. We will therefore make the following additions to the BASFOR description of N cycling that will be transferred from the Part I paper:

'…Dissolved inorganic nitrogen (DIN) *is taken up by the trees from the soil, and nitrogen returns to the soil with senescence of leaves, branches and roots, and also when trees are pruned or thinned. Part of the N from senescing leaves is re-used for growth. The availability of mineral nitrogen is a Michaelis-Menten function of the mineral nitrogen pool and is proportional to root biomass.* The model does not include a dissolved organic nitrogen (DON) pool and therefore does not account for the possible uptake of bio-available DON forms (e.g. amino acids, peptides) by trees. *Transformation between the four soil nitrogen pools are similar to those of the carbon pools, with mineral nitrogen as the loss term…'*

Further, in the discussion on challenges and limitations in the modelling study, line 583, we will mention the uncertainty in $N_{supply}$ related to the non-inclusion of DON supply and uptake:

'iv) *Nitrogen deposition likely contributes a minor fraction (on average 20% according to the model) of total ecosystem N supply (heavily dominated by soil organic N mineralization), except for the very high deposition sites (up to 40%).* The fraction of $N_{dep}$/$N_{supply}$ may even be smaller considering the pool of DON (not included in BASFOR), from which bio-available organic N forms may be taken up by trees in significant quantities in non-fertile, acidic organic soils (Jones and Kielland, 2002; Warren, 2014; Moreau et al., 2019). *Thus, in many cases the Ndep fertilisation effect may be marginal and difficult to detect, because it may be smaller than typical measurement uncertainties and noise in C and N budgets.'*

Additional references:

Jones, D.L. and Kielland, K.: Soil amino acid turnover dominates the nitrogen flux in permafrost-dominated taiga forest soils, Soil Biology Biochem., 34, 209–219, https://doi.org/10.1016/S0038-0717(01)00175-4, 2002.

Moreau, D., Bardgett, R.D., Finlay, R.D., Jones, D.L. and Philippot, L.: A plant perspective on nitrogen cycling in the rhizosphere, Funct. Ecol., 33, 540–552, https://doi.org/10.1111/1365-2435.13303, 2019.

Warren, C.R.: Organic N molecules in the soil solution: What is known, what is unknown and the path forwards, Plant Soil, 375, 1–19, https://doi.org/10.1007/s11104-013-1939-y, 2014.

The authors may also reflect on the trends of decreasing leaching of inorganic N from forests in NE USA and N Europe. What in their models could drive this phenomenon? It could be related to higher tree growth (more C) in response to management or environmental change? How should C be coupled to N?

In this paper we set out to tackle the issue of how regional (spatial) differences in climate (and also soil and forest age structure) influence the response of C sequestration to N deposition. However, we did not specifically seek to address the issue of how forest N losses have responded over the last 2-3 decades to changes in management, climate or other environmental factors. We unfortunately do not possess the depth of historical data (neither long-term DIN leaching nor NPP data) necessary to investigate this empirically at our sites. Others have reported long term changes in N or nitrate losses/export over the last few decades, as pointed out by the Referee (e.g. Verstraeten et al., 2012, Atmospheric Environment 62: 50-63; Goodale et al., 2003, Ecosystems 6:75–86; Bernal et al., 2012, PNAS, 109, 9, 3406–3411)

From a modelling viewpoint, the main factors in BASFOR likely to reduce DIN leaching over the last 2-3 decades are the assumed decreasing trends in total $N_{dep}$ since the 1980's (see Fig. R1-A below), and the increasing trends of GPP and NPP (and increasing N uptake) for the forest stands still at the aggrading stage or in response to increasing atmospheric $CO_2$ (Fig. R1-A). We did not include a changing climate in our simulations over the 20[th] century (though inter-annual meteorological variability is accounted for). The forests included in the study range from ~25 to >150 years old. This means that the younger stands (<40 years) mostly experienced decreasing $N_{dep}$ (since ~1980), and sharply increasing NPP as part of their initial growth phase (before reaching peak LAI and canopy closure). By contrast, the oldest stands experienced increasing $N_{dep}$ and increasing NPP during a large part of the 20[th] century, and then decreasing $N_{dep}$ after ~1980 and stabilized NPP over the last 30 years. The age of the forest therefore influences, in model simulations, the extent to which decreasing $N_{dep}$ after 1980 would translate into reduced N leaching. But the level of $N_{dep}$ in itself is important: for the high $N_{dep}$ locations, a more important reduction in N losses may be expected, following the decrease in $N_{dep}$, than at N-limited sites, as suggested in Fig. R1E-F.

We have described these patterns in the new sub-section added to 3.1 together with the new Figure 1 (= Fig. R1 introduced above). To further address the referee's question as to what could drive observed recent downward trends in leaching in the model, we have made three additional model scenario runs besides the baseline simulation of Fig. R1. The corresponding figures are shown below (Fig. R2, R3, R4) and will be added to the paper's supplement as Figures S7, S8 and S9. These additional model runs test the effects of increasing $N_{dep}$ or $CO_2$ separately on forest productivity and N losses. In Fig. R2, the $CO_2$ mixing ratio is kept constant at 310 ppm (~ the mean value over the period 1900-2010); in Fig. R3, $CO_2$ increases exponentially as in the baseline run, but $N_{dep}$ is constant at 1.5 g (N) m$^{-2}$ yr$^{-1}$ at all sites; in Fig. R4, $N_{dep}$ is constant at 3.0 g (N) m$^{-2}$ yr$^{-1}$ at all sites. These additional model runs and supplementary figures will be referred to in the discussion of the revised version, by adding the following text as part of *Section 4.2 Limitations and uncertainties in the approach for quantifying the dC/dN response:*, starting line 587:

'...*smaller than typical measurement uncertainties and noise in C and N budgets*.

A further limitation to our estimates of the dC/dN response, based on the analysis of the spatial (inter-site) variability in C and N fluxes, is that these forests are not in steady state with respect to $N_r$ deposition and ambient $CO_2$. Some stands have been affected by, and may be slowly recovering from, excess $N_r$ deposition in the second half

of the 20[th] century; while the more remote sites may always have been N-limited. Figure 1 showed that the modelled GPP of the older forests increased through most of the 20[th] century, but stabilized when $N_{dep}$ started to decrease after the 1980's, while total N losses also declined over the last 2-3 decades. This is consistent with observations of decreasing N (nitrate) leaching at long term study sites in N-E USA (Goodale et al., 2003; Bernal et al., 2012) and N Europe (Verstraeten et al., 2012; Johnson et al., 2018; Schmitz et al., 2019).

In our model analysis, the declining trend in $N_r$ deposition appears to be the primary driver for the modelled reduced N losses since the 1980's. This can be inferred from additional model scenario runs shown in Fig. S7, S8 and S9 of the Supplement. In Fig. S7, a constant $CO_2$ mixing ratio of 310 ppm (i.e. the mean value over the period 1900-2010), used instead of the exponential increase since the 19[th] century, does not greatly alter overall productivity patterns, nor the decreasing trend in N losses over the period 1980-2010 (Fig S7E-F), compared with the baseline run (Fig. 1). By contrast, in scenarios shown in Fig. S8 and S9, the constant $N_{dep}$ levels at all sites of 1.5 and 3.0 g (N) m$^{-2}$ yr$^{-1}$, respectively, together with the exponential $CO_2$ increase, remove the decreasing trends in $N_r$ losses over the period 1980-2010. Meanwhile, in constant $N_{dep}$ scenarios the increase in GPP over the whole period is fairly monotonous, in response to a steadily increasing $CO_2$ (Fig. S8B-C), without the inflexion point around 1980 simulated in the baseline run (Fig 1B-D). In real-life stands, however, decadal decreases in N losses or exports have been observed without any significant reductions in $N_{dep}$ (Goodale et al., 2003). Other potential factors such as increased denitrification, longer growing season, plant N accumulation, changes in soil hydrological properties or temperature, historical disturbances may also play a role (Bernal et al., 2012). Many such factors are not considered in our model, and neither is long term climate change.

*The EC-based flux data suggest that the Ndep response of forest productivity…'*

Additional references

Bernal, S., Hedin, L.O., Likens, G.E., Gerber, S. and Buso, D.C.: Complex response of the forest nitrogen cycle to climate change, P. Natl. Acad. Sci. USA, 109(9), 3406–3411, https://doi.org/10.1073/pnas.1121448109, 2012.

Goodale, C.L., Aber, J.D. and Vitousek, P.M.: An unexpected nitrate decline in New Hampshire streams, Ecosystems, 6, 75–86, https://doi.org/10.1007/s10021-002-0219-0, 2003.

Johnson, J., Graf Pannatier, E., Carnicelli, S., Cecchini, G., Clarke, N., Cools, N., Hansen, K., Meesenburg, H., Nieminen, T.M., Pihl-Karlsson, G., Titeux, H., Vanguelova, E., Verstraeten, A., Vesterdal, L., Waldner, P. and Jonard, M.: The response of soil solution chemistry in European forests to decreasing acid deposition, Glob. Change Biol., 24, 3603–3619, https://doi.org/10.1111/gcb.14156, 2018.

Schmitz, A., Sanders, T.G.M., Bolte, A., Bussotti, F., Dirnböck, T., Johnson, J., Peñuelas, J., Pollastrini, M., Prescher, A.-K., Sardans, J., Verstraeten, A. and de Vries, W.: Responses of forest ecosystems in Europe to decreasing nitrogen deposition, Environ. Pollut., 244, 980 –994, https://doi.org/10.1016/j.envpol.2018.09.101, 2019.

Verstraeten, A., Neirynck, J., Genouw, G., Cools, N., Roskams, P. and Hens, M.: Impact of declining atmospheric deposition on forest soil solution chemistry in Flanders, Belgium, Atmos. Environ., 62, 50–63, https://doi.org/10.1016/j.atmosenv.2012.08.017, 2012.

[Figure]

**Figure R2.** *(= additional Fig. S7 in revised Supplement)*. Alternative model scenario using a constant $CO_2$ mixing ratio of 310 ppm through the entire modelling period, showing simulations of (B) gross primary productivity (GPP), (C) GPP normalized to the 2010 value, (D) net ecosystem productivity (NEP), (E) total N losses by leaching and gaseous emissions, and (F) total N losses normalized to 2010.

[Figure]

Figure R3. *(= additional Fig. S8 in revised Supplement)*. Alternative model scenario using a constant $N_{dep}$ level of 1.5 g (N) m$^{-2}$ yr$^{-1}$ at all sites through the entire modelling period, showing simulations of (B) gross primary productivity (GPP), (C) GPP normalized to the 2010 value, (D) net ecosystem productivity (NEP), (E) total N losses by leaching and gaseous emissions, and (F) total N losses normalized to 2010.

[Figure]

Figure R4. *(= additional Fig. S9 in revised Supplement)*. Alternative model scenario using a constant $N_{dep}$ level of 3.0 g (N) m$^{-2}$ yr$^{-1}$ at all sites through the entire modelling period, showing simulations of (B) gross primary productivity (GPP), (C) GPP normalized to the 2010 value, (D) net ecosystem productivity (NEP), (E) total N losses by leaching and gaseous emissions, and (F) total N losses normalized to 2010.

On p. 15 potential net effects of N supply on C sequestration efficiency are discussed. The authors mention that C sequestration in a high C/N component like wood would be one explanation (used also by Magnani et al.) among the many complex and nonlinear interactions between N and C. Is it at all possible and in line with findings in N-15 tracer studies that the majority of the N added goes into wood? The answer from many studies appears to be no (e.g., Nadelhoffer et al. 1999).

In the text p. 13-14 (lines 551-556) the argument we make is that the $dNEP/dN_{dep}$ response should logically be steeper in forests than in short, non-woody semi-natural (SN) vegetation, since a large fraction of C is stored in high C/N components (wood) in forests, versus SOM with a much lower C/N ratio in non-woody SN ecosystems. By that statement, we had not actually meant to extrapolate to nitrogen storage and to imply that "…*the majority of the N added goes into wood*…", as suggested by the Referee. Nevertheless, the most recent synthesis of $^{15}$N tracer experiments by Du and de Vries (Environ. Pollut., 242, 1476–1487, 2018) does indeed suggest that tree biomass is the primary sink for the added nitrogen in both boreal and temperate forests (about 70%), with the remaining 30% retained in soil. For carbon, our forest ecosystem model suggests that up to 60-80% of NECB is sequestered in above-ground biomass (branches and stems), which would be consistent with the partitioning of the N sink. Since we do not have a model for our non-woody SN sites, we cannot however provide a comparison between the two types of vegetation (F *versus* non-woody SN) of the modelled fractions of C and N stored above and below ground.

Regarding other studies on the fate of added N, in the introduction to the paper we did originally provide some background and references, l 100-106 on p.3, which we have now expanded and revised with a contrasting view on the contribution of tree and soil carbon sequestration in response to N deposition:

'…The questions of the allocation and fate of both the assimilated carbon (Franklin et al., 2012) and deposited nitrogen (Nadelhoffer et al., 1999; Templer et al., 2012; Du and de Vries, 2018) appear to be crucial. It has been suggested that $N_r$ deposition plays a significant role *in promoting the carbon sink strength only if N is stored in woody tissues with high C/N ratios (>200–500) and long turnover times, as opposed to soil organic matter (SOM) with C/N ratios that are an order of magnitude smaller (de Vries et al., 2008). Nadelhoffer et al. (1999) argued on the basis of a review of $^{15}$N tracer experiments that soil, rather than tree biomass, was the primary sink for the added nitrogen in temperate forests*. However, based on a recent synthesis of $^{15}$N tracer field experiments (only including measurements of $^{15}$N recovery after > 1 year of $^{15}$N addition), Du and de Vries (2018) estimated that tree biomass was the primary sink for the added nitrogen in both boreal and temperate forests (about 70%), with the remaining 30% retained in soil. A*t sites with elevated N inputs, increasingly large fractions are lost as nitrate ($NO_3^-$) leaching. Lovett et al. (2013) found in north-eastern US forests that added N increased C and N stocks and the C/N ratio in the forest floor, but did not increase woody biomass or aboveground NPP.*"

Additional references:

Du, E. and de Vries, W.: Nitrogen-induced new net primary production and carbon sequestration in global forests, Environ. Pollut., 242, 1476–1487, https://doi.org/10.1016/j.envpol.2018.08.041, 2018.

Franklin, O., Johansson, J., Dewar, R.C., Dieckmann, U., McMurtrie, R.E., Brännström, Å and Dybzinski, R.: Modeling carbon allocation in trees: a search for principles, Tree Physiol., 32, 648–666, https://doi.org/10.1093/treephys/tpr138, 2012.

Templer, P.H., Mack, M.C., Chapin, F.S. III, Christenson, L.M., Compton, J.E., Crook, H.D., Currie, W.S., Curtis, C.J., Dail, D.B., D'Antonio, C.M., Emmett, B.A., Epstein, H.E., Goodale, C.L., Gundersen, P., Hobbie, S.E., Holland, K., Hooper, D.U., Hungate, B.A., Lamontagne, S., Nadelhoffer, K.J., Osenberg, C.W., Perakis, S.S., Schleppi, P., Schimel, J., Schmidt, I.K., Sommerkorn, M., Spoelstra, J., Tietema, A., Wessel, W.W. and Zak, D.R.: Sinks for nitrogen inputs in terrestrial ecosystems: A meta-analysis of $^{15}$N tracer field studies, Ecology, 93, 1816–829, https://doi.org/10.1890/11-1146.1, 2012.

This calls for an analysis of the physiological processes in which interactions between the cycles of N and C are particularly important. Modelling is necessary, there is no doubt about that, but it needs to make best use of all the data available including recent findings. These are many, but the authors could perhaps consider some, which describe non-linear biological controls (e.g., Kallokoski et al. 2013, Tree Physiol. 33, 1145- , show that wood cell formation is similar in N-limited and N-fertilized trees during early summer, but then cease in the former but continue in the latter, and Högberg et al. 2010, New Phytol. 187, 485- , show that tree belowground C allocation is greatly reduced by additions of N; such relations may be interconnected).

We have acknowledged the limitations in modelling in our study in several places in the discussion, with respect to mechanisms that are not included (e.g. l. 580-582, l.594-596, l.659-661, etc), though at this point we are unable to make such changes in the model. We did mention (l. 720-723) that '…*excess Nr deposition reduces soil – especially heterotrophic – respiration in many temperate forests*…' through, amongst other things (Janssens et al., 2010) , '… *a decrease in below-ground C allocation and the resulting root respiration, permitted by a lesser need to develop the rooting system when more N is available (see also Alberti et al., 2015)*…'. But we will make it clearer in Section 4.2.1 ('*Limitations and uncertainties*…') that this and other important mechanisms, such as mentioned above by the Referee, affect non-linear C/N relations but are not included in the model, and therefore represent a significant limitation to our analysis. We will add a fifth item (v) in section 4.2.1, line 587:

'v) Non-linear biological controls that affect C/N relations but are not explicitly considered in the model. For example, BASFOR does consider that N addition can reduce below-ground C allocation (e.g. Högberg et al., 2010), resulting in decreased soil $R_{aut}$ and $R_{het}$ (Janssens et al., 2010), but does not account for the possible consequences of a stimulation of wood cell formation from mid-summer onwards and a delay in the cessation of tracheid production in late season (Kalliokoski et al., 2013).'

Additional References:

Högberg, M.N., Briones, M.J.I, Keel, S.G., Metcalfe, D.B., Campbell, C., Midwood, A.J., Thornton, B., Hurry, V., Linder, S., Näsholm, T. and and Högberg, P.: Quantification of effects of season and nitrogen, supply on tree below-ground carbon transfer to ectomycorrhizal fungi and other soil organisms in a boreal pine forest, New Phytol., 187, 485–493, https://doi.org/10.1111/j.1469-8137.2010.03274.x, 2010.

Kalliokoski, T., Mäkinen, H., Jyske, T., Nöjd, P. and Linder, S.: Effects of nutrient optimization on intra-annual wood formation in Norway spruce, Tree Physiol., 33, 1145–1155, https://doi.org/10.1093/treephys/tpt078, 2013.

I would suggest that the authors rethink and reword a part of the reporting of results (lines 387-393). Firstly, forests 30-60 years are not young, especially not in Central Europe, in the sense that they have a low demand for N because of a low biomass as stated by the authors. Older forests may have a larger biomass for sure, but this is because of their trunks, tissues with much less biological activity and N demand than foliage and fine roots. On the contrary, 30-60 yrs-old forest most probably have fully closed canopies and a very high demand for N.

Secondly, the idea of such forests leaching more N because of less canopy interception of water (and hence greater runoff), is also unlikely. Check with hydrologists if they see more runoff from forests 30-60 yrs-old than from older forests! Moreover, foresters would describe forests < 30 years old as young; in the context of rotational forestry in Europe 30-60 yrs-old forest are middle-aged.

We agree that the adjective '*young*' was not appropriate to describe the 30-60 yr age class, even if we in fact mostly used the comparative '*young**er**', by opposition to 'mature' (>80 yrs). We have shown in our additional figure R1 (see above) that, in the model, the initial phase of rapidly increasing GPP and fast decreasing N losses lasts around 20 years, therefore we agree that for trees aged 30-60 the NUE or $N_{loss}$ fractions, calculated over a recent 5-yr period, are no longer affected by that initial growth phase characterized by large N losses. Nevertheless, if NUE or $N_{loss}$ fractions are calculated over the whole period since the forest was established ('lifetime'), then the initial ~20-yr

phase has a greater weight in the 30-60 yr-old forests lifetime calculation than in the >80 yr-old stands. We will therefore rephrase this paragraph in the following way:

Line 383 '…By contrast, the analogous term for nitrogen, the $N_{upt}$ fraction of total $N_{supply}$, is a much more variable term, both between sites of the network and between the 5-yr and lifetime simulations (Fig. 2, S4–S6). Modelled lifetime CSE and NUE values are compared in Fig. 3 with the 5-yr values, as a function of stand age, indicating that (i) the older forests of the network (age range ~80–190 yrs) tend to have larger NUE than younger or middle aged forests (~30–60 yrs), but (ii) the difference in NUE between the two age groups is much clearer if NUE is calculated over the whole period since planting (lifetime). As shown in *(the new)* Fig. 1 *(= Fig. R1 above)*, BASFOR predicts large N losses in young stands (<20-25 years), in which lower N demand by a smaller living biomass, combined in the early years with enhanced $N_{miner}$ from higher soil temperature (canopy not yet closed) and with a larger drainage rate (smaller canopy interception of incident rainfall), all lead to larger NMIN losses. The 22 forests sites of this study were past this juvenile stage, but observation (ii) is a mathematical consequence of high N losses during the forest's early years having a larger impact on lifetime calculations in middle-aged than mature forests. NUE tends to reach 70-80% on average after 100 years and is smaller calculated from lifetime than from a 5-yr thinning-free period. For forests younger than 60 years, lifetime NUE is only around 60%.'

**More specific comments**:

Lines 73-74: Shouldn0t this be phrased the other way around: ". . . with no further C uptake response at high Ndep levels (Ndep > 2.2-3 g m-2 yr-1) followed by large N losses by leaching and gaseous emissions."

We are not sure if the Referee means '*followed*' in a temporal sense. The absence of productivity response to $N_{dep}$ above a certain threshold is derived in our study from a spatial analysis (a comparison between sites), not from time series. To reduce the ambiguity on causality in the sentence, we will rephrase in the following way:

'…patterns of gross primary and net ecosystem productivity versus $N_{dep}$ were non-linear, with no further growth responses at high $N_{dep}$ levels ($N_{dep} > 2.5$–3 g (N) m$^{-2}$ yr$^{-1}$) but accompanied by increasingly large ecosystem N losses by leaching and gaseous emissions.'

Line 140: add fires and insect attacks here.

'…Severe storms, fire outbreaks and insect infestations may have a similar effect'

Line 145: in some regions, e.g. in N. Europe, there are many N-fertilizer experiments.

We will add the following two references after '…*manipulation plots…*' in the sentence:

Nohrstedt, H.-Ö.: Response of coniferous forest ecosystems on mineral soils to nutrient additions: a review of Swedish experiences, Scand. J. Forest Res., 16, 555–573, 2001.

Saarsalmi, A. and Mälkönen, E.: Forest fertilization research in Finland: a literature review, Scand. J. Forest Res., 16, 514–535, 2001.

Line 216: you write even, but maybe mean seven?

No. We mean this (and will correct thus) :

'…baseline BASFOR runs were produced for all 31 forest sites of the network, **including also those stands** for which the model was not calibrated…'

Line 266-268: the use of another definition of NUE is widespread; I understand that you want to use an acronym, but it is unfortunate to use one that commonly has another meaning.

We agree it is best to avoid confusion, and we therefore suggest to change NUE (N use efficiency) to NUPE (N uptake efficiency) in the text, tables and figures. We will change the sentence lines 272-273, to further dissociate NUPE from NUE:

'…Note that i) NUPE is a different concept from the nitrogen use efficiency (NUE), often defined as the amount of biomass produced per unit of N taken up from the soil, or the ratio $NPP/N_{upt}$ (e.g. Finzi et al., 2007), and ii) …'

May I also suggest that you use Nmob = N mobilized, rather than Nmin, which means that you overlook organic N compounds as N sources.

Since the model does not account for dissolved organic N pools and uptake by trees, we believe it is preferable to stick to NMIN, which is explicit, rather than use NMOB which might imply otherwise.

Line 373: do you have clear evidence that Nmin does not change over time? Other authors discuss N oligotrophication and report that runoff of mineral N from forest decreases.

The soil mineral N concentration and leaching/export do change over time in the model, as shown and discussed above in Fig. R1 in our response to the Referee's comments on observed long term decreasing trends in forest N losses. Our sentence on line 373 ('*Since there is no significant long term (multi-annual) change in NMIN*') was indeed slightly misleading; by this we did not mean that mineral N concentrations and loss fluxes were stable over multi-decadal time scales. Rather, since NMIN is the soil inorganic N pool (g (N) $m^{-2}$), mineral N is transient and does not accumulate in the model because it leaves the root zone and effectively disappears (except for the fraction that is taken up by trees). There are inter-annual changes in NMIN, but the rate of change of the mineral N pool dNMIN/dt (g N $m^{-2}$ $yr^{-1}$) is insignificant compared with the annual rates of $N_{dep}$, $N_{upt}$, $N_{leach}$ and $N_{miner}$. We will therefore rephrase this sentence:

'…*Since the modelled long term (multi-annual) changes in the transient NMIN pool are negligible compared with the magnitudes of the N input and output fluxes, the dNMIN/dt term is not represented as an arrow in the budget plots, and the total mineral $N_{supply}$ … is basically balanced by N uptake… and losses…*'

Line 402: it is interesting to learn in which direction the non-linearity develops.

In Fig. 3 of the paper we presented the differences in modelled CSE between sites, which were plotted versus the age of the different sites. Since many other factors differentiate our forest sites apart from age, the CSE trends versus stand age could have been affected by co-varying factors. In Fig. R5-A, shown below, modelled CSE is plotted for all sites as a function of time elapsed since the stands were established, and similar trends are reproduced as in Fig. 3 of the paper, i.e. a decrease in modelled CSE from 25-35% in the age class 30-60 yrs down to around 20-25% for the stands older than 100 yrs.

The non-linearity we mention on line 402 is related to the increase with age of the $R_{het}$/GPP ratio, shown in Fig. R5-A. We will add Fig. R5 to the supplement (as Fig. S10) and rephrase our sentence on lines 400-402 of the manuscript:

'…*in the model, $R_{eco}$ in 30 to 60-yr old stands represents a smaller fraction of GPP than in mature stands*. From Eq. (1) it can readily be shown that CSE = 1 - $R_{aut}$/GPP - $R_{het}$/GPP, which is roughly equivalent to 0.5 – $R_{het}$/GPP, since $R_{aut}$ is constant and approximately 0.5 for all species in the model. By contrast, BASFOR predicts that the $R_{het}$/GPP ratio increases steadily with age at each site, after the initial establishment phase (Fig. S10-A). This induces a decline in modelled CSE from 25-35% in the age class 30-60 yrs down to around 20-25% for the older forests (Fig. S10-B). This also implies a non-linearity developing over time of GPP versus soil and litter layers C pools, since $R_{het}$ is assumed to a linear function of fast and slow C pools in litter layers and SOM. *Lifetime CSE values are slightly smaller*…'

[Figure]

Figure R5. *(= additional Fig. S10 in revised Supplement)*. Modelled time courses for all forests of the study of (A) the ratio of heterotrophic respiration ($R_{het}$) to gross primary productivity (GPP) and (B) the carbon sequestration efficiency (CSE = NEP/GPP). Short term excursions are related to thinning events.

Line 525: BASFOR may be mechanistic, but the vital interactions are not discussed and clarified in the description of the model.

We will include a more thorough description of the model (imported from the Part I paper), as indicated above in response to one of the Referee's general remarks.

Line 595: it is unclear if internal N supply is a component of soil fertility.

Internal N supply reflects the overall ability of soil microorganisms to mineralize dead organic matter and deliver plant available N. Both abundance and diversity of microbial and fungal communities are required to optimize SOM mineralization, but these depend on many factors that are not treated explicitly in the model, which uses empirically optimized SOM mineralization potentials. We will therefore add 'internal N supply' to the list on line 595, in the sense that site-specific limitations to SOM mineralization are not explicitly accounted for in the model.

Line 616: more thorough discussions about optimal allocation theory, especially C-N interactions, are found in, e.g., Franklin et al. (2012, Tree Physiol. 32,648- ).

A substantial discussion of optimal allocation theory is well beyond the scope of this paper. However we agree it makes sense to include a reference to Franklin et al. (2012), as well as Du and de Vries (2018) and Templer et al. 2012), in the introduction, line 100:

'…The questions of the allocation and fate of both the assimilated carbon (Franklin et al., 2012) and deposited nitrogen (Nadelhoffer et al., 1999; Templer et al., 2012; Du and de Vries, 2018) appear to be crucial. It has been suggested that $N_r$ deposition plays a significant role…'

Additional references:

Du, E. and de Vries, W.: Nitrogen-induced new net primary production and carbon sequestration in global forests, Environ. Pollut., 242, 1476–1487, https://doi.org/10.1016/j.envpol.2018.08.041, 2018.

Franklin, O., Johansson, J., Dewar, R.C., Dieckmann, U., McMurtrie, R.E., Brännström, Å and Dybzinski, R.: Modeling carbon allocation in trees: a search for principles, Tree Physiol., 32, 648–666, https://doi.org/10.1093/treephys/tpr138, 2012.

Templer, P.H., Mack, M.C., Chapin, F.S. III, Christenson, L.M., Compton, J.E., Crook, H.D., Currie, W.S., Curtis, C.J., Dail, D.B., D'Antonio, C.M., Emmett, B.A., Epstein, H.E., Goodale, C.L., Gundersen, P., Hobbie, S.E., Holland, K., Hooper, D.U., Hungate, B.A., Lamontagne, S., Nadelhoffer, K.J., Osenberg, C.W., Perakis, S.S., Schleppi, P., Schimel, J., Schmidt, I.K., Sommerkorn, M., Spoelstra, J., Tietema, A., Wessel, W.W. and Zak, D.R.: Sinks for nitrogen inputs in terrestrial ecosystems: A meta-analysis of $^{15}$N tracer field studies, Ecology, 93, 1816–829, https://doi.org/10.1890/11-1146.1, 2012.

Line 716: It is OK to cite Fog here, but why not cite authors, which discuss similar phenomena in forests (like Berg & Matzner 1997 Environ. Rev. 5, 1- ).

We will also cite Berg and Matzner here. Additional reference:

Berg, B. and Matzner, E.: Effect of N deposition on decomposition of plant litter and soil organic matter in forest systems, Environ. Rev., 5, 1-25, https://doi.org/10.1139/a96-017, 1997.

Lines 743-744: below-ground autotrophic respiration does not exactly follow photosynthesis, but is also affected by seasonality in C below-ground allocation (Högberg et al. 2010 New Phytol. 187, 485- ).

This sentence will be rephrased as follows:

'…Since the below-ground autotrophic (root and rhizosphere) respiration component is regulated to a large extent by photosynthetic activity (Collalti and Prentice, 2019), as well as seasonality in below-ground C allocation (Högberg et al., 2010), and contributes a large part of Rsoil on an annual basis…'

Line 780: what is the difference between fertility and nutrient availability in this context?

Fertility includes, but is not limited to, the pool of nutrients available in the soil. Legout et al. (2014) phrase it this way: 'The definition of the chemical fertility of forest ecosystems should not be limited to the pool of plant available nutrients in the soil but must also integrate the cycling and recycling of nutrients characteristic of biogeochemical cycling'

Legout, A., Hansson, K., Van der Heijden, G., Laclau, J.-P., Augusto, L. and Ranger, J.: Fertilité chimique des sols forestiers: concepts de base (in French), Revue forestière française, 4–2014, 413–424, https://doi.org/10.4267/2042/56556, English translation available at http://mycor.nancy.inra.fr/ARBRE/wp-content/uploads/2015/02/SP_4_Chemical-fertility-offorest-soils-basic-concepts.pdf, 2014.

All texts to Figures and Tables should be self-explanatory. Thus, acronyms should thus always be explained in these texts.

We will review the text in each figure caption and provide explanations where required.

Figures 3-6 and 8: these take some time to comprehend. A reader will need some guidance. And the text in the boxes are difficult to read and understand.

We will provide more explicit captions for these figures.

Figures 9 & 10: the texts by the symbols are difficult to read as sometimes they come on top of each other.

We will try to remediate this issue, but this might be tricky since the symbols also overlap and the labels can't be moved too far from their symbol.

**Summary**: Flechard et al. use a "meta-modelling" analysis of forest C fluxes and balance to examine the response of these processes to N deposition for ~30 sites (22 forests). They confirm that estimates of C gain from N deposition are smaller if environmental drivers are considered first.

**General comments**:

Overall, the analysis seems generally reasonable, and broadly supports the past re-analysis of forest C gain from N deposition by Sutton et al. (2008), which showed a much smaller C gain than imputed from the widely-critiqued Magnani et al. (2007) study. This text seems a bit long for that main take-home, with a data set only a bit larger – though analyzed in greater detail that that earlier dataset. It would be nice to have somewhat more focus in parsing this overall NEP response (i.e., more GPP vs less Reco or Rh?) beyond the surprisingly large reported GPP response.

Extensive discussion space is used on C sequestration efficiency (CSE =NEP/GPP), though it's not apparent quite what this adds over more in-depth examination of the individual C flux responses that go into this ratio. In particular, discussion of mechanistic explanations for the N effects on GPP and Rh (or Rsoil) would seem to be more directly related here – i.e., to explain saturation of the GPP response (discussed reasonably), or suppression of decomposition as a substantial portion of the overall dC/dN response. The direct effects of N on decomposition process appears largely restricted to the last page of the Discussion, and they merit much greater attention earlier and throughout the manuscript.

The reason we discuss CSE variability extensively is that we use various estimates of mean CSE to provide the step from the calculated $dGPP/dN_{dep}$ (effect on gross assimilation) to $dNEP/dN_{dep}$ (effect on C sequestration). In a nutshell, this is key to understanding how the paper works. It is clear that nitrogen addition can impact both assimilation (C gain) and respiration (C loss), and we choose to treat the two steps separately: 1) the assimilation (GPP) step by BASFOR meta-modelling, because we are confident that the model is reasonably well calibrated and constrained for GPP; and 2) the respiration ($R_{eco}$) step appears to be less well understood in terms of its ecological controls, and CSE is a useful normalized indicator or proxy to describe the fraction of C assimilated that is not lost by respiration. Discussing $R_{eco}$ in more detail would be less handy because it scales with GPP.

The flux data show a much wider range of measurement-based CSE values than does the model, and therefore either A- the measurements are imperfect (measurement uncertainties are discussed in the Part I paper), and/or B- the model is imperfect as it does not reproduce the natural variability of observations (as discussed in both Part I and II of the study). Of course we know that both A- and B- are true to some extent; but the combination of both measurements and model(s) helps close knowledge gaps.

Once a $dGPP/dN_{dep}$ value was estimated by model-based normalization for non-nitrogen effects (step 1 of the approach), we had to rely (for step 2) on various mean estimates of CSE to translate the response to $N_{dep}$ of gross photosynthesis into a response of net ecosystem productivity or C sequestration. This was because the variability in $CSE_{obs}$ was not fully reproduced by the model, and faced with the uncertainty in individual CSE values, the reasonable approach was to test various mean CSE values and examine the plausibility of the different results (as shown in Table 2). Hence the extended discussion on the potential ecological controls of CSE, which do include a discussion of N effects on $R_h$. Therefore we do think that we have in effect provided *'…discussion of mechanistic explanations for the N effects on GPP and Rh (or Rsoil)…'* in the paper. An explicit outline of our approach was given on lines 587-597, which we believe summarizes the above arguments, and fully answers the Referee's question about why it is necessary to discuss CSE and its variability:

*'…The EC-based flux data suggest that the $N_{dep}$ response of forest productivity is clearer at the gross photosynthesis level, in patterns of (normalized) GPP differences among sites, than at the NEP level, where very large differences in CSE among sites lead to a de-coupling of $N_{dep}$ and NEP. The response of GPP to $N_{dep}$ appeared to be reasonably well constrained by both EC flux measurements and BASFOR modelling, which is why we chose to normalize GPP, not NEP.*

*The significantly better model performance obtained for GPP than for $R_{eco}$ and NEP (Fig. 6 in Flechard et al., 2019) likely reveals a relatively poor understanding and mathematical representation of $R_{eco}$ (especially for the soil heterotrophic and autotrophic components), and the factors controlling their variability among sites. The large unexplained variability in CSE and C sequestration potentials may also involve other limiting factors that could not be accounted for in our measurement/model analysis, since they are not treated in BASFOR. Such factors may be related to soil fertility, ecosystem health, tree mortality, insect or wind damages in the previous decade, incorrect assumptions on historical forest thinning, all affecting general productivity patterns…'*

The reported response of a saturation of the growth response to N dep, coinciding with an increase in N losses, fits exactly within expectations of N saturation theory (e.g., Aber et al. 1989, 1998, BioScience), which deserves more explicit recognition and discussion.

The Referee is right; these seminal papers by Aber et al. will be referred to in the revised paper. Text added to introduction, line 106:

'…*but did not increase woody biomass or aboveground NPP*. In fact, Aber et al. (1989) even predicted 30 years ago that the last stage of nitrogen saturation in forests, following long term exposure to excess $N_r$ deposition, would be characterized by reduced NPP or possibly tree death, even if during the early or intermediate stages the addition of N could boost productivity with no visible negative ecosystem impact beyond $NO_3^-$ leaching. In that initial theory, Aber et al. (1989) suggested that plant uptake was the main N sink and led to increased photosynthesis and tree growth, while N was recycled through litter and humus to the available pool; this fertilization mechanism would saturate quickly, resulting in nitrate mobility. However, observations of large rates of soil nitrogen retention gradually led to the hypothesis that pools of dissolved organic carbon in soils allowed free-living microbial communities to compete with plants for N uptake. A revision of that theory by Aber et al. (1998) hypothesized the important role of mycorrhizal assimilation and root exudation as a process of N immobilization, and suggested that the process of nitrogen saturation involved soil microbial communities becoming bacterial dominated, rather than fungal or mycorrhizal dominated in pristine soils.

Text added to introduction, line 154:

'…*implying that the net dC/dN response was likely non-linear,* in line with an overview of dC/dN response results from various approaches (De Vries et al., 2014a), possibly due to the onset of N saturation as predicted by Aber et al. (1989), and associated with enhanced acidification and increase sensitivity to drought, frost and disseases (De Vries et al., 2014b)."

Reference added to discussion, lines 549-550:

'…*the highly non-linear response depends on current and historical Ndep exposure levels, and on the degree of N saturation* (Aber et al., 1989, 1998)*, although other factors…*'

Additional references:

Aber, J.D., Nadelhoffer, K.J., Steudler, P. and Melillo, J.M.: Nitrogen Saturation in Northern Forest Ecosystems: Excess nitrogen from fossil fuel combustion may stress the biosphere, BioScience, 39, 6, 378-386, https://doi.org/10.2307/1311067, 1989.

Aber, J., McDowell, W., Nadelhoffer, K., Magill, A., Berntson, G., Kamakea, M., McNulty, S., Currie, W., Rustad, L. and Fernandez, I.: Nitrogen Saturation in Temperate Forest Ecosystems, BioScience, 48, 11, 921–934, https://doi.org/10.2307/1313296, 1998.

De Vries, W., Dobbertin, M.H., Solberg, S., van Dobben, H. and Schaub, M.: Impacts of acid deposition, ozone exposure and weather conditions on forest ecosystems in Europe: an overview, Plant Soil, 380, 1–45, https://doi.org/10.1007/s11104-014-2056-2, 2014b.

A less central suggestion: The authors state the importance of detailed site-level N deposition values over estimated modeled ones. While believable, this point would be supported more substantively by showing it directly, e.g., by comparing estimated v measured N deposition values, and quantitatively comparing dC/dN results for these two types of N deposition estimates.

In the Part I companion paper of this study, we provided extensive comparison and discussion of the $N_r$ deposition levels based on our in-situ measurements versus modelled deposition values from the EMEP chemical transport model (CTM). We believe it is unnecessary to repeat this comparison in the present paper (Part II) and we have referred to Part I for more details in Methods (Section 2.1). As to a comparison of dC/dN results based on $N_{dep}$ estimates from in-situ measurements versus CTM outputs, they would scale according to the slope (0.74) of the linear regression between the two $N_{dep}$ estimates (since the intercept is negligeable) (Fig. R6). Since $N_{dep}$ is lower by around 25% in EMEP results versus the in situ estimates, the dC/dN estimates obtained on the basis of EMEP $N_{dep}$ data would be approximately 33% larger than those obtained on the basis of in situ data, used in the paper.

[Figure]

Figure R6. Comparison of total $N_{dep}$ rates between in-situ measurement-based estimates and EMEP CTM outputs for the forest sites of this study.

Overall, the manuscript might be revised to reduce sometimes redundant-seeming extensive discussion of CSE, and provide greater and more direct emphasis on its novel insights (beyond Sutton et al. 2008, or classic N saturation theory). These points will be addressed in the detailed comments below.

**Detailed comments**:

**Abstract**

Line 65 – somewhere in the abstract, specify the number of sites included in this analysis

This will be added to lines 65-66:

'…in combination with eddy covariance CO2 exchange fluxes from a Europe-wide network of 22 forest flux towers…'

Line 67-71 – The reduction of dC/dN from considering factors other than N deposition was for GPP, not NEP, right? that should be clear in the abstract, which describes this response in terms of C sequestration, which generally aligns more closely with NEP. Similarly, be clear about which C cycle term yielded the 40-50 gC/gN response.

The sentence on lines 67-69 will be rephrased thus:

'…The response of forest net ecosystem productivity to nitrogen deposition ($dNEP/dN_{dep}$) was estimated after accounting for the effects on gross primary productivity (GPP) of the co-correlates by means of a meta-modelling standardization procedure, which resulted in a reduction by a factor of about 2 of the uncorrected, apparent $dGPP/dN_{dep}$ value…'

Line 71: $dC/dN$ will be changed to $dNEP/dN_{dep}$

Line 74 – text indicates that the $dC/dN$ response saturates above 2.5-3.0 gN/m2/yr "due to" leaching and other losses.. but the latter don't appear to be measured here? If this attribution is from the model analyses, do indicate that as a modeled result.

Leaching and other (gaseous) N losses were measured at some sites, and the results were shown and discussed in the companion (Part I) paper. The model also indicated increased N loss fractions at the upper end of the $N_{dep}$ range. However, we agree that the causality implied by 'due to' was inappropriate in this sentence, and we have proposed, in response to a comment by the other Referee (#1), to modify the sentence in the following way:

'…patterns of gross primary and net ecosystem productivity versus $N_{dep}$ were non-linear, with no further growth responses at high $N_{dep}$ levels ($N_{dep}$ > 2.5–3 g (N) m$^{-2}$ yr$^{-1}$) but accompanied by increasingly large ecosystem N losses by leaching and gaseous emissions.'

**Main text**:

Line 81-93. The cited references provide examples of experimental studies that indicate little or no increase in C sequestration from N addition, and might be presented as some of the conflicting evidence for a universal N-deposition induced C sink, rather than challenging the entire notion of this phenomenon in its entirety

We believe that we present references for both sides of the argument (A- nitrogen deposition is a major driver of C sequestration, versus B- nitrogen deposition affects C sequestration very little, versus C- anything in between) in a balanced way, which shows that the experimental evidence is conflicting . But it appears that some authors do question the notion entirely, e.g. Nadelhoffer et al. (Nature, 398, 1999) write '…*that elevated nitrogen deposition is unlikely to be a major contributor to the putative CO$_2$ sink in forested northern temperature regions.*'

Line 92. Is the Dezi et al. 2010 reference a model-based analysis of $dC/dN$? If so, clarify that it's different from the empirical approaches of the other studies

We will make it clear this was a modelling study:

'…to 121 (in a model-based analysis by Dezi et al., 2010), …'

Line 93-97. In this review of $dC/dN$ values – and throughout the manuscript – be clear as to which values pertain to which C pool (i.e., tree, soil, or whole ecosystem) or which specific C cycle processes.

The C pools considered will be specified for each reference:

'…ranging from 61– 98 for above-ground biomass increment in US forests (Thomas et al., 2010), 35–65 for above-ground biomass and soil organic matter (Erisman et al., 2011; Butterbach-Bahl and Gundersen, 2011), 16–33 for the whole ecosystem (Liu and Greaver, 2009), 5–75 (mid-range 20–40) for the whole ecosystem in European forests and heathlands (de Vries et al., 2009), and down to 13–14 for aboveground woody biomass in temperate and boreal forests (Schulte-Uebbing and de Vries, 2018), and 10–70 for the whole ecosystem for forests globally, increasing from tropical, to temperate to boreal forests (de Vries et al., 2014a; Du and de Vries, 2018).'

Line 107 – 120. It's not wholly apparent why this review of basic N balance processes is needed? That is, nearly every forest N budget shows that new N deposition supplies only a small fraction of plant annual N demand compared to

internal N recycling; the key (missing?) point is that the value of N deposition is that it may be acquired directly or at little energetic cost to plants, and can accumulate over time.

We agree, this is a key point that should be mentioned. We will add the following sentence on line 120:

Importantly, unlike other ecosystem mechanisms for acquiring N from the environment (resorption from senescing leaves, biological $N_2$ fixation, mobilization and uptake of N from soil solution or from SOM), the nitrogen supplied from atmospheric deposition comes at little or zero energetic cost (Shi et al., 2016), especially if absorbed directly at leaf level (Nair et al., 2016).

Additional references:

Shi, M., Fisher, J.B., Brzostek, E.R. and Phillips, R.P.: Carbon cost of plant nitrogen acquisition: global carbon cycle impact from an improved plant nitrogen cycle in the Community Land Model, Glob. Change Biol., 22, 1299–1314, https://doi.org/10.1111/gcb.13131, 2016.

Nair, R.K.F., Perks, M.P., Weatherall, A., Baggs, E.M. and Mencuccini, M.: Does canopy nitrogen uptake enhance carbon sequestration by trees?, Glob. Change Biol., 22, 875–888, https://doi.org/10.1111/gcb.13096, 2016.

Line 121. Add specific citations to this sentence critiquing "some previous estimates" for failing to account for factors other than N deposition.

We will add a reference to Magnani et al. (2007) on line 122.

Line 122-130. This section seems somewhat oversimplified in pitching its novelty: N deposition can, but does not necessarily covary with gradients of other environmental variables; this covariation often depends on the geographic region selected.

We agree that N deposition does not necessarily co-vary spatially with other environmental variables, and that is not what we meant. We meant that if there is co-variation then it must be factored out (we wrote, line 125: '…*if Nr deposition is co-correlated with any of these other drivers..*').

Line 126, we will change 'is usually' to 'can be' in '…*as is usually the case in spatial gradient survey analyses across a wide geographic domain…*'

The Magnani et al. (2007) simple regression analysis indeed failed to consider this covariation, but its problems seemed very effectively addressed by the Sutton et al. (2008) reanalysis, in demonstrating the need to consider variation in factors besides N deposition. Is the goal in this manuscript to do a similar analysis of tower-based C balance measurements in greater depth than that one? Other gradient analyses have also considered N deposition along with other environmental drivers, sometimes also considering nonlinear responses (e.g., Solberg et al. 2009, Thomas et al. 2010).

We are of course well aware that other gradient studies, such as these cited by the Referee, have investigated the multiple controls (including the potential covariation with $N_{dep}$) of forest productivity. We recognized in the Discussion (4.1 and 4.2) that i) our flux network dataset was much smaller compared with studies based on long term growth monitoring plots and large-scale forest inventories, and therefore unsuited to large-scale multiple regression-type analyses such as provided by e.g. Solberg et al. (2009); and ii) our final estimates of $dNEP/dN_{dep}$ were not significantly different from recent reviews. We will add references to such studies on line 143:

'…Altogether, these complex interactions mean that it is far from a simple task to untangle the $N_r$ deposition effect on ecosystem C sequestration from the impacts of climatic, edaphic and management factors, when analysing data from diverse monitoring sites situated over a large geographic area (Laubhann et al., 2009; Solberg et al., 2009; Thomas et al., 2010).'

Additional References:

Laubhann, D., Sterba, H., Reinds, G.J. and de Vries, W.: The impact of atmospheric deposition and climate on forest growth in European monitoring plots: An empirical tree growth model, Forest Ecol. Manag., 258, 1751–1761, https://doi.org/10.1016/j.foreco.2008.09.050, 2009.

Solberg, S., Dobbertin, M., Reinds, G.J., Andreassen, K., Lange, H., Garcia Fernandez, P., Hildingsson, A. and de Vries, W.: Analyses of the impact of changes in atmospheric deposition and climate on forest growth in European monitoring plots: A stand growth approach, Forest Ecol. Manag., 258, 1735–1750, https://doi.org/10.1016/j.foreco.2008.09.057, 2009.

Nevertheless, the use of flux tower data in our study provides reliable measurement-based estimates of NEP, or NECB if there are no disturbances, which provide more direct measures of net ecosystem-scale C storage, compared with inventory-based studies, which must rely on critical assumptions regarding below-ground biomass and SOM balance to derive NECB estimates. There is value in both approaches, and they complement each other. What our study lacks in terms of site number and statistical power, is compensated by high temporal resolution data for assimilation and respiration fluxes at two dozen sites. This has enabled us to calibrate our forest ecosystem model using a multiple constraint Bayesian procedure, yielding generic model parameters for each tree class (Cameron et al., 2018). We did not adjust model parameters 'manually' for each site in this study, but allowed the Bayesian calibration algorithm to optimize all parameters at once, constrained by an a priori range (probability density function) for each parameter, and by all available calibration data (C and N fluxes, C stocks, tree heights and diameters, LAI, etc) with appropriate uncertainty estimates on the measurements.

The re-analysis by Sutton et al. (2008) of the Magnani et al. (2007) flux tower dataset did show convincingly that covariation of climate and $N_{dep}$ needed to be considered, and we have acknowledged (line 535) that our analysis came to the same conclusion. However we wish to stress a few key methodological differences, which give our study the novelty value questioned by the Referee:

- Our approach is a model-enhanced data analysis, by which measurement-based GPP is standardized into GPP* by the model. This is by contrast to the purely model-based (data-free) analysis used with the Edinburgh Forest Model (EFM) in the Sutton et al. (2008) paper. Disentangling the effects of $N_{dep}$ vs. climate was then done simply by running the EFM n times (i.e. once for each of the n forest sites) and calculating a multiple linear regression of the lifetime average NEP vs. {lifetime average $N_{dep}$ + Temperature + Precipitation}.
- The role of soils was not analysed in Sutton et al. (2008), i.e. the EFM was run with site-specific soil data, but soil properties did not appear in the multiple regression.
- The meta-modelling approach of the present paper involved running BASFOR $n^2$ times at each site (i.e. $n^3$ in total for all n sites) for the climate and soil normalization runs, by swapping climate data, or soil physical properties, between sites.
- This means that the 2008 analysis was a non-mechanistic analysis that could – fundamentally – never rule out any of the correlated variables as being the main driver of variation in NEP, as we point out in 2.2.4; while the new analysis uses the strength of mechanistic modelling by actually doing an input-sensitivity analysis.
- The EFM had not at the time undergone any thorough, multiple contraint calibration (in contrast to BASFOR). EFM results for individual runs were not compared to data: all that was compared to data was the regression line of productivity vs. $N_{dep}$.
- Because the new (BASFOR) modelling is grounded in real data, the new paper can identify a key role for differences between sites in level of N-saturation, which was not possible before. The new analysis in terms of Carbon Sequestration Efficiency clarifies the important implications of the results.

To summarize, the novelty of the study (including both Parts I and II) lies in the methods employed (detailed measurements of dynamic C and N fluxes at each site; model-enhanced analysis and untangling of the flux data and inter-relationships), rather than in the key end results themselves (non-linear response of C sequestration to $N_{dep}$, decline of forest productivity at N-saturated sites), which admittedly have long been known. However, with the development and standardization of flux tower networks in Europe (ICOS) or worldwide (FLUXNET), we can envisage further such studies, at many more sites, using measurement-model fusion techniques.

Line 175-176. Elaborate on what exactly is meant by "soil C pools rely on various assumptions or empirical models for their estimation." Assumed and modeled soil C can often vary markedly from measured values; how well do these assumptions work?

By "soil C pools" we did not mean specifically SOM pools, but more generally below-ground C pools, including therefore also fine and coarse roots. Below-ground C pools are much more difficult to evaluate on the basis of measurements than above-ground stocks. SOM stocks are evaluated on the basis of soil cores, but often there is large spatial heterogeneity in soil depth, in vertical horizon structure, etc, and therefore we have to rely on the assumption that the spatial samping scheme is statistically representative of the whole ecosystem (or the flux tower footprint). The root C pools are evaluated on the basis of allometric relationships to the above-ground vegetation, which effectively are empirical models of the ratio of below-ground to above-ground tree C stocks. In short, we are not dealing with measurements per se, but with measurement-derived quantities, whose uncertainties need to be recognized and quantified for model evaluation and calibration. We will change '*soil C pools*' to '*below-ground C pools*' in the sentence, and add '… (e.g. flux partitioning procedure to derive GPP from NEE, allometric relations for tree C stocks, spatial representativeness of soil core sampling for SOM).' after '*…on the basis of measured data*', line 176.

Line 184. Specify the minimum and/or mean number of years of EC data are used to compute the C fluxes of interest here.

The mean number of years of EC data was 5 (site-specific details provided in Table S6 of the companion Part I paper).

'…the C datasets include multi-annual (on average, 5-year) mean estimates of NEP, GPP and $R_{eco}$…'

Line 202-204. Does this model's soil dynamics allow it to represent the inhibitory effect of N deposition on soil decomposition? If not, this point should receive explicit attention in the Methods and/or Discussion, on how this effect of N was considered in this analysis.

Indeed the model does not contain any such mechanism, and we will make this clear in the discussion on the effect of N addition on soil respiration, line 728:

'…Of the five afore-mentioned mechanisms potentially involved in the suppression of soil respiration by N addition, only the first one (control by N availability of the root/shoot allocation ratio) is functional in BASFOR, and therefore our simulations do not include the other inhibitory effects of excess N on mycorrhizal, fungal and bacterial respiration.'

Line 207-208. It's certainly difficult to reliably simulate N loss fluxes to DON and N2, and it's correspondingly understandable that this set of model-based estimates would not include them. However, when measured, these fluxes can dominate ecosystem N loss fluxes and should thus receive more attention as to the uncertainties introduced by their omission in these calculations.

We fully agree that these fluxes can be significant, or even dominant, as discussed in the companion Part I paper, which focused on uncertainties in the C and N fluxes derived from observations. We wrote explicitly in the present paper (lines 273-276) that these fluxes are not considered in BASFOR, but unfortunately there are not enough data to address the Referee's question about uncertainties from a modelling viewpoint. We propose to add the following sentence to the conclusion, line 785:

'…N_r losses, especially as $NO_3^-$ leaching. Further sources of uncertainty in our forest ecosystem model involve missing – but possibly large – terms of the N cycle, such as $N_2$ fixation, $N_2$ loss by denitrification, DON uptake by trees and DON leaching.'

Line 246 / Section 2.2.3 – why focus so much text here and the Discussion on the ratio of C sequestration _efficiency_, CSE, (NEP / GPP and similar), rather than NEP itself and its component parts (i.e., increased GPP? Suppressed Reco or Rh)? The Introduction provides no context or central questions for focusing on questions concerning the CSE ratio, and so this emphasis seems somewhat unexpected and extraneous here, and the lengthy text in the Discussion (~line 600-700)

We have addressed, in our opening paragraph and response to the Referee's general comments (see above), the rationale for the focus on the CSE term in this study. We have explained how the paper focuses first on the response of assimilation (GPP) to $N_{dep}$ and other factors (through meta-modelling), and second on the factors (including $N_{dep}$) controlling respiration or $R_{eco}$, through CSE by proxy. However, we agree that the CSE concept, which is central for the discussion but is not introduced until Section 2.2.3, should be mentioned in the introduction. We propose to do this at the end of the first paragraph, line 87, and refer to the companion Part I paper, in which the concept was first introduced:

'…actually sequestered in the ecosystem. Indeed, it is possible to view this ratio of NECB to GPP as the efficiency of the long term retention in the system of the assimilated C, in other words a carbon sequestration efficiency (CSE = NECB/GPP) (Flechard et al., 2019).'

Line 265 – late text (line 273) indicate N fixation wasn't considered, so be consistent with that point here

We are not sure what the Referee means. $N_2$ fixation is not considered in the model (as stated line 273), and therefore it follows we cannot account for $N_2$ fixation in our definition of NUE in Eq. 6-8.

Line 314-316 – identify what is "the broad pattern of GPP vs N dep. in Flechard et al. (2019)."

We will add a brief description of the pattern in this sentence:

'…relationships reported in Flechard et al. (2019), i.e. a non-linear increase and eventual saturation of GPP as $N_{dep}$ increases beyond a critical threshold, did not show any marked difference…'

Line 373-374 – in this N balance (N mineralisation + N dep – N plant uptake – N leach – N emissions), what about accumulation of N in soils or soil organic matter? Often a very large if not the largest sink.

There are several interesting N balances that could be examined. In the paper (lines 369-378), we focus on the balance of NMIN, i.e. inorganic N in the soils, and no other N-balance. One could also be interested in Nsoil (=NMIN + NLITT + NSOMF + NSOMS), which the Referee is referring to, or also Nsys (= Nsoil + Ntree).

We choose to study the NMIN balance because mineral N is the only source of N available to the trees in our model (BASFOR does not consider DON supply and uptake, as explained above in reply to Referee #1), and our objective was to understand how increased N-availability to trees from N-deposition affects growth, C storage and other GHG fluxes. The only processes that affect that NMIN-balance, in the model, are the ones that we mention, i.e. increases in the NMIN pool from Nminer and Ndep, and decreases from Nupt, Nleach, Nemission. We ignore adsorption of NMIN to soil particles and the reverse (release from soil particles back to the free NMIN pool) because those processes are assumed to be small and in a stable mutual balance, i.e. the amount of adsorbed (unavailable) mineral N is not likely to change much over the lifetime of the trees, as soil pH etc. do not fluctuate much. It thus seems reasonable to not include those in the model.

The referee is suggesting to study the long-term accumulation of N as a chemical part of soil organic matter (e.g. when tree senescence adds more organic matter than is mineralized, so soil organic N increases over time). This is indeed interesting in itself and calculated by BASFOR as the net change over time in NLITT+NSOMF+NSOMS.

However, we believe that this is not a separate process of direct relevance to the balance of NMIN and to the central question of the study (what is the impact of $N_{dep}$ on C sequestration). This soil organic N balance reflects the shift of organic matter from plants to soil, minus mineralization of that SOM, in contrast to mineralization itself which directly contributes to N availability and the supply of N to trees that enables C sequestration.

We will add a sentence line 369, before the description of the N budgets, to clarify this point:

'For nitrogen, by contrast to carbon, the focus of the budget diagrams is not on changes over time of the total ecosystem (tree + soil, organic + mineral) N pools. Rather, we examine in Fig. 2 and S4–S6 the extent to which $N_r$ deposition contributes to the mineral N pool (NMIN), which in the model is considered to be the only source of N available to the trees and therefore acts as a control of C assimilation and ultimately sequestration. In these diagrams for NMIN, *the largest (horizontal) arrows indicate the modelled internal ecosystem N cycling terms…*'

Line 457 – suggest "at the low_er_ N dep sites. . ." That is, 1.0 g/N/m2 is often considered elevated.

Agreed, we will write 'lower'.

Line 466 – clarify which "this set" is meant – higher or lower N dep group?

The answer to the Referee's question is provided on the following line (467), where we specify that the regressions were '*…either calculated over the whole range of 22 sites, or for a subset of 18 sites that excludes the four highest deposition sites (>2.5 g (N) m$^{-2}$ yr$^{-1}$)*'. (data are provided in Table 2 in italics or bold characters, respectively)

Line 468-485. These dC-GPP/dN values are simply _enormous_ and correspondingly difficult to fathom – even when reduced from 425 to 234 gC/gN! How do these compare to empirical NPP values? Presumably a 50% GPP to NPP efficiency would yield something like 212 to 117 gC/gN, still well beyond empirical NPP responses. How /why is it so large compared to eventual dC-NEP/dN response?

There are several arguments to consider in response to the Referee's question.

1- We show in the paper that it is misleading to use $GPP_{obs}$ directly in a regression versus $N_{dep}$ alone, that observation-based GPP must be standardized for climate, soil and age, and that only GPP* should be used for the purpose of calculating a response to $N_{dep}$. Therefore the figure of 425 g (C) g$^{-1}$ (N), cited above by the Referee and taken from Table 2, is precisely that: an uncorrected, overestimated response, that we show in Table 2 only for the purpose of comparison. The only correct number to consider here is the one derived from GPP*, obtained when $GPP_{obs}$ is corrected for climate, soil and age, i.e. 234 g (C) g$^{-1}$ (N) (see Table 2, as explained on lines 471-472).

2- From this dGPP*/dN$_{dep}$ slope, the Referee is right in saying that the model would assume a theoretical reduction of ~50% from GPP to NPP to account for autotrophic respiration, thus dNPP/dN$_{dep}$ would be of the order of 117 g (C) g$^{-1}$ (N). But NPP is still not NEP; there is C allocation to exudation and mycorrhizae, which ends up being respired too, and heterotrophic respiration of SOM from free-living microbes ($R_{het}$). Eddy covariance flux towers tell us that ecosystem respiration ($R_{eco}$) is around 70-90% of GPP, more precisely 75% on average in the case of our forest dataset, i.e. NEP is around 25% of GPP (CSE~25%) on average (though with large variability, as pointed out in the paper) *(See also Figure R5 above, provided in our response to Referee #1)*. This means that $R_{het}$ removes around 25% of GPP (on average) from NPP; i.e., from the above dNPP/dN$_{dep}$ estimate of 117 g (C) g$^{-1}$ (N), about 0.25*234 is further removed to account for $R_{het}$. The resulting dNEP/dN$_{dep}$ from this back-of-the-envelope calculation is 50-60 g (C) g$^{-1}$ (N), and comparable in magnitude to what we show in Table 2 from the proper analysis.

3- A ratio of the order of 0.5 for NEP/NPP is also suggested by the analysis by Du and De Vries (Environmental Pollution, 242, 1476-1487, 2018), who estimate that the part of global forest NPP that is supported by external N inputs is 3.48 Pg C yr$^{-1}$ (see their Table 7), while the corresponding figure for the net global forest biome C sink (NEP) is 1.83 Pg C yr$^{-1}$ (see their Table 9), i.e. a NEP/NPP ratio of 0.53.

4- If the ratios of NPP/GPP and NEP/GPP are both fairly constrained (by the literature and by flux towers, respectively) and mutually consistent in our analysis, and if the final dNEP/dN$_{dep}$ responses are considered plausible by the Referee, then the dNPP/dN$_{dep}$ response should be, too. Our model was calibrated (see Bayesian calibration paper by Cameron et al., 2018, to which we refer in both papers) using multiple constraints, including measured NEE and evapotranspiration (eddy covariance at all sites), soil heterotrophic respiration (where available), carbon stocks in above- and below ground tree pools, in soil organic matter, also tree heights and diameters at breast height (at different dates wherever available), leaf area index, soil water content, and soil N emissions. We believe that this multiple constraint approach should ensure that NPP is not massively over-estimated, based on the model's asumptions.

The Referee argues that the NPP response is much (unreasonably) larger than empirical responses, but empirical NPP estimates are commonly made through the proxy of biomass production (BP), whereby BP is assumed to constitute the largest fraction of NPP. However, field measurements in forests show substantial variation in the BP/GPP ratio (e.g. Vicca et al., Ecology Letters, 15: 520–526, 2012). Also, some studies have considered only above-ground (not total) NPP responses to N$_{dep}$, which would make the number smaller. In our study we chose to focus on the responses of GPP and NEP to N$_{dep}$, because we have measurement-based values for both terms, but not the response of NPP, due to the above uncertainties and because no reliable NPP measurements were available.

Line 495-502. These dC-NEP/dN values (~40-60 gC/gN) seem more consistent with empirical responses: to what extent are these values due to modeled plant vs soil C sequestration? Is the soil C sink from additional litterfall inputs or from suppressed decomposition?

This question is addressed in Fig. 4, which shows the fractions of the net ecosystem carbon balance (NECB) that are stored in above-ground tree compartments (CLBS: carbon in leaves, branches and stems), in roots (CR), litter (CLITT) and in soil organic matter (CSOM). The results are discussed on lines 408-418; clearly the model allocates most of the C storage to above-ground tree parts (woody biomass), but over a lifetime the fraction stored in wood vs. SOM depends on the age of the forest.

Line 535. Yes, the results here seem to confirm that of Sutton et al. (2008). How does this analysis provide additional insights beyond that one?

We have addressed this question in detail previously, in response to another comment by the Referee (see above).

Line 537. Provide citations for observations of N losses. Thresholds of 0.8 – 1.0 g N/m2/yr for N leaching have been reported commonly (e.g., MacDonald et al. 2003, Global Change Biology, and similar).

We will modify this sentence on line 537:

'… *Observations and model simulations both indicate that the N$_{loss}$ fraction of N$_{supply}$ increases with N$_{dep}$*, consistent with widespread observations of increasing NO$_3^-$ leaching above N$_{dep}$ thresholds as low as 1.0 g (N) m$^{-2}$ yr$^{-1}$ in European forests (Dise and Wright, 1995; De Vries et al, 2007; Dise et al., 2009), and exacerbated by large C/N ratios (> 25) in the organic horizons (Gundersen et al., 1998; MacDonald et al., 2002). Higher thresholds for N$_{dep}$ around 2.5 g (N) m$^{-2}$ yr$^{-1}$ (Dise and Wright, 1995; Van der Salm et al., 2007) typically indicate advanced saturation stages. *Thus, at many sites but especially those with N$_{dep}$ > 1.5–2 g (N) m$^{-2}$ yr$^{-1}$,…*'

Additional references:

De Vries, W., van der Salm, C., Reinds, G.J. and Erisman, J.W.: Element fluxes through European forest ecosystems and 1205 their relationships with stand and site characteristics, Environ. Pollut., 148, 501–513, https://doi.org/10.1016/j.envpol.2006.12.001, 2007.

Dise, N.B. and Wright, R.F.: Nitrogen leaching from European forests in relation to nitrogen deposition, Forest Ecol. Manag., 71, 153–161, https://doi.org/10.1016/0378-1127(94)06092-W, 1995.

Dise, N.B., Rothwell, J.J., Gauci, V., van der Salm, C. and de Vries, W.: Predicting dissolved inorganic nitrogen leaching in European forests using two independent databases, Sci. Total Environ., 1225 407, 1798–1808, https://doi.org/10.1016/j.scitotenv.2008.11.003, 2009.

Gundersen, P., Callesen, I. and de Vries, W.: Nitrate leaching in forest soils is related to forest floor C/N ratios, Environ. Pollut., 102, 403–407, https://doi.org/10.1016/B978-0-08-043201-4.50058-7 , 1998.

MacDonald, J.A., Dise, N.B., Matzner, E., Armbruster, M., Gundersen, P., Forsuis, M.: Nitrogen input together with ecosystem nitrogen enrichment predict nitrate leaching from European forests, Glob. Change Biol., 8, 1028–1033, https://doi.org/10.1046/j.1365-2486.2002.00532.x, 2002.

Van der Salm, C., de Vries, W., Reinds, G.J. and Dise, N.B.: N leaching across European forests: Derivation and validation of empirical relationships using data from intensive monitoring plots, Forest Ecol. Manag., 238, 81–91, https://doi.org/10.1016/j.foreco.2006.09.092, 2007.

Line 540. These responses are exactly as expected – i.e., The saturating response of ecosystem NPP to N deposition, and corresponding increase in N losses, are standard predictions of classic N saturation theory as originally proposed (e.g., Aber et al. 1989 & 1998, BioScience). Discuss how this work provides an advance over that prior set of expectations.

Our results are consistent with (or not significantly different from) previously published thresholds for early and advanced N saturation. In that sense the end results are not new. However, to our knowledge, this may be the first time such results have been published based on eddy covariance GPP and NEP datasets, complemented by in-situ N flux (deposition, emission, leaching) measurements. As mentioned earlier, the multiplication of flux towers worldwide holds much promise for generalized coupled C/N studies, but this can only be achieved if these eddy covariance tower sites are also equipped to quantify nitrogen inputs and losses to the same degree of accuracy as for $CO_2$.

A reference to Aber et al. (1989, 1998) will be added to line 542:

'…beyond which growth and C sequestration were not further increased or even reversed, as predicted in classical N saturation theory by Aber et al. (1989, 1998).'

Line 560-570. This paragraph states that the detailed, more-precise N deposition measurements improve calculation of dC/dN responses.. and while plausible, it should first be demonstrated how these estimates compare with the simpler alternative.

We have addressed this question in detail previously, in response to another comment by the Referee (see above, Fig. R6).

Line 580-582. This conclusion on Reco vs N dep does not seem to have been discussed in the Results?

The variability of $R_{eco}$ and CSE was discussed extensively in the companion (Part I) paper, and we agree that we need to add a reference to Flechard et al. (2019) on line 580. However, note that at this stage we do not draw any 'conclusion on $R_{eco}$ vs $N_{dep}$', as the Referee suggests, but merely point to the large unexplained variance in CSE and $R_{eco}$.

Line 585. Per above, the annual N input is small relative to annual N demand. But its accumulation over time can support a much larger fraction of N demand.

That is a valid point. We will add the following sentence to line 586:

'…measurement uncertainties and noise in C and N budgets. Conversely, the effect may be delayed and may manifest even after $N_r$ deposition levels have decreased, as the past N accumulation in soil may support later growth through enhanced N supply.'

Line ~590. This would seem one of several places to mention the effect of N on decomposition

This short section (lines 587-597) was written to introduce the later discussion items on potential drivers of the CSE (4.3), including nutrient limitation, N saturation, forest history and effects of N on soil respiration. This is why these items are mentioned here very briefly, as a preamble to the sections that follow (e.g. the effect of N on decomposition is discussed in 4.3.3). We will add the following sentence to line 597 to clarify the transition:

'…*all affecting general productivity patterns*. Since the observed variability in CSE is key to understanding and quantifying the real-world NEP response to $N_{dep}$ (beyond the relatively well constrained response of GPP in the model world), we explore some of the main issues in the following sections.'

Line 598 – 708. It's not apparent why so much emphasis is placed on carbon sequestration efficiency (CSE = NEP / GPP) rather than the component C fluxes (GPP, NPP, NEP, Rh). It seems somewhat redundant with these other responses, and a direct outcome of individual responses. What additional insights does it provide?

The CSE indicator (=NEP/GPP) was introduced in the companion (Part I) paper to make it apparent that:

1. Flux tower eddy covariance data indicate a very large range of the ratio $C_{sequestered}/C_{assimilated}$, which our ecosystem model did not reproduce, and which we believe most ecosystem models would fail to reproduce;
2. The unexplained variability in CSE could indicate large measurement uncertainties; this was discussed in some detail in the Part I paper;
3. And / or our mechanistic understanding (and therefore ecosystem models) is incomplete;
4. We believe both 2 and 3 cannot be ruled out, but by considering both measurement and model uncertainties side by side, we may further our understanding;
5. CSE is useful for interpreting differences in $R_{eco}$ between sites, because $R_{eco}$ scales with GPP, and we have a very large variability of climates (and soils) at the European scale in this dataset; therefore a normalized indicator can be compared, while absolute values cannot.

We have explained this approach in the present paper, in Methods (2.2.3, 2.2.4) and reiterated in Results (3.3, 3.4 and especially 3.5), and discussion (4.2). As discussed above in our response to an earlier comment by the Referee, it seems logical to proceed in two stages: i) response of GPP, then ii) response of NEP via the proxy of a mean CSE, since the model does not allow us fundamentally to understand differences in $R_{het}$ and $R_{aut}$ between sites.

Line 649. A large soil C stock doesn't necessarily indicate higher heterotrophic respiration responses – and can result from the opposite situation (i.e., lower Rhet allows more soil C to accumulate).

We agree with the Referee, this was an over-simplification and indeed misleading. We will rephrase thus:

'…The EN4, EN6, EN17 sites had the three largest soil organic contents (SOC, Fig. 9A), which may either have induced larger rates of heterotrophic respiration, or may instead indicate low-fertility wet soils where both assimilation and respiration are suppressed. However, EN4 has also been reported…'

Line 652. The history of N and S deposition at this site (EN8) indeed might be important. What about considering cumulative N deposition across the range of sites?

Unfortunately not all (in fact very few) measurement sites possess the historical depth of $N_{dep}$ (and also N leaching) measurements that are available at EN8. In a way we have considered cumulative $N_{dep}$ at all sites through the assumed historical curve of $N_{dep}$ as an input to the model (see Fig. R1 above, which will be added to the paper), but the temporal trends are model-derived and identical for all sites, though scaled for actual $N_{dep}$ measurements made around 2005-2010.

We believe our inclusion of Fig. R1 in the paper, and the accompanying description text (see above), in response to a comment by Referee #1, will clarify the temporal aspects of the modelling study, including the way historical $N_{dep}$ was handled.

Line 709 onward. This seems quite late for a first substantive mention of the direct effects of extra N on decomposition and belowground processes, often shown to be of comparable magnitude as many aboveground responses (e.g., Janssens et al. 2010, Frey et al. 2014).

Our paper's ultimate objective was to derive $dNEP/dN_{dep}$ estimates using a data-model fusion approach. The discussion of N effects on below-ground respiratory processes, although fully relevant and contributing to the overall discussion, was not the central theme. The occurrence of this specific discussion topic in the last part of the paper made sense with respect to the logical flow of the paper:

1. Describe model simulations for the C and N cycles (short-term and lifetime), as the foundation for the scenarios / input-sensitivity simulations used in 2:
2. Describe method and results for the model-based normalization of the GPP response to $N_{dep}$
3. Step from $dGPP/dN_{dep}$ to $dNEP/dN_{dep}$, using the CSE proxy
4. Discuss limitations and uncertainties of the approach
5. Discuss ecological controls of CSE variability

In addition: does the modelling approach consider these processes, or would it miss them?

We repeat here the response we made earlier to a similar question by the Referee:

*\*\*\*\*\*\*\*\*\*\*\* Indeed the model does not contain any such mechanism, and we will make this clear in the discussion on the effect of N addition on soil respiration, line 728:*

*'…Of the five afore-mentioned mechanisms potentially involved in the suppression of soil respiration by N addition, only the first one (control by N availability of the root/shoot allocation ratio) is functional in BASFOR, and therefore our simulations do not include the other inhibitory effects of excess N on mycorrhizal, fungal and bacterial respiration.' \*\*\*\*\*\*\*\*\*\**

Line 714. Add citation(s) for this "traditional theory of role of N..."54

We will refer to Alexander (1977). Additional reference:

Alexander, M.: Introduction to soil microbiology, 2nd ed., John Wiley and Sons, London, 467pp., 1977.

Line 737-742. This content seems more appropriate to the Methods, as a general data synthesis activity part of this study. Similarly line 743-763 seem more appropriate to Results.

The data were collected as part of the data synthesis activity and described in Methods (section 2.3.2, Table S7) of the companion (Part I) paper, to which we refer on line 746 of the present paper. We don't think it is necessary to repeat this information in Methods of the present paper. The short description on lines 743-763 does not actually describe results of the present study, but supplementary information that comes in support of our analysis and therefore fits better in the discussion section.

Line 775. NEP dC/dN of 40-50 is on the lower end of inventory? Many inventory-based assessments seem to show something in this range, with ~20-25 for trees and 20-25 gC/gN for soil.

We agree with the Referee, and will change the wording of this sentence accordingly:

'…and comparable with current estimates obtained from inventory data and deposition rates…